# Analysing the sensitivity of a blowing snow model (SnowPappus) to precipitation forcing, blowing snow, and spatial resolution

Ange Haddjeri[1], Matthieu Baron[1], Matthieu Lafaysse[1], Louis Le Toumelin[1], César Deschamps-Berger[2,4], Vincent Vionnet[3], Simon Gascoin[4], Matthieu Vernay[1], and Marie Dumont[1]

[1]Univ. Grenoble Alpes, Université de Toulouse, Météo-France, CNRS, CNRM, Centre d'Études de la Neige, Grenoble, France
[2]Instituto Pirenaico de Ecología, Consejo Superior de Investigaciones Científicas (IPE-CSIC), Zaragoza, Spain
[3]Meteorological Research Division, Environment and Climate Change Canada, Dorval, QC, Canada
[4]Centre d'Etudes Spatiales de la Biosphère, CESBIO, Univ. Toulouse, CNES/CNRS/INRAE/IRD/UPS, Toulouse, France

**Correspondence:** ange.haddjeri@umr-cnrm.fr

**Abstract.** Accurate snow cover modeling is a high stake for mountain regions. Alpine snow evolution and spatial variability result from a multitude of complex processes including interactions between wind and snow. The SnowPappus blowing snow model was designed to add blowing snow modeling capabilities to the SURFEX/Crocus simulation system for applications across large spatial and temporal extents. This paper presents the first spatialized evaluation of this simulation system over a 902 km$^2$ domain in the French Alps. Here we compare snow cover simulations to the spatial distribution of snow height obtained from Pléiades satellites stereo-imagery and to snow melt-out dates from Sentinel-2 and Landsat 8 time series over three snow seasons. We analyzed the sensitivity of the simulations to three different precipitation datasets and two horizontal resolutions. The evaluations are presented as a function of elevation and landform types. The results show that the SnowPappus model forced with high-resolution wind fields enhances the snow cover spatial variability at high elevations allowing a better agreement between observed and simulated spatial distributions above 2500 m and near peaks and ridges. Model improvements are not obvious at low to medium altitudes where precipitation errors are the prevailing uncertainty. Our study illustrates the necessity to consider error contributions from blowing snow, precipitation forcings, and unresolved subgrid variability for robust evaluations of spatialized snow simulations. Despite the significant effect of the unresolved spatial scales of snow transport, 250 m horizontal resolution snow simulations using SnowPappus are found to be a promising avenue for large-scale modeling of alpine snowpacks.

## 1 Introduction

Snow cover in mountainous terrains is characterized by an important variability at multiple temporal and spatial scales (Pomeroy and Gray, 1995; Clark et al., 2011; Anderson et al., 2014; Mott et al., 2018). This variability results from a large diversity of slopes, aspects, and elevations which induce a high variability of precipitation, wind, temperature, and radiation. Accurate snowpack modeling is key to describing this high variability in hydrological applications, climate projections, and hazard forecasting in mountainous terrains (IPCC, 2022; Morin et al., 2020).

The spatial variability of precipitation amount and phase at different scales is known to be one of the main sources of snowpack variability at the mountain range scale (1-100 km) (Clark et al., 2011). These patterns are specific to the regional topography and atmospheric flow, with regions of increased or decreased precipitation (Colle et al., 2013). At the slope scale (a few hundred meters), preferential deposition and snowfall enhancement are the main processes responsible for snowfall variability (before snowflake settlement) (Mott et al., 2018). The preferential deposition is the result of the interaction of the near-surface flow field with particle trajectories, creating areas of local accumulation (Lehning et al., 2008). Snowfall enhancement is a process in which surface flows are responsible for locally increased air moisture, leading to the formation or maintenance of low-level clouds, ultimately enhancing solid precipitations through a seeder-feeder mechanism (Bergeron, 1965; Choularton and Perry, 1986; Minder et al., 2011). Post-depositional processes finally, determine the snow variability at the slope scale and below. Blowing snow transport, snow sublimation, snow redistribution by avalanches, snow compaction, and melt are the main processes at work (Winstral et al., 2002; Bernhardt and Schulz, 2010; Mott et al., 2018). Blowing snow transport produces a mass transfer of snow from windward areas to leeward deposition zones. Additional mass loss occurs in those events due to the sublimation of suspended snow Liston and Sturm (1998); Yang et al. (2010). The heterogeneity of snow mass loss due to snow melt (Brauchli et al., 2017) and surface sublimation (Pomeroy et al., 1998; Strasser et al., 2008) also contribute to snowpack variability in alpine terrain.

The multi-scale variability of the alpine snowpack makes it challenging to represent in numerical models. Various approaches have been developed for that purpose. Numerical Weather Prediction (NWP) models at the kilometer scale or higher resolution models can explicitly represent the orographic precipitation patterns, and in some cases local low-level cloud formation, triggering snowfall enhancement (Lehning et al., 2006; Vionnet et al., 2017; Wang and Huang, 2017; Monteiro et al., 2022). However, the quantification of the precipitation is still impacted by important uncertainties with errors in precipitation amounts and phases, localization, and timing increased with coarse grid size (Clark et al., 2011; Ménard et al., 2019; Lundquist et al., 2019). Downscaling tools can be used to better represent the local meteorology from NWP models (Sen Gupta and Tarboton, 2016; Marsh et al., 2023; Bernhardt et al., 2010; Mital et al., 2022).

Due to the complex intertwining of snow variability processes, in particular blowing snow transport, the community found benefits in the development of dedicated high-resolution models coupled to (e.g. Vionnet et al., 2014; Sharma et al., 2021) or forced by atmospheric models (e.g., Lehning et al., 2006; Liston et al., 2007; Marsh et al., 2020; Baron et al., 2024; Quéno et al., 2023). The spatial evaluation of this type of system, dedicated to modeling part of the observed snow spatial variability is a challenge in itself. The snow modeling community has long been evaluating this kind of model locally. Evaluations can be carried out using direct measurements of the variable of interest, for example for blowing snow modeling, Vionnet et al. (2014) evaluated directly the simulated blowing snow fluxes of the Meso-NH/Crocus system against locally measured fluxes from a Snow Particle Counters (Sato et al., 1993). Amory et al. (2021) also compares simulated snow transport occurrence and mass fluxes to on-field observations in Antarctica. Similarly, Baron et al. (2024) compares the simulated snow transport occurrence and transport mass fluxes from the SnowPappus model to field observations. One of the main drawbacks of this method is the low number of direct snow transport observations found in literature, combined with the very high spatial variability of snow transport fluxes. As a result, spatialized snow simulations are more classically evaluated using measurements of snow

height and Snow Water Equivalent (SWE). Prasad et al. (2001) and Liston et al. (2007) compare snow simulations done using the SnowTran-3D system (Liston and Sturm, 1998; Liston et al., 2007) with SWE measurements from manual snow surveys. Marsh et al. (2020) compare simulated SWE from the Canadian Hydrological Model (CHM) to point observations. Mott et al.

(2008) evaluate the Alpine3D system (Lehning et al., 2006) comparing simulation with an interpolated map of snow height measurements. The use of interpolation methods in Mott et al. (2008) is justified by the high measurement density of the study area. However, in the majority of other study areas, in situ measurements are too sparse to characterize the complex spatial variability of alpine snow cover (Pepin et al., 2015; Bales et al., 2006; Vernay et al., 2022; Pomeroy et al., 2009). This situation explains the recent use of snow remote sensing (unmanned aerial vehicle, satellite) as a means of spatial evaluation

for large-scale snow cover models. Those methods are still limited by the low availability of remotely sensed variables, mainly snow cover fraction and derived variables or snow height. For example, Vionnet et al. (2021) compare CHM snow simulations (Marsh et al., 2020) with airborne Lidar snow height maps as well as Sentinel 2 snow cover maps. Very recently, (Quéno et al., 2023) used Lidar aquired snow height maps to evaluate the ability of the FSM2oshd framework (Quéno et al., 2023) of representing snow accumulation and erosion areas.

In the above-cited studies using the snow height, SWE, or snow presence related variables, the evaluation experiments are generally carried out using a single set of meteorological inputs. However, it is known that meteorological inputs explain an important part of the variability (Clark et al., 2011; Colle et al., 2013; Mott et al., 2018) and uncertainty Raleigh et al. (2015); Günther et al. (2019) in simulated snow height and SWE. While the evaluations of model inputs and simulated processes are usually conducted separately, they are often interdependent. Thus, it is difficult to determine if potential errors come from

the model or the input variable. For instance, a good simulation of snow height requires both accurate precipitation input and a robust snow evolution model. To robustly assess the value of a distributed snow model, the uncertainty in meteorological forcing must be considered.

To improve the snow spatial variability of the French snow modeling system, Baron et al. (2024) have developed a novel explicit blowing snow transport model, SnowPappus, coupled with the Crocus physical snow simulation model (Vionnet et al.,

2012). This model explicitly represents the vertically integrated saltation and suspension mass fluxes, as a function of wind speed and surface snowpack properties simulated by Crocus. This system is implemented on a regular grid to simulate snowpack evolution at a mountain range scale (approx. 100000 $km^2$), for multiple snow seasons. Baron et al. (2024) focused on the evaluation of the model at point scale, demonstrating its ability to accurately simulate blowing-snow fluxes and occurrence at observation stations. However, spatial evaluations of simulations on large distributed domains are currently missing.

The goal of this paper is to present a spatial evaluation of the SnowPappus blowing snow simulation framework (Baron et al., 2024), considering the uncertainty of precipitation estimates through different data sources that exhibit contrasted spatial patterns. The evaluation is based on Pléiades snow height maps (Deschamps-Berger et al., 2020) and Sentinel-2/Landsat 8 snow melt-out dates (Gascoin et al., 2019) and covers three consecutive snow seasons. We discuss the relative influence of precipitation sources, and blowing snow implementation on the simulated snow cover variability and how their interactions

can affect the evaluation of a distributed snow model. By using two contrasted horizontal resolution simulations, we also

analyze how the unresolved spatial scales of blowing snow can affect simulation results at a 250 m spatial scale. Finally, we emphasize the main challenges to be solved for more advanced evaluations of spatialized snow simulations.

## 2 Data and Methods

### 2.1 Study area

The simulation study site is located east of Grenoble in the French Alps (Fig. 1). It covers 902 $km^2$ including the Grandes-Rousses and Arves massifs. This area exhibits a complex topography, with elevations ranging from 700 up to 3900 meters a.s.l., a wide range of snow and temperature conditions, and different types of landscape features such as valleys, forests, alpine pastures, lakes, and glaciers. In this area, the local knowledge of precipitation flow patterns tells us that most winter storms come from North-Western flows usually giving increased precipitation on western slopes. The study area includes the
Col du Lac Blanc observatory (Guyomarc'h et al., 2019) where Baron et al. (2024) evaluated the blowing snow fluxes and occurrence simulated by SnowPappus. It counts 41 distinct ice patches or glaciers of various sizes as defined in the RGI Consortium (2017), forested areas, and three hydroelectric dams which underline the hydrological importance of the area.

As our simulation system is only able to represent snow in open areas, the forests, glaciers, lakes, and rivers in the study zone are masked in our simulations and observation datasets. For the forest mask, the BD FORET® V2 dataset has been used
(IGN©, 2021a) with a masking threshold of 25% of forested sub-pixel area. Waterways are masked following data from BD TOPO® (IGN©, 2021b). In the valley, the urban areas around the city of Bourg d'Oisans, Allemont, and Saint Michel de Maurienne are also discarded from the analysis.

### 2.2 Grid generation and spatial resolution

The simulations are based on two grid resolutions, 30 m and 250 m. Both simulation grids are built using as reference the
French 5 m RGE ALTI® digital elevation model (DEM) from IGN© (2021c). The 5 m high-resolution DEM is resampled using the average method to 30 m and 250 m horizontal resolutions using GDAL/OGR contributors (2023). At 250 m resolution, the 902 $km^2$ full simulation area is composed of 14 443 grid points. At 30 m horizontal resolution, the test zone is composed of 1 005 699 simulation points.

A geomorphons classification (or landform) is performed on the 250 m resolution DEM, for a more detailed analysis of simulation domain features. Geomorphons introduced by Jasiewicz and Stepinski (2013) are defined as the fundamental structural elements of a landscape. Here, we used the Whitebox Geospatial (2023) classification tool, an open-source software from Lindsay (2014). The algorithm is based on a line-of-sight analysis with similarities to the more classical topographic position index (TPI). This method identifies a set of topographic patterns corresponding to specific terrain attributes and landform
types. The advantage of this type of classification is that it allows topographic information to be conveyed mimicking the result of a classification process carried out by a human analyst. Another advantage of the geomorphons classification is that it

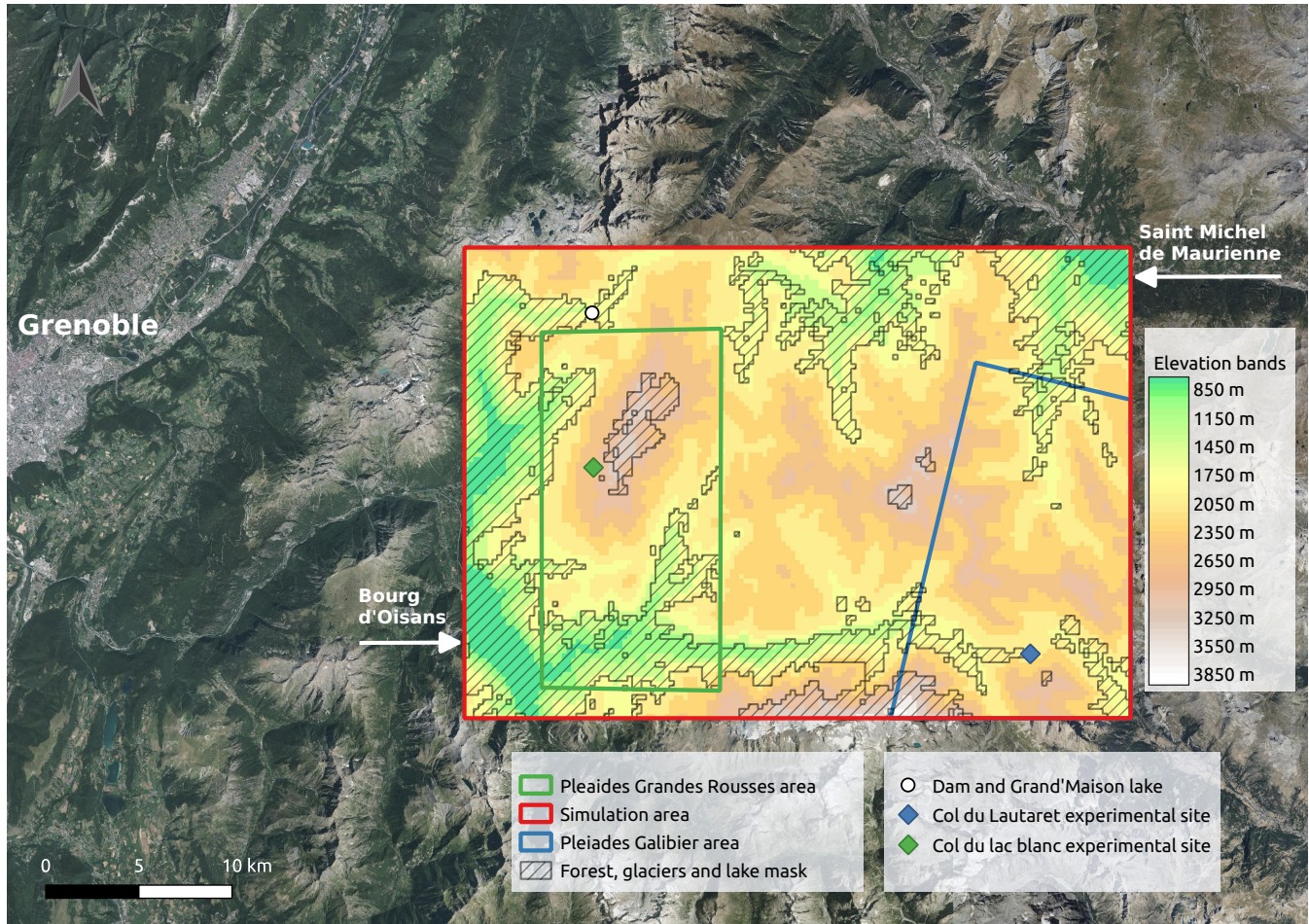

**Figure 1.** Map of the simulation domain (red) and evaluation areas (Pléiades green and blue, snow melt out date in red). Points of interest are shown as well as forest, glacier, and lake masks. The classification of 250 m pixels is visible with a 300 m step elevation in shades of blue. The summer aerial photography base map is from IGN© (2022).

adapts to the surrounding terrain and can lead to the identification of landform elements regardless of their scale. Jasiewicz and Stepinski (2013) identify the 10 most common landform elements used for geomorphons classification (peak (summit), ridge, shoulder, spur (convex), slope, hollow (concave), footslope, valley, pit (depression) and flat, illustration can be found in Fig. 3 of Jasiewicz and Stepinski (2013)). This allows us to generate a simple, intuitive, and scale-independent landform map of our simulation domain (Jasiewicz and Stepinski, 2013). Figure 2 illustrates the geomorphons classification result (called landform classification in the following) of each 250 m grid cell of our simulation domain in terms of the 10 most common landform elements. Figure 3 illustrates the elevation distribution of the landform elements of our simulation domain. No landform feature corresponding to Jasiewicz and Stepinski (2013) 'Shoulder', 'Footslope', and 'Flat' are found in our simulation domains. Fi-

nally, Appendix Fig. A1 demonstrates the good representativeness of the landform features frequency of our domain compared to the entire French Alps and Pyrenees.

For our analysis, each 250 m simulation grid cell is classified in terms of landform types (see Fig. 2) and into 300 m elevation bands (see Fig. 1).

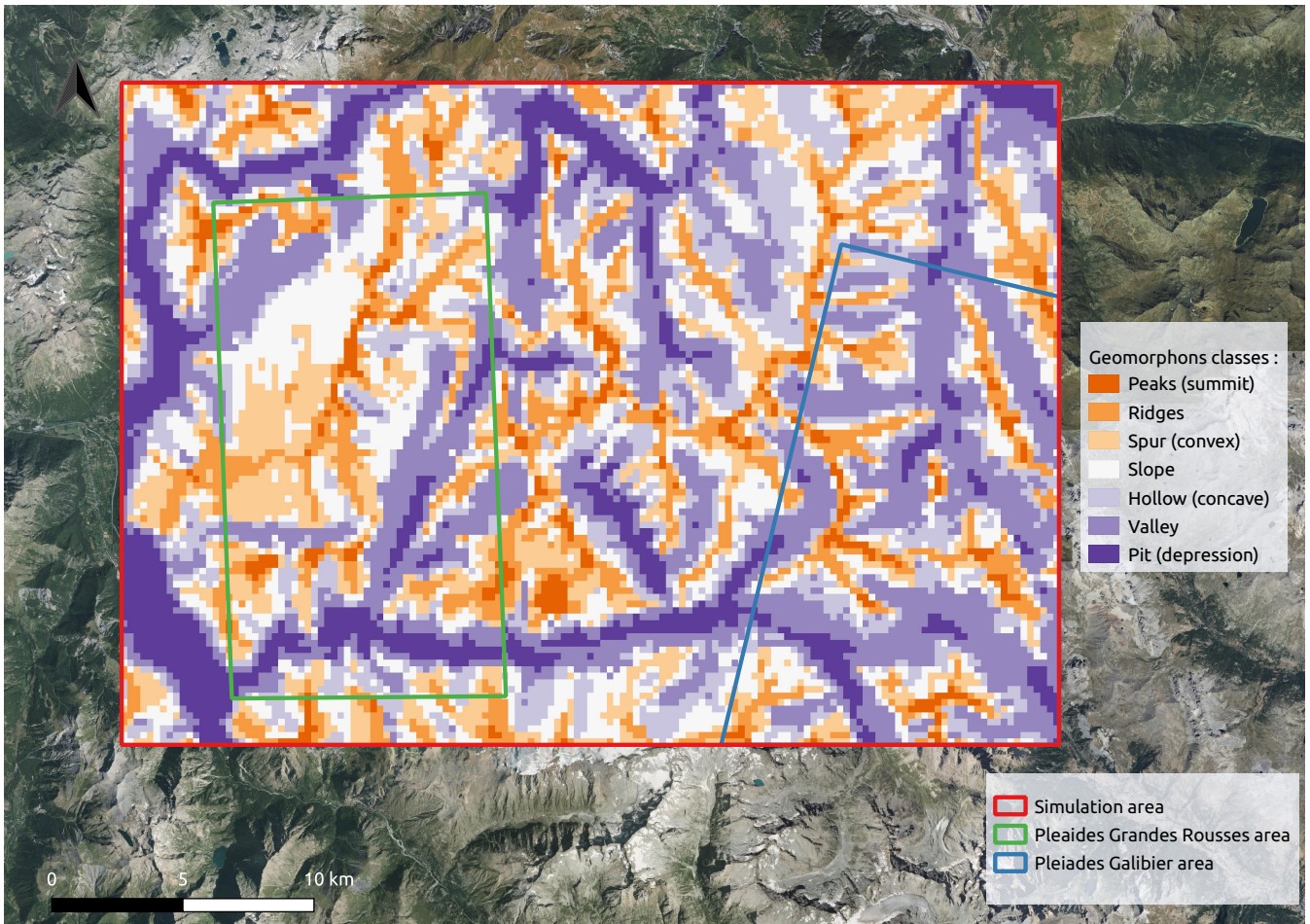

**Figure 2.** Map showing the result of the landform classification performed on our 250 m simulation domain. Each grid cell is classified in terms of the 10 most common landforms described by Jasiewicz and Stepinski (2013). (Peak (summit), ridge, shoulder, spur (convex), slope, hollow (concave), footslope, valley, pit (depression), flat) In our domain, "flat", "footslope", and "shoulder" landforms are not present. The summer aerial photography base map is from IGN© (2022).

## 2.3  Crocus snow model

This paper uses the well-established 1D snow model Crocus (Brun et al., 1989; Vionnet et al., 2012), used operationally by the French weather forecast agency (Météo-France), in support of avalanche hazard forecasting (Morin et al., 2020). This snow

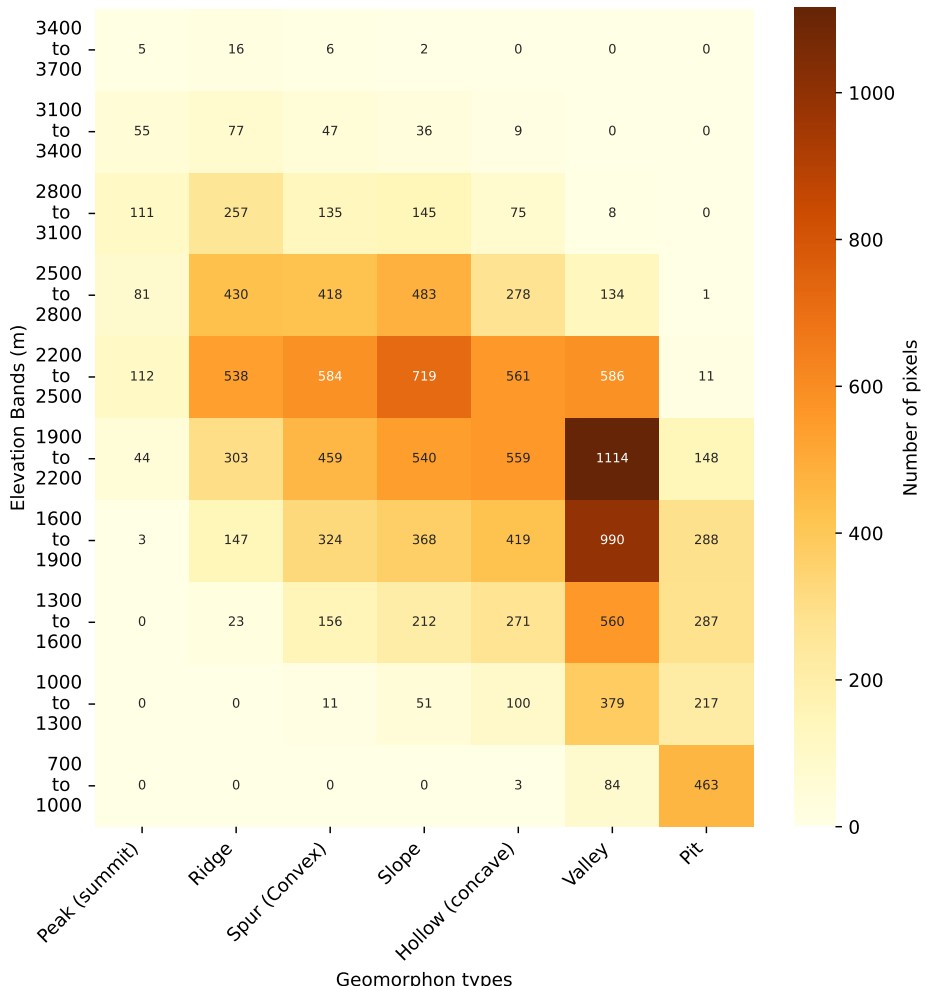

**Figure 3.** Distribution of simulation pixels in terms of 300m step elevation bands and the landform classification. The number of pixels corresponding to each class is shown.

model relies on a dynamical layering of the snowpack and an explicit representation of snow metamorphism and microstructure up to 50 snow layers (Vionnet et al., 2012). The Crocus snow model is coupled to the ISBA ground and land cover model and is embedded in the SURFEX modeling framework (Decharme et al., 2011; Masson et al., 2013). The snowpack is simulated on a squared grid of 250 m or 30 m spacing. Simulations are run over 3 years, starting from 02 August 2017 06:00 to 02 June 2020 06:00, using a 15-minute time step. The initial ground conditions are taken from a 10-year (2007-2017) simulation driven by the SAFRAN reanalysis (Vernay et al., 2022).

## 2.4 SnowPappus blowing snow model

The SnowPappus blowing snow model has been specially developed to describe snow variability due to blowing snow when the
Crocus model is applied in the range of 25 m to 250 m horizontal resolutions (Baron et al., 2024). SnowPappus model simulates
blowing snow occurrence, horizontal transport fluxes, and blowing snow sublimation rate on each grid cell as a function of 2D
atmospheric forcing and snow surface properties. Then, a blowing snow mass balance is used to quantify eroded or accumulated
snow amounts and modify the simulated snow profiles accordingly. Model methods and parameterizations used to represent
the different blowing snow processes are described in detail and discussed against existing literature in Baron et al. (2024).

SnowPappus can run over multiple snow seasons and very large domains (about $10^5$ km$^2$ at 250 m resolution or $10^6$ simulation points), in a reasonable computing time (less than half a day on a single computing node, depending on simulation outputs).

## 2.5 Meteorological forcings

### 2.5.1 SAFRAN simulations

As most spatialized applications of the Crocus snow model over the French Alps, our simulations are forced by the SAFRAN
meteorological reanalysis (Vernay et al., 2022). The SAFRAN analysis system uses as a background, vertical profiles of air
temperature, humidity, and the precipitation fields of the ARPEGE NWP system and assimilates near-surface meteorological
observations to generate, with an hourly time step, the typical sets of inputs required by a snow model such as Crocus (Durand
et al., 1993). The SAFRAN reanalysis provides these meteorological inputs over 23 climatologically homogeneous areas
(named massifs) covering the whole French Alps (Vernay et al., 2022). Here, only the SAFRAN reanalysis corresponding
to the Grandes-Rousses massif was used to build the simulation meteorological forcing. For a distributed use, the SAFRAN
analysis is linearly interpolated as a function of elevation, for each simulation point (as in Vionnet et al. (2016)) to match
the gridded geometry of our simulation domain. Solar radiations are projected according to the slope inclination and aspect
of each pixel and masked in the case of shadows from the surrounding topography similarly to Revuelto et al. (2018) and
Deschamps-Berger et al. (2022). In SAFRAN, all meteorological variables are assumed to be constant inside a massif for a
given elevation. Whereas large-scale biases are low thanks to the assimilation of numerous gauges observations (Vernay et al.,
2022), a large part of the spatial variability of precipitation remains unresolved under this assumption (Vionnet et al., 2016;
Quéno et al., 2016; Vionnet et al., 2019; Deschamps-Berger et al., 2022). Therefore, other precipitation estimates are also
considered in this study, respectively from AROME NWP system and ANTILOPE radar-based analysis. An illustration of the
mean daily SAFRAN precipitation spatial variability can be seen in Appendix Fig. B1 (a). This map informs us of the spatial
and inter-model variability and of the absolute precipitation values of each precipitation dataset.

### 2.5.2 AROME precipitation simulations

AROME is the high-resolution non-hydrostatic NWP system operated by Meteo-France for short-range forecasts. It uses a 1.3 km grid spacing (Seity et al., 2011). Due to high temperature and radiative biases of AROME in mountain areas (Quéno et al., 2020; Gouttevin et al., 2023), and to focus our study on the uncertainty of precipitation spatial patterns rather than on all meteorological forcing uncertainties, we only consider the total precipitation amount (liquid and solid) from AROME in our simulations. Thus, the precipitation phase is set to solid if the SAFRAN air temperature is lower than 274.15 K and liquid if the air temperature is above 274.15 K. Although more advanced and continuous phase functions are available in the literature (Froidurot et al., 2014; Vionnet et al., 2022), this choice was made for consistency between all simulations, as this threshold is also used by the SAFRAN system. SAFRAN analysis data are used for all the other forcing variables. An illustration of the mean daily AROME precipitations can be seen in Appendix Fig. B1 (d).

### 2.5.3 ANTILOPE precipitation simulations

ANTILOPE is an hourly precipitation estimation analysis product (rain rate), combining radar and rain/snow gauges with 0.01° (1 km) resolution (Champeaux et al., 2009). ANTILOPE precipitation fields are the kriging result of available gauge measurements, using radar-estimated precipitation fields as external drift. In a mountainous context, the skill of the product can be affected by a partial or total mask of the radar beams and other common radar measurement errors (Faure, 2017; Yu et al., 2018), although the use of observations improves the precipitation estimates compared to raw radar precipitation. Since ANTILOPE doesn't discriminate solid from liquid precipitations, we used information from SAFRAN to classify rainfall and snowfall as well as all the other forcing variables. The precipitation phase is set to solid if the SAFRAN air temperature is lower than 274.15 K and liquid if the air temperature is above 274.15 K. SAFRAN analysis data are used for all the other forcing variables. An illustration of the mean daily ANTILOPE precipitations can be seen in Appendix Fig. B1 (c).

### 2.6 DEVINE wind downscaling model

Wind fields are important drivers of turbulent and mass exchanges at the surface of the snowpack. Wind fields also profoundly determine the onset and evolution of drifting and blowing snow episodes. The interaction between mesoscale winds and local topography induces strong modifications both in terms of speed and direction that deeply influence the spatial variability of wind fields at the local mountain scale (Whiteman, 2000). Neither interpolated NWP systems operating with a km resolution (Seity et al., 2011) nor massif-scale SAFRAN reanalyses (Vernay et al., 2022) would permit us to take into account the local influence of terrain on wind fields. As the local scale wind patterns are crucial to determine high-resolution snow patterns (Musselman et al., 2015), we downscaled AROME wind fields (1.3 km grid spacing) down to a 30 m grid using the DEVINE downscaling model (Le Toumelin et al., 2022). This method leverages a convolutional neural network to emulate the behavior of the complex atmospheric model ARPS, previously run on a large number of synthetic topographies and in turn increase the spatial resolution of wind fields in complex terrain.

The DEVINE downscaling model has been previously evaluated over a large number of in-situ stations (Le Toumelin et al., 2022, 2023), including stations located in our study domain. It has been shown that DEVINE wind outputs can represent wind acceleration along ridges and summits, deceleration windward, and some deflection around topographic obstacles but can not represent more complex wind patterns such as thermal winds or recirculation areas.

## 2.7 Model experiments

To disentangle the impacts of the simulation of blowing snow from the precipitation forcing, 250 meter resolution simulations were run with and without the SnowPappus transport module and with the 3 different precipitation datasets, resulting in 6 different 250 m simulations. All simulations utilize the default options, the GM98 parametrization of blowing snow occurrence, and the flux limiter is activated (Guyomarc'h and Mérindol, 1998; Baron et al., 2024). In addition, blowing snow sublimation is activated using Simplified Blowing Snow Model-like equations (Essery et al., 1999; Baron et al., 2024). To investigate the role of unresolved transport variability of snow transport, 2 additional simulations were done at 30 m resolution with and without the SnowPappus transport module (see HR standing for High Resolution). Due to the higher numerical cost of these simulations and the increased complexity of downscaling and validating weather forcing at this resolution, 30 m resolution simulations were only run with the SAFRAN precipitation fields.

As the 30 m resolution simulations are used in this paper to evaluate the contribution of unresolved spatial variability in 250 m simulations, we do not evaluate 30 m simulations at their native resolution but after a resampling on the 250 m resolution grid using ESMF first-order conservative method (Earth System Modeling Framework et al., 2023).

Regardless of the simulation configuration, 10-year SAFRAN spinup simulations (without SnowPappus) were performed to produce the ground's initial condition.

A summary of all the simulation experiments is found in Table 1:

| Experiment name | Precipitation forcing | Blowing snow transport mode | Computing resolution |
|---|---|---|---|
| Safran | SAFRAN | No transport | 250m |
| Safran with transport | SAFRAN | Transport | 250m |
| Safran HR | SAFRAN | No transport | 30m |
| Safran HR with transport | SAFRAN | Transport | 30m |
| Arome | AROME | No transport | 250m |
| Arome with transport | AROME | Transport | 250m |
| Antilope | Antilope | No transport | 250m |
| Antilope with transport | Antilope | Transport | 250m |

**Table 1.** Overview of the different simulation experiments, their model configurations and naming. Important note, the computing resolution is different from the evaluation resolution. All simulations are evaluated at 250m resolution (see Sect. 2.8).

## 2.8 Reference dataset and evaluation methods

### 2.8.1 Snow melt-out dates

We use the annual snow melt-out dates (SMOD) distributed by Theia (level 3 product of the Snow collection Gascoin et al. (2019)). This product has a 20 m spatial resolution and indicates the last date of the longest continuous snow period. It is obtained by the linear interpolation in the time dimension of all the single-date (level 2) snow cover area products between 1 September and 31 August (Gascoin et al., 2019). These products are generated from Sentinel-2 and Landsat 8 images. The SMOD is given in days starting from September 1st. We resampled the SMOD to the 250 m resolution grid of our model using

the Earth System Modeling Framework et al. (2023) (ESMF) first-order conservative method.

We use the same definition to compute the SMOD from the model output. To compute the longest continuous snow period, the snow height is first averaged by day and then a snow height threshold of 20 cm is applied, as this value was found to be optimal in the sensitivity analysis of Deschamps-Berger et al. (2022) at the same horizontal resolution. For this threshold, the accuracy is found to be 91.5% (See Fig. 10 of Gascoin et al., 2019). The forest, glaciers, and lakes on the domain are masked

out both for the observations and simulations using the mask defined in Fig. 1 and Sect. 2.1.

### 2.8.2 The Pléiades stereo-imagery

In order to evaluate more directly the spatial variability of the simulated snow height, we used snow height maps derived from Pléiades stereo-images (Deschamps-Berger et al., 2020; Marti et al., 2016). In both cases, snow height is defined following a vertical axis. The Pléiades raw images have a 0.5 m horizontal resolution and are acquired on request. Snow height maps

are obtained by combining two surface elevations of the same area, one with snow and one snow-free. Each surface elevation map is derived from pairs or triplets of Pléiades images. The reference snow-off surface elevation is calculated from Pléiades stereo-images acquired on 28 September 2016 for the Galibier area and 24 October 2018 for the Grandes Rousses area. A winter image acquisition gives a second elevation map with snow on the ground. The difference between the two elevation fields provides a snow height map. The generated snow height map has a horizontal resolution of 2m. Two observation zones

have been selected based on available images. The first one covers 66 km$^2$ around the Col du Lac Blanc experimental site in the Grand-Rousse mountains. The second observation zone covers 168 km$^2$ around the Lautaret experimental site in the Galibier mountains. Two dates with snow on the ground have been analyzed for each observation zone. For the Grandes Rousses area, two images are available for a different year but at the same period of the season (Pléiades 1 on 13 May 2019 and Pléiades 2 on 4 May 2020). For the Galibier area, the two images give a view at different times of the same season (Pléiades 3 on 23 January

2018 and Pléiades 4 on 16 March 2018).

Due to the method used to reconstruct the snow height map, usual Pléiades observations contain a few no-data pixels where the quality of the observation is lower due to cliffs, steep slopes, clouds, or shaded areas. The snow height map was filtered to exclude values out of the [-0.5 m; 20 m] range. According to previous Pléiades snow maps evaluations (Deschamps-Berger et al., 2020), averaging 2 m Pléiades observation resolution to 250 m reduces the vertical snow height standard error

to approximately 0.3 and 0.4 m. Negative snow heights are kept as removing them would slightly positively bias the averages.

For comparison with snowpack simulations, the 2 m resolution Pléiades snow height map is resampled to the 250 m horizontal resolution grid using a conservative method, similarly to (Deschamps-Berger et al., 2020). The entire 250 m pixel is set to No Data if more than 70% of the 2 m pixels is No Data, this threshold ensures the minimum representativeness of the pixel's snow height.

The forest, glaciers, and lakes on the domain are masked both for the Pléiades observations and simulations using the mask defined in Sect. 2.1. An additional mask of the No Data pixels in the Pléiades data is applied to the simulations for accurate comparison.

### 2.8.3   Observation data summary

All observations are summarized in Table 2. For the comparison with simulations, the observed SMOD and snow height are
grouped by altitude in various elevation bands. The reference altitude used for the observation is the 250 m DEM described in Sect. 2.2. This elevation band classification is one of the most often used in the literature because the elevation is one of the main topographic drivers of snow cover variability (Grünewald et al., 2014; Vionnet et al., 2022; Monteiro et al., 2022; Tong et al., 2009; Deschamps-Berger et al., 2022; Vionnet et al., 2017). However, for a more advanced analysis of the snow cover spatial structure, we also present our results grouped according to the landforms presented in Sect. 2.2.

| Satellite observation | Observed area | Variable | Observation date | Native resolution | Aggregated resolution |
|---|---|---|---|---|---|
| Pléiades 1 | Grandes Rousses region (GR) | Snow height (m) | 13 May 2019 | 2 m | 250 m |
| Pléiades 2 | Grandes Rousses region (GR) | Snow height (m) | 4 May 2020 | 2 m | 250 m |
| Pléiades 3 | Galibier region (Glb) | Snow height (m) | 23 January 2018 | 2 m | 250 m |
| Pléiades 4 | Galibier region (Glb) | Snow height (m) | 16 March 2018 | 2 m | 250 m |
| Sentinel 2 A | Entire simulation domain | SMOD (days) | 1 September 2017 to 1 september 2018 | 20 m | 250 m |
| Sentinel 2 B | Entire simulation domain | SMOD (days) | 1 September 2018 to 1 September 2019 | 20 m | 250 m |

**Table 2.** Summary of satellite observations used in this study. The observed area, variables, resolutions, and dates are detailed for each observation. The aggregated resolution reminds the resampling operation done on the observation data before comparison with simulations.

## 2.9   Synoptic scores

To summarise the simulation and observation comparison, a combination of three scores has been chosen. The mean bias, standard deviation ratio, and Spatial Probability Score value (SPS) are used to quantify the similarity between the simulated and observed snow height distribution.

The mean bias (MB) defined in Eq. 1 describes the mean bias between the simulation and the observation over a given area.

$$\mathrm{MB}(\mathrm{S}) = \langle \mathrm{S} \rangle - \langle \mathbb{O} \rangle \tag{1}$$

where $\langle \mathrm{S} \rangle$ represents the mean value of the simulation distribution and $\mathbb{O}$ the observation distribution.

The standard deviation ratio (or $\sigma$ ratio in the following) (Eq. 2) is the ratio between the simulated standard deviation and the observed one.

$$\sigma \text{ ratio}(\text{S}) = \frac{\sigma(\text{S})}{\sigma(\mathbb{O})} \qquad (2)$$

with $\sigma(\text{S})$ the simulated standard deviation and $\sigma(\mathbb{O})$ the observed standard deviation.

The Spatial Probability Score (SPS) is a distance metric between two Cumulative Distribution Functions (CDF) defined in Eq. 3 as the quadratic discrepancy measure between the simulated CDF, noted F and $P_{obs}$ the empirical observed CDF.

$$\text{SPS}(F) = \int_{\mathbb{S}} \left( F(x) - P_{obs}(x) \right)^2 \mathrm{d}S. \qquad (3)$$

with $x \in S$, and $S$ the volume of possible values of the indicator of interest. This score is derived from the Continuous Ranked
Probability Score (CRPS) primarily used to evaluate probabilistic forecasts in ensemble forecasting (Candille and Talagrand, 2005). Here, it is used to compare two spatial distributions, similarly to the description in Goessling and Jung (2018), although using a non-binary probability field for the perfect observation. In short, this metric quantifies the similarity of two probability densities. It is close to the Wasserstein distance (Rüschendorf, 1985) used in (Vionnet et al., 2021) although here of order 2. A low SPS value means a small distance between the observed and the simulated distributions. Two identical distributions give
an SPS score of 0. The SPS unit is identical to the unit of the target variable (m for snow height, days for SMOD).

## 3 Results

### 3.1 Spatial variability of the snow height

This section compares observed and simulated snow height from the 8 experiments described in Table 1 to the 4 Pléiades observations (Table 2).
An example of simulated and observed snow height variability can be found in Fig. 4. In this figure, snow height is simulated on 13 May 2019 for the Safran, Arome, and Antilope precipitation forcings, and observed for the Grandes Rousses area on the same date. For each precipitation forcing, the subfigure shows from one side the impact of the SnowPappus blowing snow model on the simulated snow height, where we can see areas of increased and decreased snow height, most visible along the ridges and peaks, and in each left sub-maps the same simulation without snow transport. It can be observed that the primary
areas of model snow accumulation are located in regions of high observed snow height. However, the opposite is not true, many areas with high observed snow accumulation are not present in the simulations. Additional maps for the 13 May 2019 comparing resolutions can be found in Appendix Fig. C1.

### 3.1.1 Grandes Rousses area

In this section, we compare the two Pléiades snow heights images of the Grandes Rousses to simulations. For a quantitative
evaluation of these snow height maps, we summarized simulations and observations in terms of spatial distributions. For

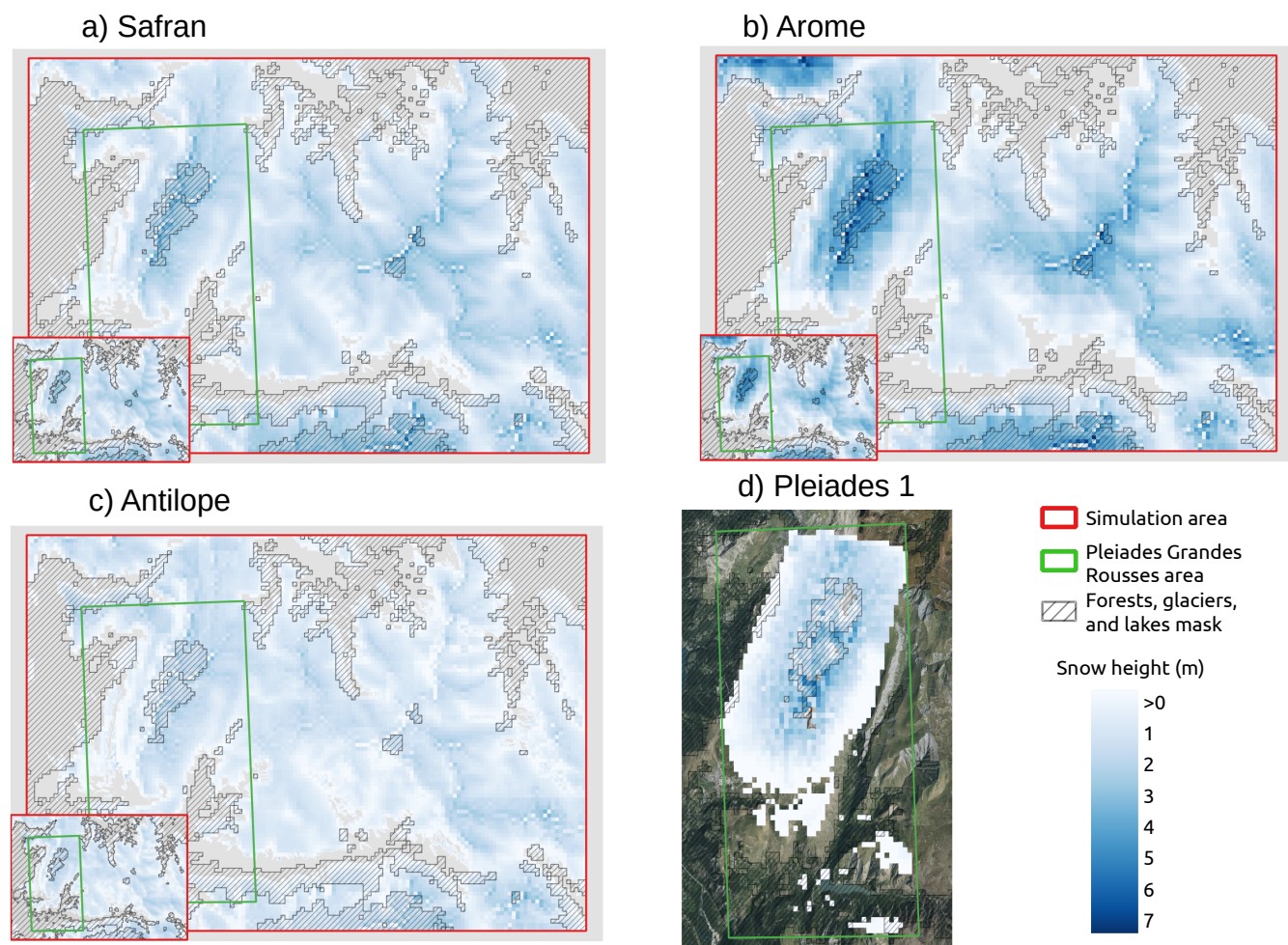

**Figure 4.** Maps illustrating raw simulation outputs and Pléiades observed snow height on the 13 May 2019 10:00. (a) Safran 250 m simulation with snow transport and without snow transport in the left sub-map. (b) Arome 250 m simulation with snow transport and without snow transport in the left sub-map. (c) Antilope 250 m simulation with snow transport and without snow transport in the left sub-map. (d) The Pléiades 1 observed snow height resampled to 250 m. The absence of snow in simulations is indicated in grey. For d) the summer aerial photography base map is from IGN© (2022).

each Pléiades subdomain, simulated snow heights are discretized using 300 m elevation bands. Each snow height distribution describes the internal snow height spatial variability within each elevation band. The associated spatial distributions obtained in the Grandes Rousses region (green box in Fig. 1) are compared in Fig. 5 (a-b) for 13 May 2019 (same data as Fig. 4) and 4 May 2020. The skill of these distributions is summarized through synthetic scores presented in Fig. 6 (a-b-c-d-e-f).

Figure 5 (a) shows a major impact of the precipitation input on the simulated snow height distribution, regardless of the use of snow transport in simulations. Precipitation inputs impact the simulated snow height distribution's shape, variance, and

mean value. Simulation with AROME precipitations leads to the highest snow height on average, for all elevation bands and the two evaluation dates. This result is especially notable for Fig. 5 (a) at the elevation band of 2500 to 2800 m on the 13 May 2019 evaluation. The AROME precipitation gives a mean snow height of about 3 m representing a positive bias compared to the Pléiades observations mean (1.87 m), consistently with Fig. 6 (a). The standard deviation of the AROME simulations overestimates Pléiades' standard deviation at low elevations on both dates Fig. 6 (b-e). It is more realistic between 2500 and 2800 m, and dependent on snow transport at higher elevations.

Simulations with the ANTILOPE precipitation, on the other hand, have a negative snow height bias compared to the observation. For the 2500 to 2800 m elevation band, the ANTILOPE simulations give a mean snow height of only 1.30 m. This negative bias tends to increase with altitude as illustrated in Fig. 6 (a). The standard deviation of ANTILOPE simulations is closer to observations than AROME simulations. Similar results are found for 4 of May 2020 (Fig. 5 (b), Fig. 6 (d-e-f)). For the SAFRAN and SAFRAN HR simulations, results are consistent between the simulation resolution (250m and 30m) but with varying skill between both evaluation years. At all elevations, the SAFRAN and SAFRAN HR simulations show a lower bias compared to observations than simulations forced by AROME and ANTILOPE. However, the SAFRAN mean value of 4 May 2020 (around 2 m) underestimates the observed snow height. A similar result is found at both resolutions and with or without snow transport. At most elevations, the standard deviation is similar to ANTILOPE forcing and close to the Pléiades observations.

The representation of snow transport is found to only impact snow height bias for the upper elevation band (with a reduction of the mean snow height). The fact that this impact is beneficial or detrimental depends on the choice of the meteorological forcing. Below this elevation band, the impact of snow transport on the bias is found shallow to nonexistent. Figure 5 (a) also shows that for the 3 upper elevation bands (above 2500m), SnowPappus (non-hatched violin plots) increases the simulated standard deviation on average by 63.7% between snow height distribution with and without snow transport (hatched violin plots). Figure 5 (b) (above 2500m) shows a similar standard deviation increase of 67.7% for the next year when adding snow transport in the simulations. The impact of snow transport on the simulated snow height is noticeable for all precipitation inputs. Figure 6 (b-e) shows that the increase of standard deviation (above 2500m) introduced by the transport module is sometimes overestimated comparatively to the observation ($\sigma$ ratio >1), depending on the date and precipitation forcing. Below 2500 m, the snow height distributions with and without snow transport, for a given precipitation input, are broadly similar.

Finally, SPS scores (Fig. 6 (c-f)) suggest that model performances are mostly modified by SnowPappus at the highest pixels of our domain and that simulation improvements or degradation depend on combinations of model setups (transport/no transport and precipitation forcings).

### 3.1.2 Galibier area

The second snow height evaluation is done in the Galibier Mountains area (blue box in Fig. 1). It uses two snow height maps derived from Pléiades stereo-images acquired in January and March 2018, in which snow cover extends to lower elevations than in the Grandes Rousses snow height maps. At both dates, a positive bias appears for the three lowest elevation bands, regardless of the forcing and decreasing with elevation (Fig. 8 (a-d)). For instance, the Pléiades observation provides a mean snow height of 1.25 m at the 1900-2200 m elevation band on 23 January 2018, while the simulation's mean values are all above 1.74 m

of snow. At these elevations, the standard deviation is underestimated by all simulations (0.71 m for the reference Pléiades observation, 0.16 m for SAFRAN and, 0.32 m for AROME), without any significant impact of the snow transport module (see std ratio Fig. 8 (b-e)).

    For the two upper elevation bands, the main features are consistent with the evaluations on the Grandes Rousses area: at both dates, the AROME simulations exhibit the highest mean snow height corresponding to a positive bias compared to observations,

while the ANTILOPE simulations exhibit the lowest snow height corresponding to a negative bias. Contrary to the previous section, SAFRAN simulations also overestimate snow height in the Galibier area. For all precipitation forcing, the standard deviation of simulations without transport (between 0.11 m and 0.25 m in January at the upper level) is much smaller than in the observed spatial distribution (0.87 m). When the snow transport is activated, a more realistic spatial variability is obtained, although it is still underestimated below 2800 m elevation. This conclusion remains valid on 16 March (Fig. 7 (b)). As for the

Grandes Rousses evaluation, the simulations with the AROME precipitation have the highest spatial variability in snow height at all elevations and for the two dates. Conversely, the simulations with SAFRAN have the smallest standard deviation across elevation.

    For the SPS score (Fig. 8 (c-f)), the ANTILOPE forcing produces very accurate scores ($\leq 0.1$ m) between 2200 and 2800 m, and for both dates, in contrast to the results obtained over the Grandes Rousses area. For the lowest elevation band, AROME

simulation with snow transport is the closest to observations with an SPS score of 0.4 and 0.3 m in January and March. The simulations forced by SAFRAN precipitation provide the spatial distributions the furthest to Pléiades observations. This is an opposite conclusion to the results obtained on the Grandes Rousses region.

### 3.1.3 Summary: Relative impacts of snow transport and precipitation forcing

    When comparing the simulation performance between both simulation regions and all dates, common and different patterns

arise. The impact of the simulation of snow transport on the bias is restricted to the upper altitude, close to mountain summits. For all simulations, higher average snow height are obtained without transport than with transport. However, the added value of snow transport can not be established on this criteria as the bias highly depends on the precipitation forcing.

    In all simulations, lower snow heights are obtained with ANTILOPE precipitation and higher snow heights with AROME precipitation, with increasing differences with elevation. However, upon examining the various bias scores (as shown in Fig. 6,

8, 10 (a-d), it is clear that no precipitation forcing produces a snow height simulation that is systematically less biased than the

others over the different evaluation dates and areas. Across all elevation ranges, the mean driver of simulation bias appears to be the precipitation dataset choice.

The addition of snow transport increases the simulated standard deviation of snow height of 0.25 and reduces the SPS value closer to the observation. The impact of transport on scores is mainly above 2500 m with a slope increasing with elevation. For these upper elevation bands, the snow transport prevails in the choice of the precipitation dataset on this criteria. For the lower elevation bands, the main driver of simulation standard deviation appears to be the choice of the precipitation dataset. The simulations using AROME precipitation have the biggest standard deviation value of all simulations.

### 3.1.4 Impact of subgrid variability

The effect of unresolved subgrid variability can be analyzed by comparing SAFRAN and SAFRAN HR simulations. Simulation maps illustrating the expected subgrid variability can be found in Appendix Fig. C1 (a-b-c). At the 30 m grid resolution, the spatial variability of the topography (SAFRAN HR without transport vs. SAFRAN without transport) reduces the snow height and increases the 250 m scale spatial variance at high elevations for the Grandes Rousses evaluations. The impact is smaller for the Galibier assessment. In Fig. 6, 8, the difference in the different scores between a simulation with and without snow transport is smaller for simulations computed at a 30 m resolution, compared to the 250 m computed simulations (orange dashed and continuous lines are less distant or equivalent to the green dashed and continuous lines). Finally, it is observed that SAFRAN HR and SAFRAN simulations are relatively close in terms of snow height bias, $\sigma$ ratio, and SPS. Therefore, despite the significant impact of the simulation resolution at the process level, it is not obvious that the 250 m simulations would provide less accurate snow height spatial distributions than simulations computed at higher resolution and resampled on a 250 m grid.

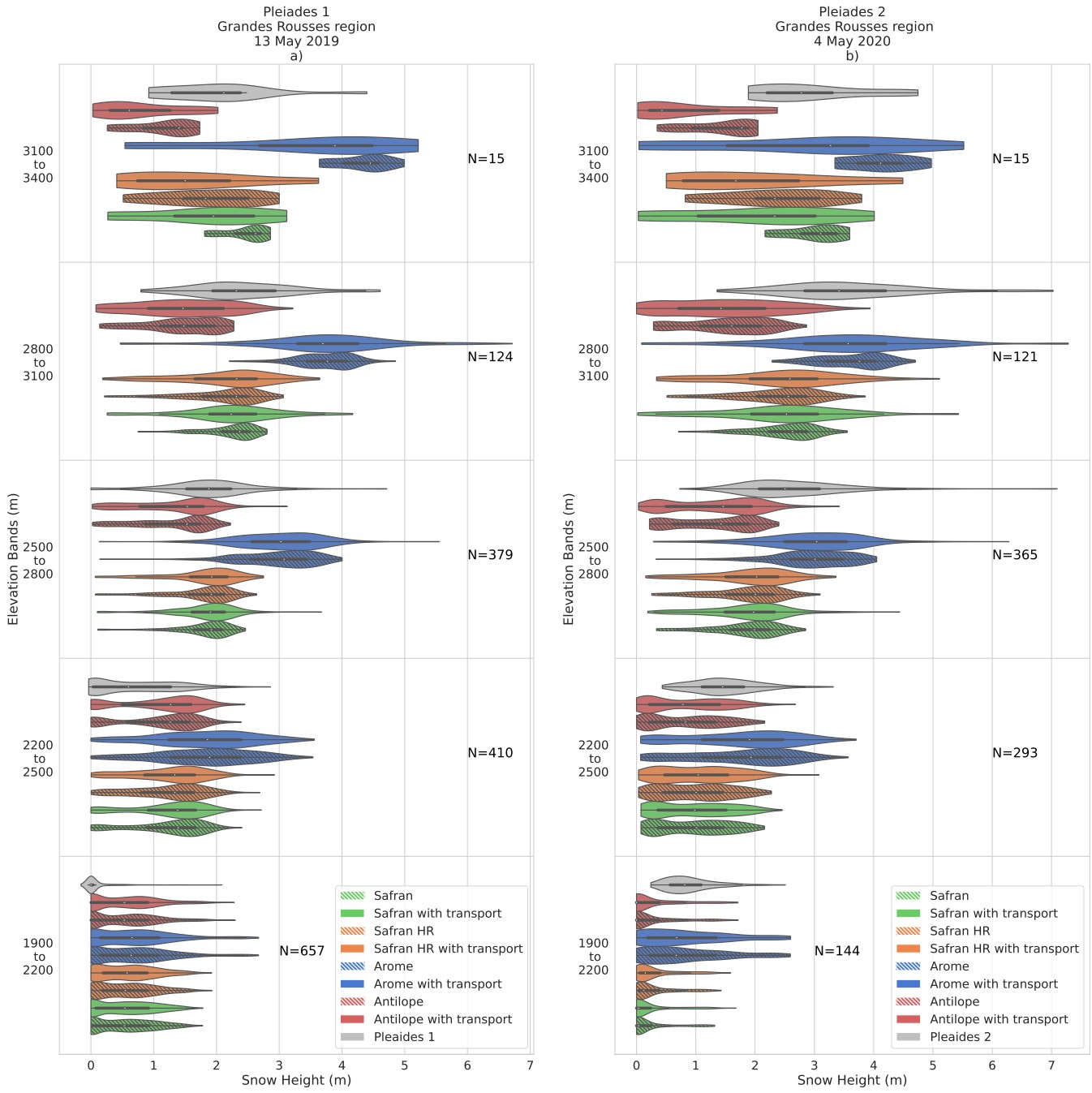

**Figure 5.** Spatial distribution of snow height (m) for 300 m elevation bands in Pléiades observations (gray) and all simulation configurations over the Grandes Rousses mountains on 13 May 2019 (Pléiades 1)(a) and 4 May 2020 (Pléiades 2)(b). The different colors represent the different precipitation forcing and horizontal resolutions while hatching is used to distinguish between the activation and deactivation of the SnowPappus blowing snow scheme. Each violin contains a small inner black box representing the first and third quartile, and a white dot representing the median distribution value. The number of samples for each subgroup is denoted by $N$.

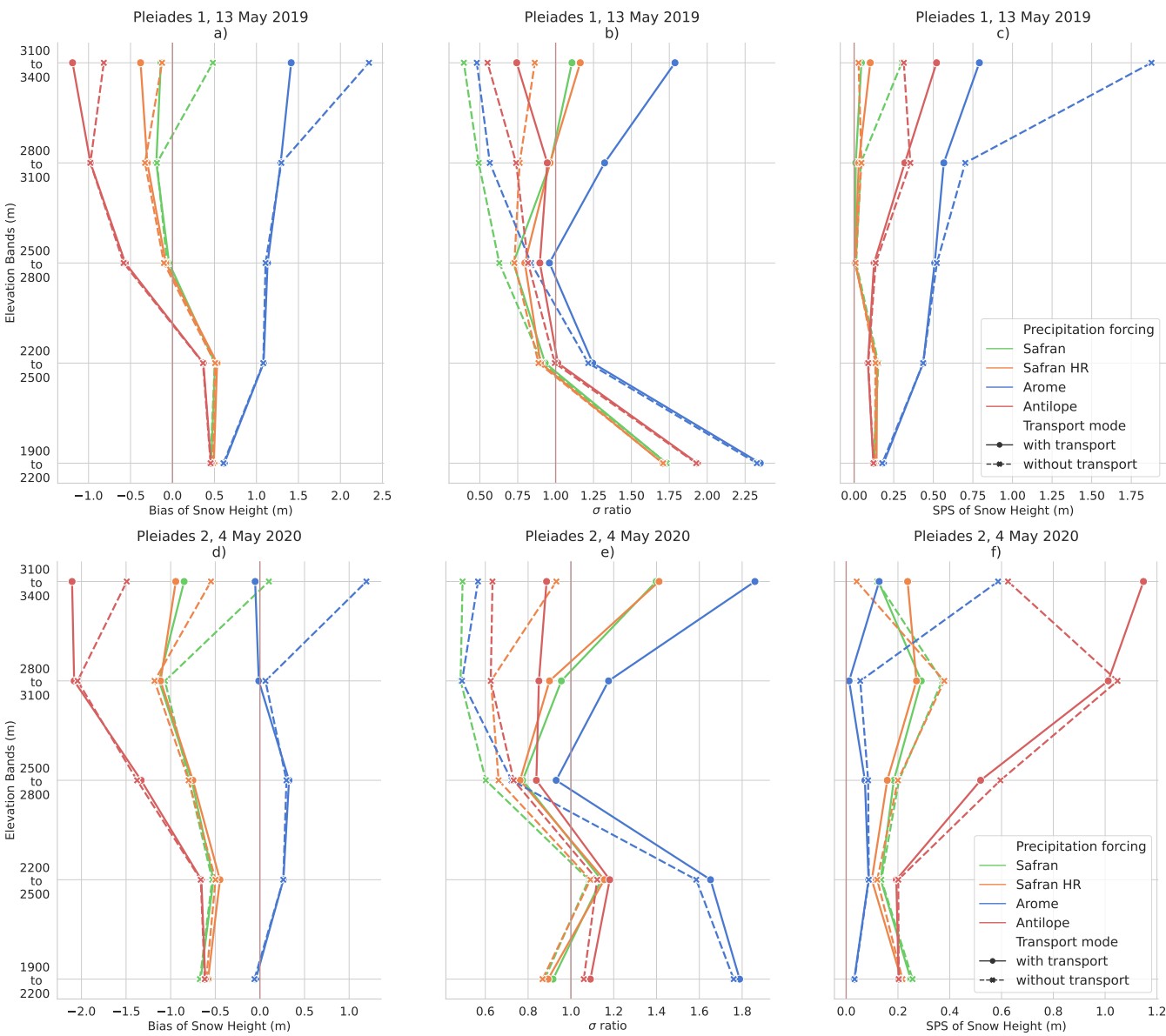

**Figure 6.** Scores comparing simulations with the Grandes Rousses Pléiades observations. (a-b-c) compares simulations with the 13 May 2019 Pléiades 1 image ; (d-e-f) is based on the 4 May 2020 Pléiades 2 image. (a-d) Bias (m) ; (b-e) Standard deviation ratio (no unit) ; (c-f) SPS (m). All scores are presented as a function of 300 m elevation intervals.

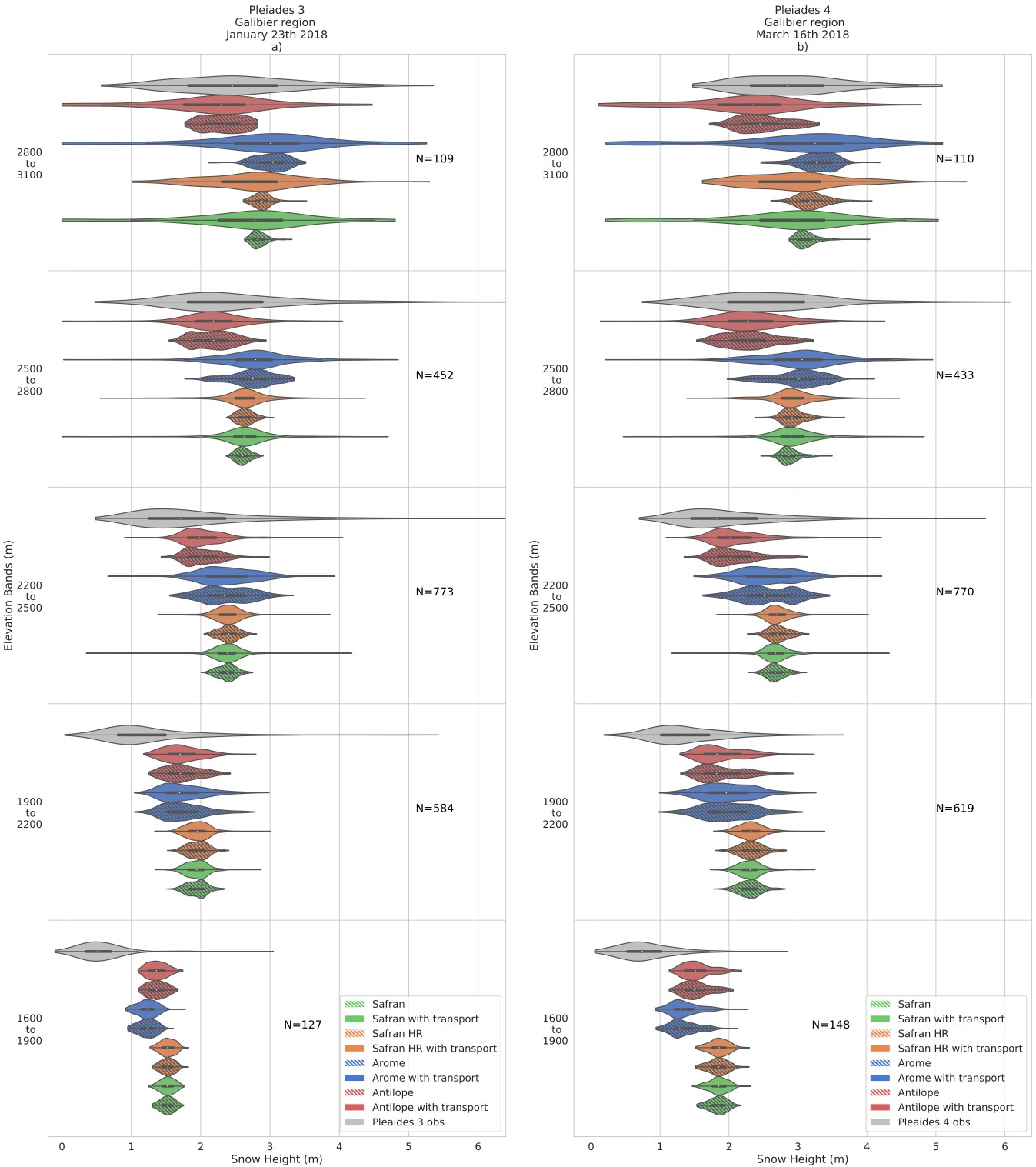

**Figure 7.** Spatial distribution of Snow height (m) for 300 m elevation bands in Pléiades observations (gray) and all simulation configurations over the Galibier on 23 January 2018 (Pléiades 3)(a) and 16 March 2018 (Pléiades 4)(b). Similar legend as in Fig. 5. Each violin contains a small inner black box representing the first and third quartile, and a white dot representing the median distribution value. The number of samples for each subgroup is denoted by $N$.

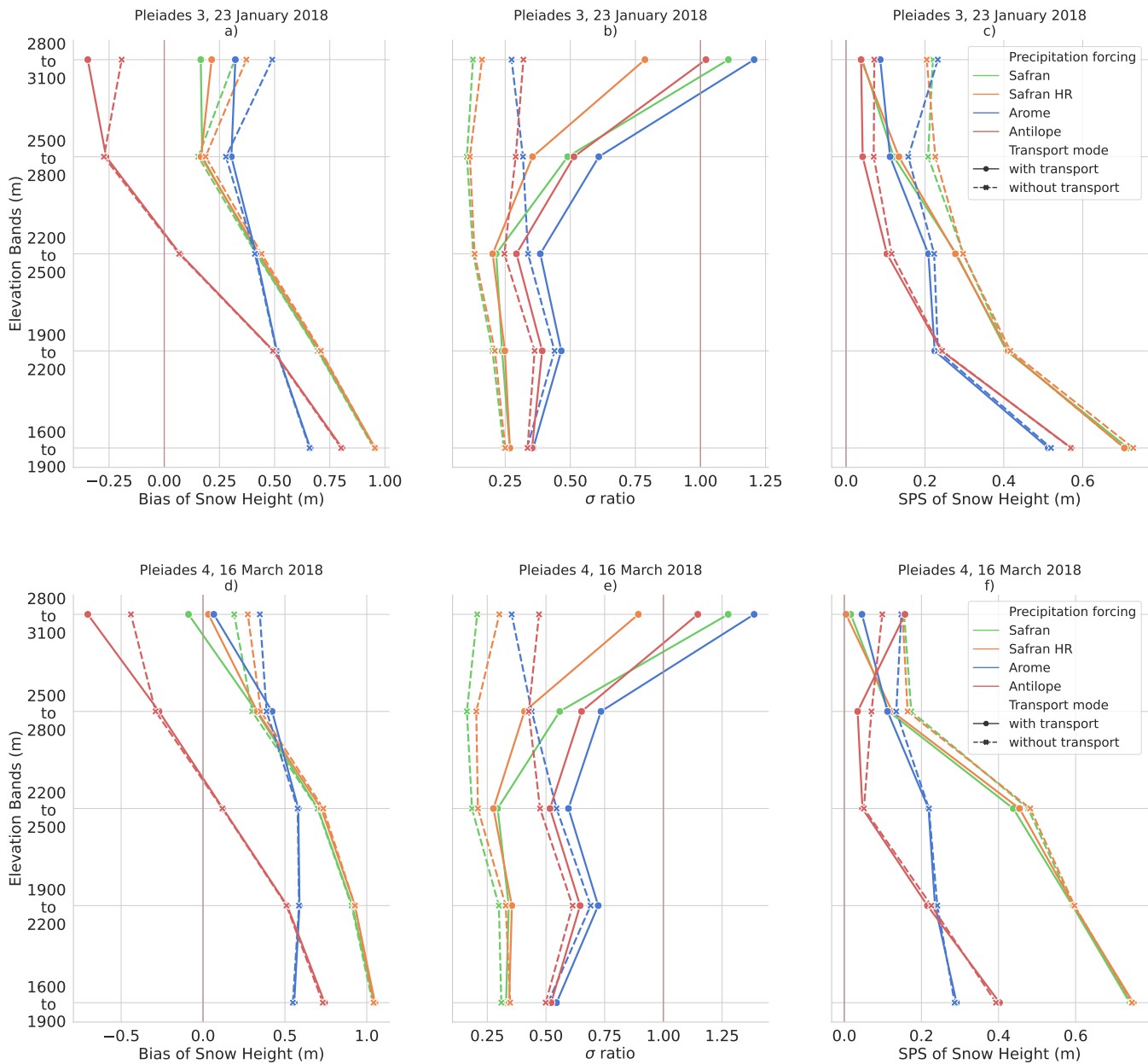

**Figure 8.** Scores comparing simulations with the Galibier Pléiades observations. (a-b-c) compares simulations with the 23 January 2018 Pléiades 3 image ; (d-e-f) is based on the 16 March 2018 Pléiades 4 image. (a-d) Bias (m) ; (b-e) Standard deviation ratio (no unit) ; (c-f) SPS (m). All scores are presented as a function of 300 m elevation intervals.

## 3.2 Spatial variability of Snow Melt-Out Date (SMOD)

The evaluations of the SMOD allow us to extend the evaluations over our entire simulation domain. It also provides complementary information on the entire snow season than the use of snow height snapshots. The associated spatial SMOD distributions for the 2017-2018 and 2018-2019 years can be found Fig. 9 (a-b). The synthetic summary of simulation performances is given by Fig. 10 (a-b-c-d-e-f).

### 3.2.1 Differences between years

There are some notable differences between the 2017-2018 and 2018-2019 snow season SMOD (Fig. 9). For the two lower elevation bands, there are on average 15 days (Fig. 9 a) and 20 days (Fig. 9 b) SMOD difference between the two years for the observation. A similar SMOD difference is found on average for the simulations. However, for the 2018-2019 snow season (Fig. 9 b), we see the observations and simulations distribution have a much bigger variance than the previous season. Continuing the analysis, the observed median SMOD show an earlier melt at all elevation for the 2018-2019 season. This observation is even more pronounced for the 3 upper elevation bands. We also note that the distribution variability is greater for this year. For the 2017-2018 snow season at elevations between 2500 and 3400 m, there was an average SMOD difference of 18 days between observations and simulations. In the subsequent season, this difference increased to an average of 39 days. In contrast, during the 2018-2019 snow season, the simulations that included snow transport were, on average, closer to the observed variance than for the other seasons.

### 3.2.2 Impact of precipitation forcing

Above 2500 m, the simulation with transport and the ANTILOPE precipitation has the largest distribution variance and lower mean SMOD value. As a result, this is the only simulation that underestimates the mean SMOD in 2018-2019 (Fig. 10 (d)) and overestimates its spatial variability (Fig. 10 (e)).

For the other precipitation forcing, in 2017-2018 the SMOD is overestimated at low elevations indicating that the snow melt-out occurred too late in the season compared to observation (Fig. 9 b). Conversely, SMOD is underestimated at high elevations with a simulated melting up to 20 days before the observed one. In 2018-2019, SMOD was overestimated at all elevations. On average, the spatial variance of SMOD is underestimated except for simulations with the ANTILOPE forcing at the upper elevation where the $\sigma$ ratio is found much above 1.

The SPS values of SMOD (Fig. 10 (c-f)) finally provided the opposite conclusion between both years with the simulations forced by AROME being the closest to the observed spatial distribution at all elevations in 2017-2018 but the furthest in 2018-2019 and conversely the simulations forced by ANTILOPE the furthest to observations in 2017-2018 and the closest in 2018-2019.

Analysis of wind speed and direction over the three simulation years indicates that the values remained consistent. Examination of the yearly solid precipitation of the SAFRAN forcing reveals that the 2018-2019 season had 21% less snowfall

compared to 2017-2018. This result is consistent with a smaller amount of solid precipitation leading to a reduced amount of snow accumulation and the observed earlier SMOD for the 2018-2019 snow season.

### 3.2.3 Impact of snow transport

Figure 9 and Fig. 10 (b) and (e) show an increase in SMOD standard deviation when activating snow transport in simulations,
consistent with the increase of snow height variance shown in Sect. 3.1. The standard deviation of simulated SMOD with transport is found to increase with altitude. This increasing spatial variance of SMOD with altitude is also observed with Sentinel 2 imagery, whereas simulation without transport keeps a standard deviation rather similar. Nevertheless, the impact of precipitation forcing on SMOD spatial variability is at least as high as the impact of snow transport. Therefore, the added value of snow transport is still difficult to establish on that criteria, although simulations based on SAFRAN and AROME forcing
would suggest that the SMOD spatial variability is improved by the snow transport module.

The addition of snow transport also decreases the SMOD for all 250 m resolution simulations, with an increasing impact on altitude. Therefore, concerning the SMOD bias, the impact of snow transport can be either beneficial in 2018-2019 or detrimental in 2017-2018 depending on the sign of the general bias of the simulation without transport.

Except with ANTILOPE in 2017-2018, simulating snow transport always results in improved SPS of SMOD. The detrimen-
tal impact of snow transport at high elevations in 2017-2018 for ANTILOPE forcing is probably linked with the much smaller estimates of precipitation by ANTILOPE (Appendix Fig. B1), for which the reasons are discussed in Sect. 4.2.

### 3.2.4 Impact of subgrid variability

The addition of snow transport in the 30 m resolution SAFRAN HR simulations increases the SMOD for the two evaluation years while on the contrary snow transport decreases the SMOD in the 250 m resolution SAFRAN simulation. In terms of
spatial variance, the increase of spatial variance by snow transport is higher at 250 m resolution than at 30 m resolution. The direct impact of resolving a finer topography in simulations without transport (differences between orange and green dashed lines) is less significant than on snow height evaluations.

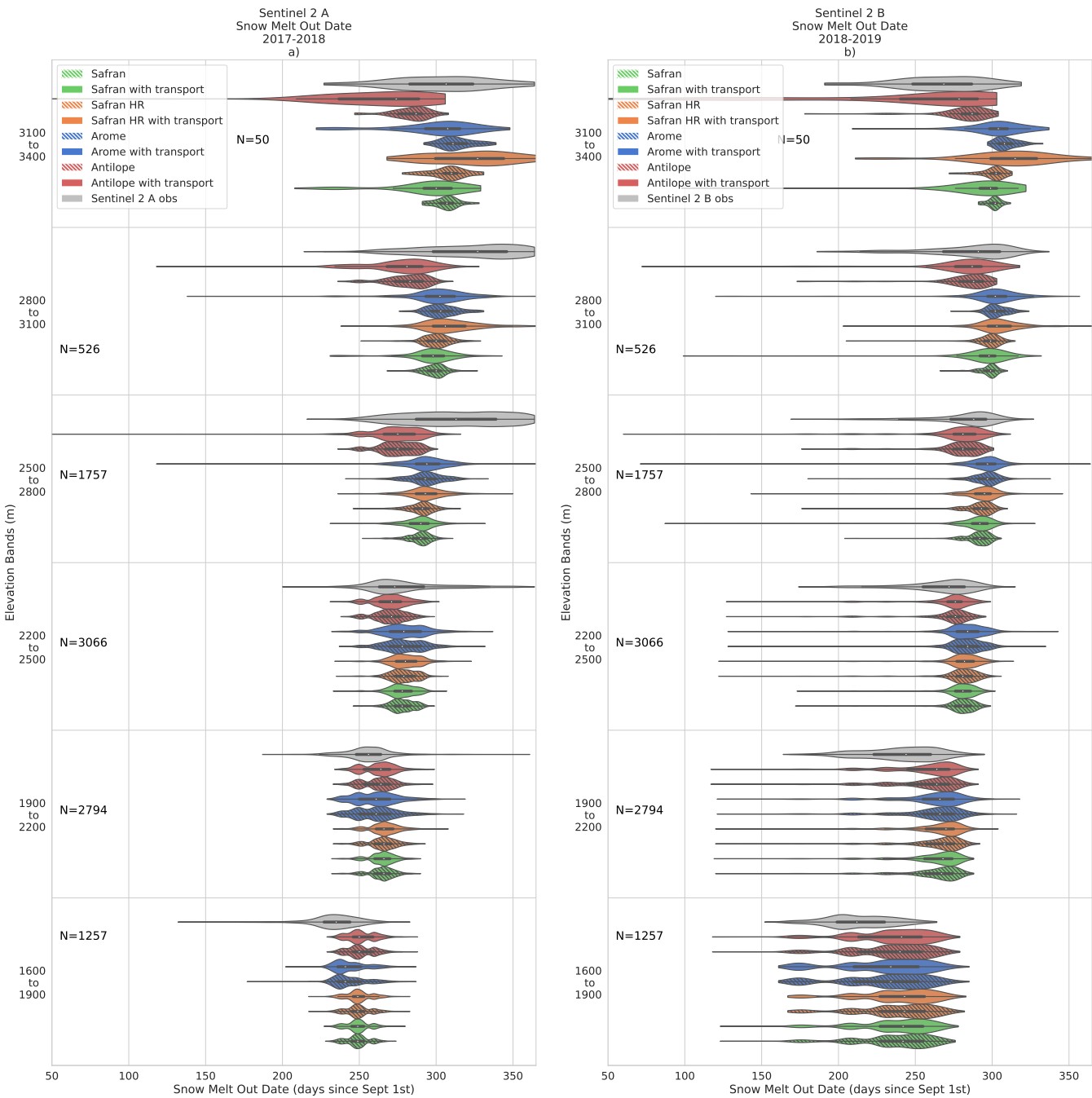

**Figure 9.** Snowpack Melt-Out Date (SMOD) depending on elevation and simulation configuration. The simulations are compared with SMOD observations of the full simulation domain, for the 2017-2018 (Sentinel 2 A) and 2018-2019 years (Sentinel 2 B). Each violin contains a small inner black box representing the first and third quartile, and a white dot representing the median distribution value. The number of samples for each subgroup is denoted by $N$.

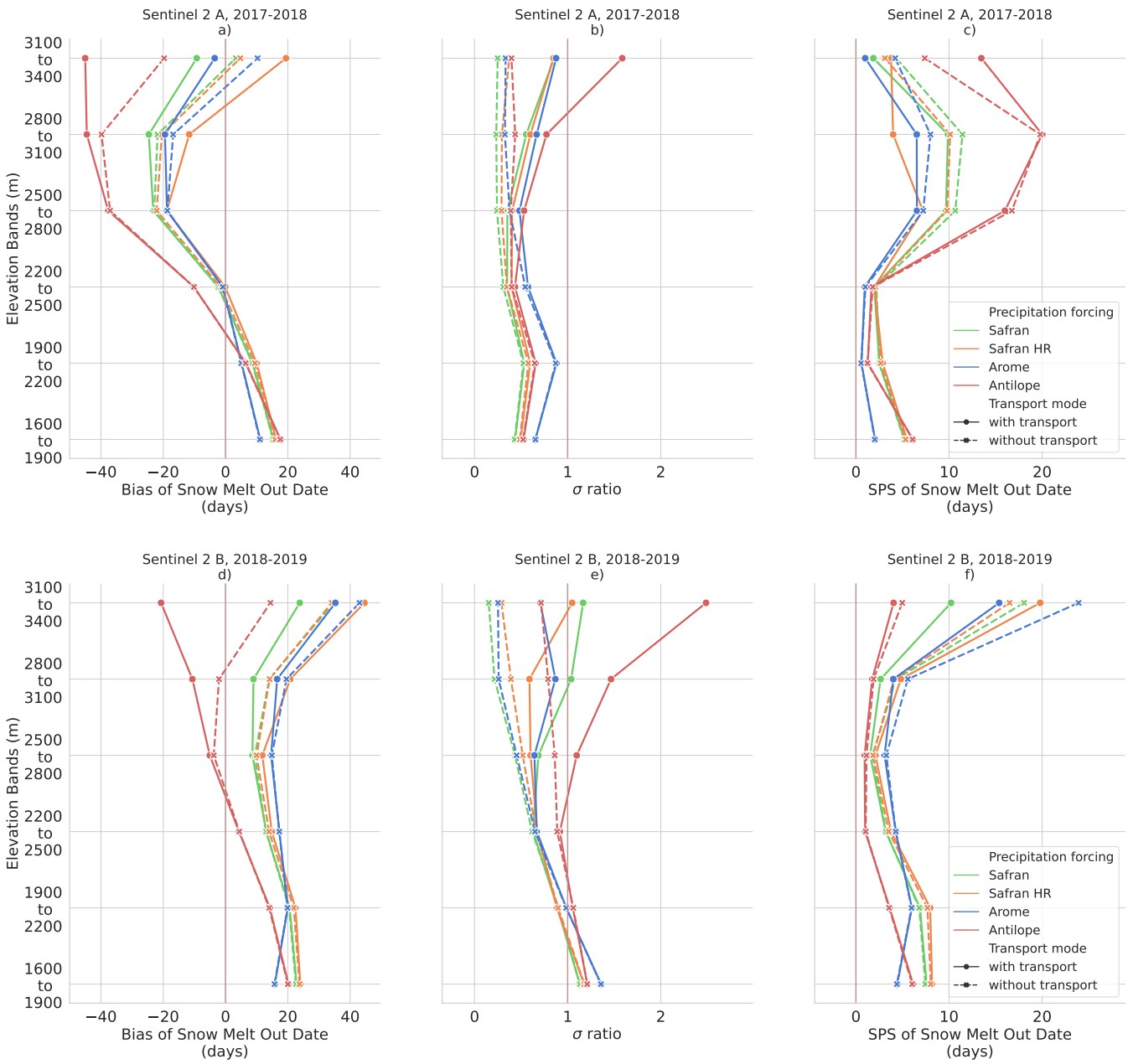

**Figure 10.** Scores comparing simulated SMOD with Sentinel 2 imagery. (a-b-c) 2017-2018 year (Sentinel 2 A) ; (d-e-f) 2018-2019 year (Sentinel 2 B). (a-d) Bias (days) ; (b-e) Standard deviation ratio (no unit) ; (c-f) SPS (days). All scores are presented as a function of 300 m elevation intervals.

## 3.3 Impact of landform types

In this section, the spatial simulated distributions of snow heights and SMOD are compared with satellite observations while
grouped by landform types instead of elevation bands, as described in Sect. 2.2 and Fig. 2.

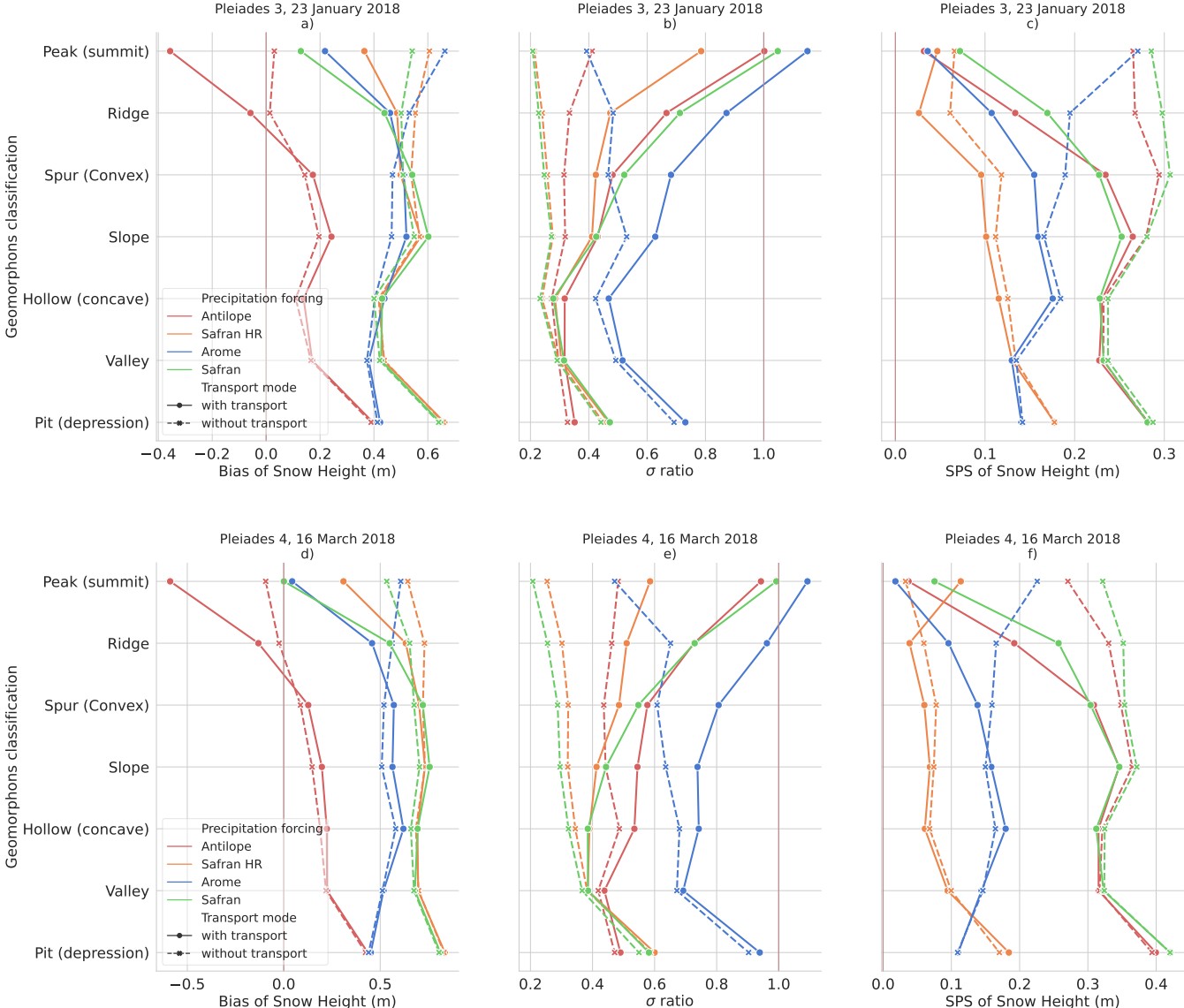

**Figure 11.** Scores comparing simulations with the Galibier Pléiades observations. (a-b-c) compares simulations with the 23 January 2018 Pléiades 3 image ; (d-e-f) is based on the 16 March 2018 Pléiades 4 image. (a-d) Bias (m) ; (b-e) Standard deviation ratio (no unit) ; (c-f) SPS. All scores are presented as a function of landform pixel classifications.

Figure 11 compares the January and March 2018 Galibier Pléiades images to snow height simulations grouped by landform types. Appendix Fig. D1 presents the same comparison using 13 May 2019 and 4 May 2020 Grandes Rousses Pléiades images. The comparison of simulation with and without transport shows that most of the snow height bias difference (Fig. 11 (a) and (d)) is found on pixels classified as 'Peak (summit)' or 'Ridges'. However, this impact on bias can be either beneficial or detrimental depending on the bias of the simulation without transport, highly dependent on the precipitation forcing, similar to the results described in the analyses based on elevation bands.

Simulations with snow transport show lower mean snow height in pixels classified as 'Peak (summit)' and 'Ridges' compared to simulations without transport. Conversely, simulations without snow transport exhibit lower snow height on 'Spur (Convex)', 'Slopes', and 'Hollow (Concave)'. This behavior can be found in each 250 m simulation, regardless of the simulation forcing (Fig. 11, Fig. D1) except for the SAFRAN HR simulations.

It can be noticed that the ordering between the different precipitation inputs does not depend on the landforms classes while it was quite dependent on elevation bands.

For all precipitation datasets, the standard deviation of simulations without transport is much smaller than in the observed spatial distribution. The landforms where the activation of transport has the most impact on snow height variance are (by order of importance): 'Peak (summit)', 'Ridges', 'Spur (Convex)', 'Slopes' and 'Hollow (Concave)'. Compared to the observed variance, this behavior is more realistic when snow transport is activated, Fig. 11 (b) and (e) for all precipitation forcings. While the simulated distribution variance better matches the order of magnitude of the observed one for 'Peak (summit)', 'Ridges', and 'Spur (Convex)', the simulated distribution variance of 'Slopes' and 'Hollow (Concave)' is seen to be still half of the observed one. As for the evaluation using elevation bands, the simulations with the AROME precipitation have the highest spatial variability for the two dates and each landform type. Conversely, the simulations with SAFRAN and no snow transport have the smallest standard deviation across landforms.

The improvement of the SPS score Fig. 11 (c) and (f) due to the representation of snow transport is major on 'Peaks (summits)' and 'Ridges' and noticeable on 'Spur (convex)' and 'Slope' classes. This improvement is obtained for all precipitation forcings (even in the ANTILOPE case where the bias has deteriorated), therefore mainly driven by the improvement in snow height variance. The SAFRAN HR simulations with and without transport have the lowest (better) SPS values for the two evaluation dates and most of the landform types while it is not the least biased. Directly examining the SAFRAN HR cumulative distribution function reveals that the shape of the cumulative distribution is more similar to the observation, despite being biased. This conclusion differs from the results obtained using a grid cell classification by elevation bands.

Figure 12 presents the two-year evaluation using Sentinel 2 SMOD and landforms. Figure 12 (a-d) show the impact of SnowPappus blowing snow simulations on simulated SMOD. As for snow height in Fig. 11, the impact on SMOD is most significant in areas classified as 'Peak (summit)' or 'Ridge'. SMOD difference when using SnowPappus is found to go up to 20 days for 'Peak (summit)'. When comparing the two evaluation years, similar behavior is found, except the 2018-2019 year appears uniformly more biased. Areas classified as 'Valley' and 'Pit' appear more biased for every precipitation forcing.

In Fig. 12 (b-e), the impact of snow transport is again mainly present on 'Peak (summit)' and 'Ridges' landforms, but with high variability between both years and a lower impact when AROME precipitation forcing is used (with already a higher

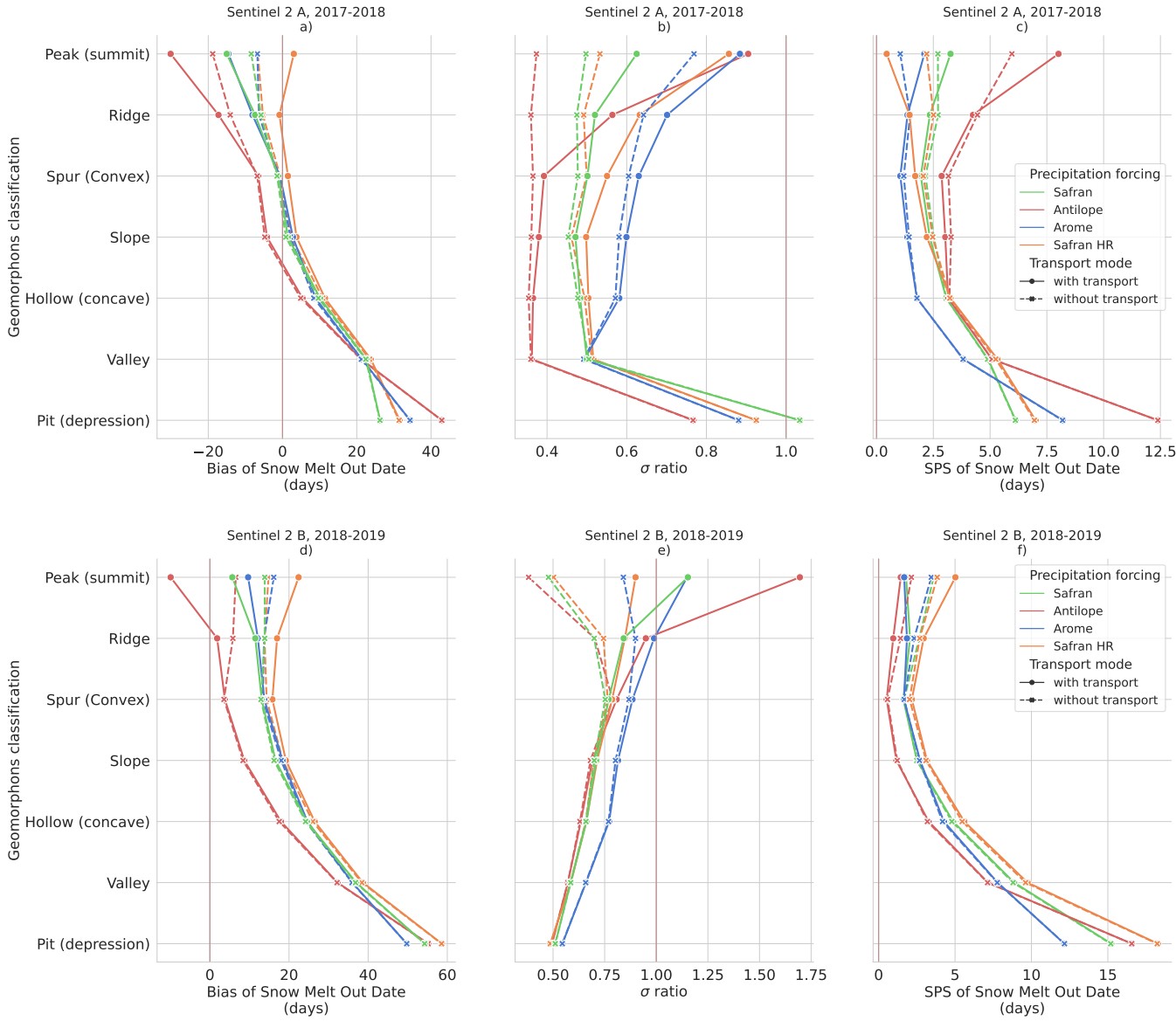

**Figure 12.** Scores comparing simulated SMOD with Sentinel 2 imagery. (a-b-c) 2017-2018 year (Sentinel 2 A) ; (d-e-f) 2018-2019 year (Sentinel 2 B). (a-d) Bias (days) ; (b-e) Standard deviation ratio (no unit) ; (c-f) SPS (days). All scores are presented as a function of landform pixel classification.

spatial variability even without transport). In 2017-2018, the impact of snow transport is always beneficial on the SMOD $\sigma$ ratio, as obtained for snow heights, but for the second season, it depends on the precipitation forcing. The simulated standard deviation of AROME simulations appears close to the observed one for all landforms but pixels classified as 'Pit'. This better

behavior of AROME was not illustrated in the evaluation done using the elevation bands. The other precipitation forcing
appears underdispersed with a standard deviation ratio of 0.5-0.75.

Looking at the SPS scores Fig. 12 (c-f), the impact of snow transport is less clear than for snow heights, similar to the results
obtained by elevation bands. SPS values are smaller using landform classification than elevation bands. All simulations appear
to have difficulties in representing SMOD distribution in pixels classified as 'Valley' and 'Pit' conversely to results of Fig.
11. The SPS scores are also more similar between the two evaluation years, contrary to the SPS score using elevation bands
grouping. Figure 10 (c) shows a distance increase between all simulations and observations distribution between 2500-3100 m.
This effect is not found in Fig. 12 (c). On the contrary to Fig. 11, the SAFRAN HR simulations have the overall highest SPS
values, and AROME simulations' $\sigma$ ratios are found to be more realistic.

The impact of the group axis on results can be quantified by comparing identical data grouped differently. Fig. 13 presents
the mean SPS values of the 8 simulations when grouped by elevation bands and landform types. We can see that SPS values
are lower when grouped by landform types. We can see that grouping by elevation gives more homogeneous SPS scores for
the different observation dates than grouping by landform where variation between dates is greater.

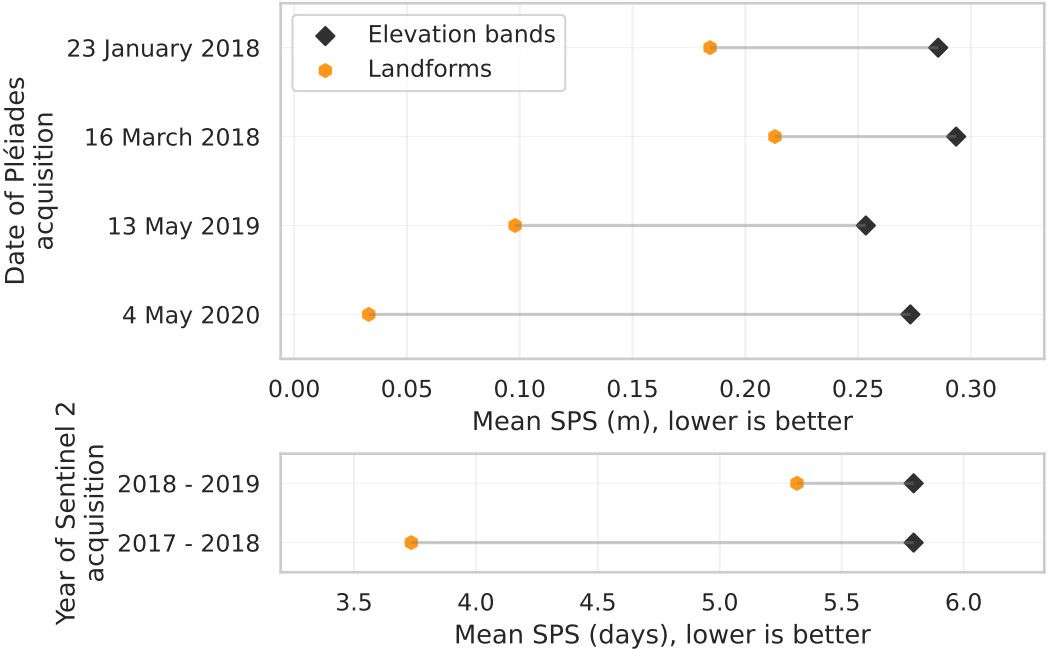

**Figure 13.** Plots of average SPS values for the different simulations when grouped with elevation bands or landform types. This graph
illustrates the impact of the grouping axis (elevation or landform types) on snow height and snow melt-out date.

### 3.4 Impact of the SnowPappus model

Correlation between observations and the different simulations can be seen in Fig. 14. This figure illustrates scatter plots of snow height for the four different Pléiades observations to the 8 different simulations (SAFRAN, SAFRAN HR, AROME, ANTILOPE, with and without activation of SnowPappus). For each simulation, we see more dispersed scatter plots for simulations with snow transport (greater variance). It can be noted that the mean difference in slope caused by the addition of snow transport is small (<0.02). Overall, AROME simulations have the steeper slopes. To quantify how correlated the simulation results are to the observations, we use the Pearson correlation coefficient.

In Table 3 and Table 4, we see Pearson correlation coefficients between Pléiades observations and simulations for each of the 8 simulations. Correlation is computed two times, for the entire area (all elevations) and restricted to elevation higher than 2700 m.

Looking at the Pearson correlation for the entire Pléiades domain, the addition of snow transport in simulations decreases the correlation coefficient in 15 of 16 experiments. Restricting the correlation to pixels to elevation above 2700 m (pixels with high snow transport probability) the addition of snow transport in simulations increases the Pearson correlation for each experiment. This result is found statistically significant (with significance set at p-value <0.05) in 15 out of 16 experiments.

We note that for the entire domain, the variability of the correlation score for a single observation is greater according to the source of precipitation than for the addition of the snow transport process in the simulations.

To better disentangle the impact of precipitation-forcing variability and the addition of snow transport, this result can be supplemented with Fig. 15, representing the mean SPS value of the 6 different 250 m simulations forced with SAFRAN, ANTILOPE, AROME, for elevations above 2700 m and the entire observation domain. In this figure, the addition of snow transport to the simulations clearly shows improvements in the mean SPS score on snow height and SMOD above 2700 m for all observations. SnowPappus lowers SPS values of respectively 0.06 m and 0.85 days above 2700 m. The improvement in SPS score by adding snow transport is logically much smaller if we look at the total snow height and SMOD distributions without restricting the elevation. However, it is important to note that the addition of the SnowPappus model does not deteriorate the SPS scores.

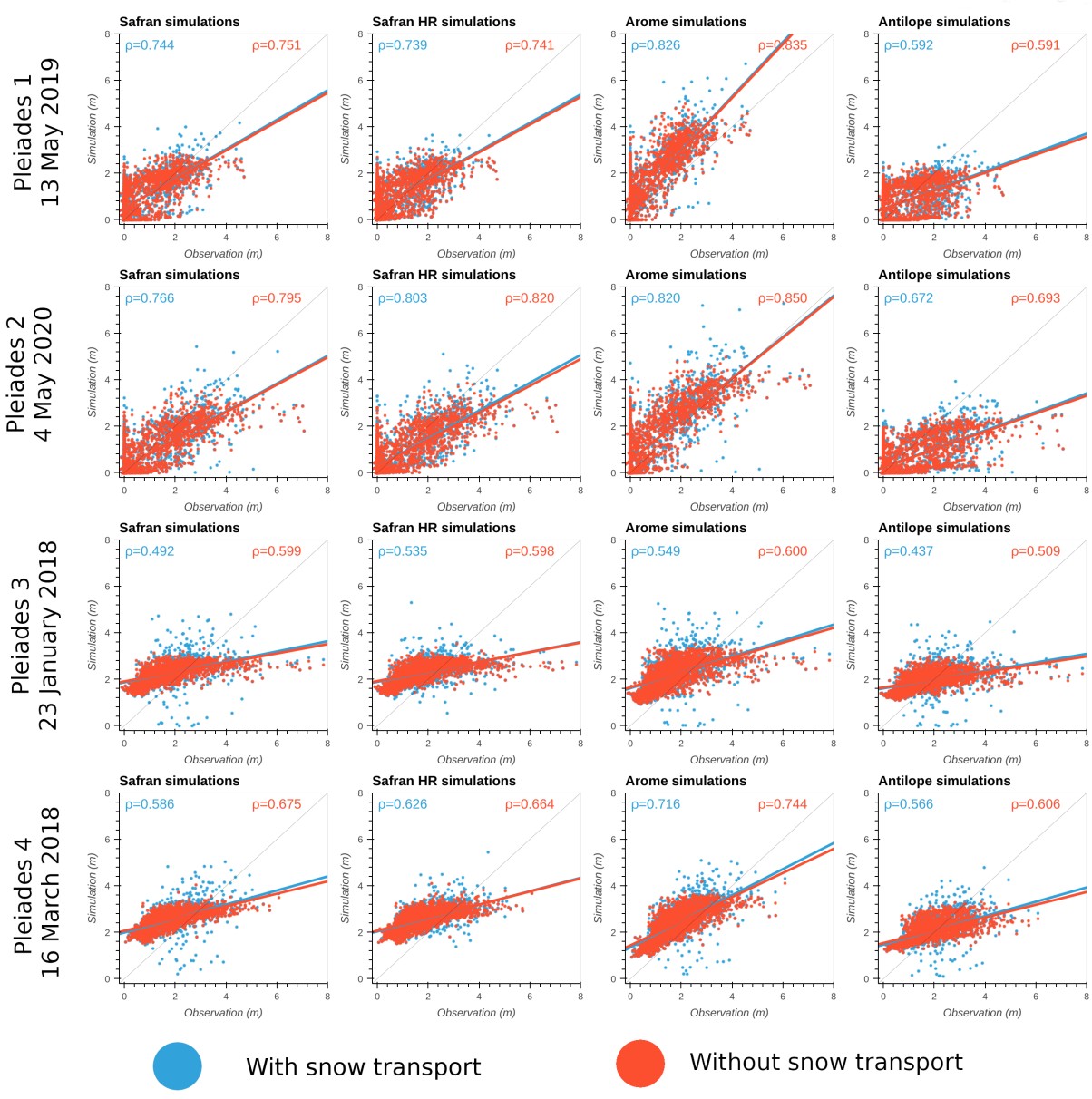

**Figure 14.** Observed vs simulated snow height scatter plots for the four different Pléiades observations (all elevations) and the different precipitation forcings. For each precipitation forcing, the simulation with snow transport is in blue and without transport in red. Linear regression line synthesizes distribution changes when adding snow transport. Additionally, the respective Pearson correlation coefficient ($\rho$) are displayed.

| Forcing | Snow transport | Pléiades 1 2019-05-13 | Pléiades 1 2019-05-13 Z>2700 m | Pléiades 2 2020-05-20 | Pléiades 2 2020-05-20 Z>2700 m |
|---|---|---|---|---|---|
| Safran | Yes | 0.744 | **0.173** | 0.766 | **0.184** |
| Safran | No | **0.751** | 0.027 | **0.795** | 0.062 |
| Safran HR | Yes | 0.739 | **0.190** | 0.803 | **0.221** |
| Safran HR | No | **0.741** | 0.097 | **0.820** | 0.173 |
| Arome | Yes | 0.826 | **0.169** | 0.820 | **0.211** |
| Arome | No | **0.835** | 0.064 | **0.850** | 0.127 |
| Antilope | Yes | **0.592** | **0.174** | 0.672 | **0.198** |
| Antilope | No | 0.591 | 0.045 | **0.693** | 0.105 |

**Table 3.** Pearson correlation coefficients between observed and simulated snow height (Grandes Rousses). For each observation/simulation pair, a second correlation is computed only with pixel elevations (Z) above 2700 m. Underlined values represent p-value >0.05. **Bold** values correspond to the best correlation value between simulation with and without the SnowPappus model.

| Forcing | Snow transport | Pléiades 3 2018-01-23 | Pléiades 3 2018-01-23 Z>2700m | Pléiades 4 2018-03-16 | Pléiades 4 2018-03-16 Z>2700 |
|---|---|---|---|---|---|
| Safran | Yes | 0.492 | **0.192** | 0.586 | **0.346** |
| Safran | No | **0.599** | 0.068 | **0.675** | 0.142 |
| Safran HR | Yes | 0.535 | 0.110 | 0.626 | **0.353** |
| Safran HR | No | **0.598** | 0.086 | **0.664** | 0.157 |
| Arome | Yes | 0.549 | **0.189** | 0.716 | **0.414** |
| Arome | No | **0.600** | 0.025 | **0.744** | 0.355 |
| Antilope | Yes | 0.437 | **0.181** | 0.566 | **0.422** |
| Antilope | No | **0.509** | 0.020 | **0.606** | 0.266 |

**Table 4.** Pearson correlation coefficients between observed and simulated snow height (Galibier). For each observation/simulation pair, a second correlation is computed only with pixel elevations (Z) above 2700m. Underlined values represent p-value >0.05. **Bold** values correspond to the best correlation value between simulation with and without the SnowPappus model.

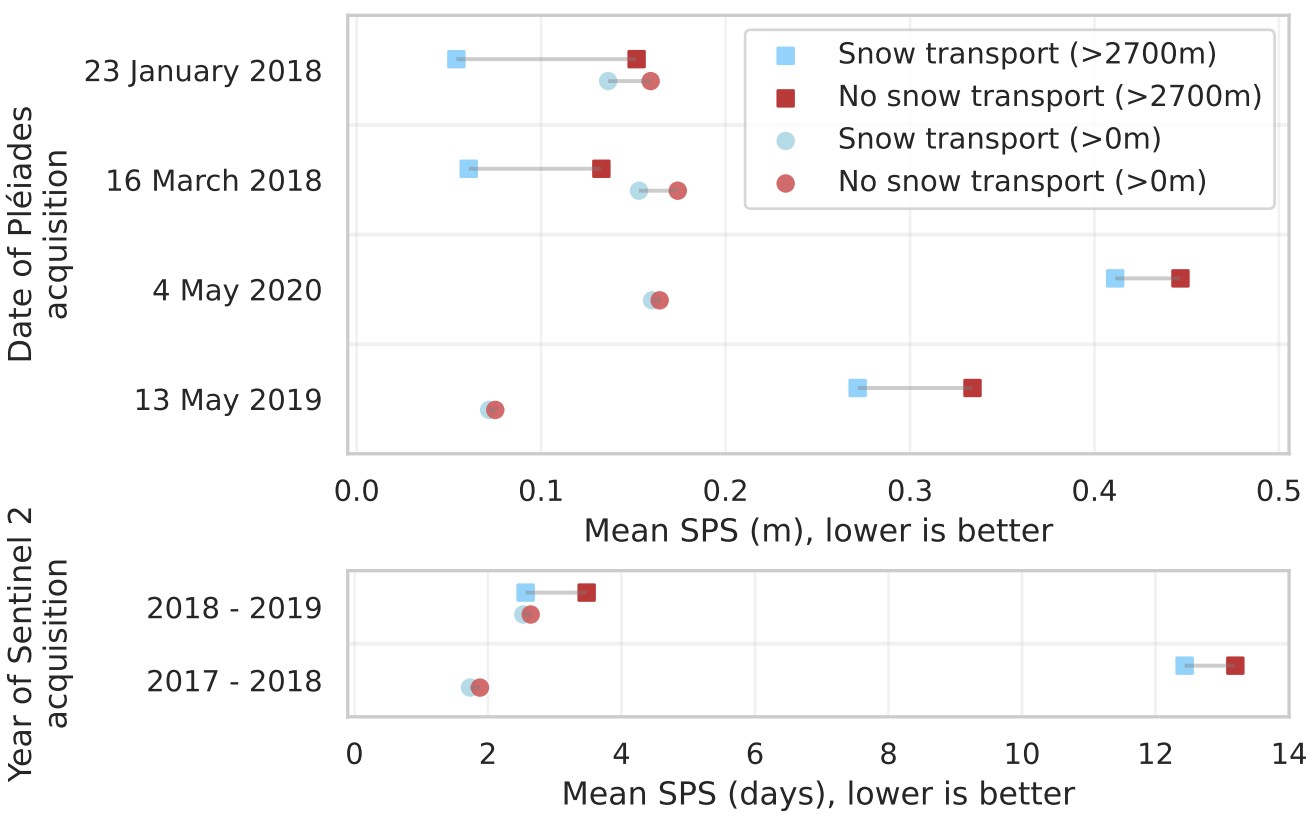

**Figure 15.** Plots of average SPS values for the three different 250 m forcings (SAFRAN, ANTILOPE, AROME) compared to observations. SPS values are computed for pixels with elevations above 2700 m (square) and pixels of the entire observation domain (all elevations, circle). This graph summarizes the impact of the SnowPappus transport model on snow height and snow melt-out date distributions.

## 4 Discussion

### 4.1 The Added value of the SnowPappus model

In this study, we compare snow height maps simulated with and without transport using various sources for the precipitation forcing. The combined impact of precipitation forcing variability and wind-blown redistribution using the SnowPappus model is visible, on snow height, in Fig. 5 and Fig. 7 and SMOD in Fig. 9. For the higher altitudes, particularly around the summit pixels, the standard deviation of the simulated snow height and snow melt-out date distribution is primarily determined by blowing snow transport. The blowing snow transport impact is mostly restricted to 'Peak (summit)' and 'Ridges' landforms (Fig. 11, Fig. D1, Fig. 12) and more generally to the upper elevation bands (above 2700-2800 m). In addition, the snow height variance is seen to increase closer to the observation using SnowPappus' simulated snow transport, regardless of the precipitation bias. Looking at the Pléiades bias values in Fig. 6 and Fig. 8 (a) and (d), blowing snow lowers in value the mean bias of all simulations with snow transport compared to simulations without snow transport. This is probably a combined effect of increased snow density and mass loss from increased sublimation. This result is consistent with Vionnet et al. (2021). In Sect. 3.4, results show the SnowPappus model consistently improves the correlation between Pléiades observations and simulations above 2700 m. This improvement was found to be significant in 15 out of 16 experiments. In our simulation domain, the elevation of 2700 m and above correspond to areas where our snow transport model has a significant impact on the simulated snow height. For the Pléiades domain as a whole, the correlation coefficient is modestly reduced with the addition of snow transport. These results are a generalization of the correlation analysis done in Baron et al. (2024) and are consistent with the previous conclusions. However, the added value of SnowPappus can not be established only on this criteria as we find the simulated bias and standard deviation dependent on the precipitation forcing. To better disentangle the effects of precipitation forcing variability and the addition of snow transport, the previous conclusions can be supplemented with Fig. 15, which shows the mean SPS value of the 6 different 250 m simulations forced with SAFRAN, ANTILOPE, AROME. Based on these results, it can be concluded that the addition of the SnowPappus model to our 250 m simulation improves the simulated snow distribution closer to the observed distribution for both snow height and SMOD.

In the literature, Bernhardt et al. (2012) studied the influence of lateral snow redistribution using SnowTran-3D (Liston et al., 2007) and found an enhancement in the simulated snow spatial pattern using lateral snow redistribution. Using the Canadian Hydrological Model (Marsh et al., 2020), Vionnet et al. (2021) find snowpack simulation without lateral snow redistribution unable to capture the spatial variability of snow cover in alpine terrain. On the opposite, the addition of lateral snow redistribution is found to provide a better estimate of snow height across elevation and increase the simulated snow height distribution variance. Very recently, Quéno et al. (2023) evaluated FSM2trans snow redistribution model (Quéno et al., 2023) and find the addition of snow redistribution necessary at hectometric or finer resolution to better represent snowpack heterogenity. Snow heights simulated using lateral snow redistribution are found to have significantly better variability. Additionally, (Quéno et al., 2023) conducted a model sensibility analysis to horizontal resolution and established that although the most realistic simulation pattern is found at 25 m resolution, the addition of lateral snow redistribution at coarser resolution is also having a positive im-

pact on snow distribution and variance. Hence, we estimate the results obtained in this study on the impact of the SnowPappus

snow transport model compatible with the literature.

## 4.2 Sensitivity to precipitation forcings

In our simulations, the mean and median snow height and SMOD values are primarily driven by the spatial variability of precipitation forcing, as shown in Fig. 5, Fig. 7, and Fig. 9. Precipitation forcing is also found to be the main driver of simulated snow height distribution standard deviation across low to medium elevations. As the Crocus snow model is known to produce

realistic snow heights and snow water equivalent when forced by well-controlled meteorological forcing (Lafaysse et al., 2017; Menard et al., 2021), the large biases of snow height and SMOD observed in the snow simulations presented most probably reflect intrinsic biases of the different precipitation dataset. In Fig. 6 and Fig. 8 (a) and (d), for a simulation with a precipitation dataset leading to an already negatively biased snow height (like ANTILOPE), the addition of snow transport further degrades the overall snow height simulation bias. Similar behavior is found on the SMOD in Fig. 10 (a) and (d) with an exception for

the 30m simulation where it increases the absolute SMOD bias. Thus the beneficial or detrimental impact of blowing snow transport on mean snow height or SMOD bias is mainly dependent on the precipitation bias. This result is consistent with Raleigh et al. (2015); Schlögl et al. (2016); Günther et al. (2019), also finding the prevalence of meteorological forcing errors in snow simulations. More specifically, when looking at the simulation forcing with the ANTILOPE precipitation source, we found that across all elevation bands, the ANTILOPE forcings have the lowest snow height and SMOD values. A negative

bias trend is seen with elevation (Fig. B1). The ANTILOPE precipitation product is primarily based on radar reflectivity measurements. technique known to be more challenging in mountainous terrains (Yu et al., 2018; Faure et al., 2019; Foresti et al., 2018; Germann et al., 2022). Meteorological radars are based on the backscatter measurement of several conic-shaped microwave electromagnetic impulses produced by the radar itself at several elevation angles (conic-shaped radar beams). When a mountain intercepts a radar beam, the reflected signal (called ground clutter) is unusable and rejected by the ANTILOPE

algorithm. If the mountain intercepts less than 70% of the beam section, the signal reflected further away is simply corrected to account for the loss of information but is still used for precipitation rate estimation. Precipitation estimation over pixels affected by ground clutters thus comes from beams that can be several thousands of meters higher than the mountain, often above the base of precipitation clouds. This leads to an underestimation of the precipitation rate over mountain ridges (Appendix Fig. B1). One possible compromise would consist of using the observed spatial variability from ANTILOPE while correcting the

elevation-dependent precipitation bias.

In contrast to ANTILOPE, our simulations with the AROME precipitation source lead, on average, to the highest snow height across the elevation bands. It is particularly visible in Fig. 4 (b), where squared patterns can be seen and come from the 1.3km AROME pixels forming distinctive precipitation patterns (see Appendix Fig. B1 d). This behavior of excessive solid precipitation at high elevations is known and coherent with other snow cover studies using AROME as the precipitation

source (Quéno et al., 2016; Vionnet et al., 2019; Monteiro et al., 2022). Multifactorial causes were discussed by Monteiro et al. (2022); Gouttevin et al. (2023). The variable biases between the western (Grandes Rousses) and eastern (Galibier) evaluation

areas suggest that orographic enhancement of precipitation might be especially overestimated by AROME on the areas the most exposed to the prevailing westerly flows.

Finally, the two SAFRAN simulations show a good agreement in terms of SMOD and snow height biases but are spatially under-dispersed by design. The SAFRAN meteorological reanalysis (Vernay et al., 2022) uses one vertical profile per climatologically homogeneous area, interpolated to all simulation points with altitude (see Sect. 2.5.1). Within one of these areas, the spatial variability is thus limited to a vertical gradient as described in Vionnet et al. (2016). This also explains why opposite snow height biases are obtained with this forcing between western (Grandes Rousses) and eastern (Galibier) areas.

### 4.3 Sensitivity to horizontal resolution

Numerical simulations rely on numerical discretization and parameterization. The choice of a spatial resolution inevitably introduces errors. This choice depends on a trade-off between scientific arguments, computing resources, and external factors. As described in Baron et al. (2024), the 250 m resolution choice of the SnowPappus scheme was motivated by the desire for applicability in large-extent systems (50000 to 100000 km$^2$, multi-decadal simulations). the numerical resolution choice impacts simulated slope inclinations and aspects and can lead to inaccuracy in the radiative balance (Baba et al., 2019). The snow radiative balance is particularly heterogeneous in mountainous areas due to complex topography and can lead to significant changes in snow properties and melting rates. Comparing simulations computed at 250 and 30 m resolution provides an analysis of the unresolved subgrid variability. A direct comparison between Appendix Fig. C1 (a-b-c) reveals the expected unresolved subgrid variability. Snow height patterns on the 250 m simulations (Appendix Fig. C1(a)) exhibit steeper variations than the 250 m re-gridded maps obtained from 30 m simulations (SAFRAN HR, Appendix Fig. C1 (b)) where the snow patterns are found smoother. To understand the origin of this subgrid variability, Appendix Fig. C1 (c) illustrates the raw 30 m simulation where snow patterns are more sophisticated. As expected (Baba et al., 2019), resolving the 30 m topography in simulation without snow transport increases the spatial variance at high elevations and reduces the 250 m averaged snow height (Sect. 3.1.4). Increasing the resolution in simulation with snow transport increases the SMOD (Sect. 3.2.4) but reduces the expected snow height variability gain compared to 250 m simulations. The complex behavior of resolution can be summarized by comparing the mean SPS value across all elevations of the SAFRAN and SAFRAN HR simulations, as in Appendix Fig. E1. This figure shows a very small mean SPS change with resolution. The mean difference is found to be lower than 0.01 m for the Pléiades observations and lower than a day (0.98) for SMOD. Until now, the resolution-dependence of snow transport models was only documented at higher resolutions (Grünewald et al., 2013; Marsh et al., 2020; Mott et al., 2010; Schneiderbauer and Prokop, 2011) but results recently submitted suggest that the resolution-sensitivity of SnowPappus in this range of spatial scales is similar to what is obtained from another snow transport model at similar resolutions (Quéno et al., 2023).

Quéno et al. (2016); Bellaire et al. (2014) and Schirmer and Jamieson (2015) also emphasize the relationship between the available resolution of the atmospheric forcing and the resulting snow simulation leading to the recommendation of forcing spatial resolution of 1 km or less, although evaluations of these data are very challenging at this scale. A much lower resolution is needed to capture the relationship between topography wind field and snow accumulation (Dadic et al., 2010) or for a complete representation of wind-driven processes (Mott et al., 2008, 2014; Bernhardt et al., 2010; Vionnet et al., 2017).

## 4.4 Impact of the grid cells classification in spatial evaluations

Classically in the snow modeling community, the quantitative spatial snow evaluations are performed by grouping simulation points (Grünewald et al., 2014; Vionnet et al., 2022; Monteiro et al., 2022; Tong et al., 2009; Deschamps-Berger et al., 2022; Vionnet et al., 2017) because (1) it allows summarizing large simulations dataset with more concise diagnostics and (2) these evaluations are less demanding for models on large areas than pixel-to-pixel evaluations. In Sect. 3.3, results are presented using landform classification instead of elevation bands. The results are partly correlated because summits and ridges mainly cover the highest elevations of the domain (Fig. 3). Thus, we consistently obtain that snow transport mainly affects snow height and SMOD on the upper elevation bands and for geomorphons 'Peaks (summit)' and 'Ridge'. However, some conclusions significantly differ between both space classifications, especially for the $\sigma$ ratio score and the SPS. A notable difference is found in the SPS snow height comparison. SPS scores are found to show lower variability for the different precipitation forcings and overall smaller SPS values using landforms grouping than elevation bands (Fig. 11, Appendix Fig. D1 (c-f)). This result can be supplemented with Fig. 13 showing the mean SPS value being consistently better using landform groups. This suggests simulated and observed distributions are closer when grouped using landforms than elevation bands. Indeed, for the landform grouping, each different landform distribution includes a relatively large spatial variability (intra-class variance) mostly due to elevation, slope, aspect, and precipitation variability (Clark et al., 2011; Freudiger et al., 2017). On the other hand, using elevation grouping the intra-class variance is mostly caused by other processes than elevation. This might suggest that our simulations better capture the altitudinal variability of snow than the other processes and the topographic-dependent variability. This result clearly illustrates the sensitivity to the choice of grid cell classification groups of the scores used.

The landform classification also provides complementary insights into the behavior of the SnowPappus model. In Fig. 11 and Appendix Fig. D1 (a-d), SnowPappus reduces the mean snow height on pixels classified as 'Peak (summit)' and 'Ridges' and increases the snow height on 'Spur (convex)', 'Slope' and 'Hollow (concave)' comparatively to simulations without snow transport. This behavior is the result of the wind spatial patterns obtained from Le Toumelin et al. (2023). It can be understood as snow on 'Peak (summit)' and 'Ridges' (ablation areas) being transported to pixels classified as 'Spur (convex)', 'Slope' or 'Hollow (concave)' (deposition areas) due to respective wind acceleration and deceleration on these pixels. This behaviour is similar between all 250 m resolution simulations while different patterns are obtained from the 30m SAFRAN HR simulation. Indeed, the finer spatial scales resolved at 30-m resolution produce more complex spatial patterns of wind fields and snow transport. When averaged at 250-m resolution, these simulations produce a more subtle influence of snow transport than simulations at the coarser 250-m resolution, with a less direct influence of the 250-m topography on the simulated snow patterns.

## 4.5 Limitations of simulations

In this study, we tested the sensitivity of simulated spatial snow height and SMOD variability to the variability of precipitation, snow transport, and spatial resolution, but numerous other processes shape snow spatial variability in the field, as reviewed in Mott et al. (2018). Those processes occur at different scales, going from the regional scale to the slope and lower scale. It is

worth noticing that neither preferential deposition nor the seeder-feeder mechanism is represented in our simulations. Additional errors in snow-wind interaction come from process modeling. In our simulation, only the blowing snow redistribution process is represented, the preferential deposition and snowfall enhancement are not. Blowing snow transport parameterization also suffers from uncertainties. The parameterizations used in our model were developed for flat areas and the question remains on using them in complex turbulent mountain topography. As discussed by Aksamit and Pomeroy (2018), turbulence gust wind, and eddies are some of the main small-scale contributors to blowing snow. In any case, the DEVINE downscaled wind does not yet take into account wind turbulence and recirculation zone. Further studies are needed to quantify the relative impact of recirculation zones over lee deceleration at the 250 m spatial resolution used in this work.

Moreover, in our simulations not all cover types and processes are represented. Forests and glaciers are masked out for comparisons in addition to unusable observed pixels (due to topographic shadows or other observation limitations). The necessity of using these masks inevitably has an impact on the results by reducing the number of pixels in use for the evaluation. For masked forest pixels, we know that snow transport is strongly inhibited in forested areas leading to the creation of snow accumulation at the forest edges (Bernhardt et al., 2012). It is not possible to account for this phenomenon solely through post-processing masks. In our simulation domain, the main forested areas are located at low elevations where snow transport is minimal or non-existent. On the other hand, glaciers are generally located in areas prone to blowing snow. Removing glaciers from the analysis significantly decreases the total number of pixels subject to snow transport at higher altitudes, but it does not introduce bias to pixels at the edges of the mask. Although the simulated and observed snow heights are not directly usable, it is important to note that snow transport is still simulated on masked glacier pixels and thus can contribute to neighbor pixels' snow balance.

### 4.6 Uncertainties and limitations of observations

In this work, simulations are evaluated against spatialized snow observations, giving a larger perspective than point snow observations. Indeed, although point scale snow observations are commonly used to evaluate spatialized simulations (Horton and Haegeli, 2022; Vernay et al., 2022; Vionnet et al., 2022; Mott et al., 2023; Marsh et al., 2020), their spatial distribution and representativeness is quite low in mountains (Grünewald and Lehning, 2015; Pomeroy et al., 2009; Pepin et al., 2015), while satellite snow observations like Pléiades snow height and Sentinel 2 SMOD have a much better spatial coverage and representativeness. Nonetheless, some errors and limitations also come from the observation side. The Pléiades observations are on-demand snapshots of snow height over a predefined extent and at a specific date. By design, this limits the usage of Pléiades images and the extent to which our conclusions stay valid. Pléiades observations of snow height can be affected by shadows, steep slopes, and the reference image used in the processing method. These limitations restrict the domain of observations and make it difficult to retrieve accurate snow height on glaciers. The Pléiades horizontal resolution of 2 m causes challenges in comparison with coarser resolution simulation. Unsimulated sub-grid processes are captured by the observation, which complexifies the analysis (Fig. 4). For the Sentinel 2 observations, the major limitation comes from the snow detection algorithm. The presence or absence of snow is not directly comparable to our simulation output. In Sect. 2.8.1, the presence or absence of snow is expressed as a function based on a given simulated snow height threshold. This raises the question of the sensitivity to this threshold (Gascoin et al., 2019; Hofmeister et al., 2022).

## 5 Conclusions

We conducted a spatialized evaluation of the blowing snow model SnowPappus (Baron et al., 2024) joint with a sensitivity analysis to alpine precipitation variability on a 902 $km^2$ area representative of the French Alps and the Pyrenees in terms of landforms. We evaluated the simulated snow height and Snow Melt Out Date (SMOD) using Pléiades and Sentinel 2 satellite products. The Pléiades comparisons were conducted in two different areas, on four different dates and during three different snow seasons. Eight different snow simulations have been run over three snow seasons with three different precipitation forcings and two simulation resolutions (Table 1). We performed simulation analysis with the aim of disentangling simulation error contributions from the SnowPappus blowing snow model, the precipitation forcings variability, and from the unresolved subgrid variability. The main conclusions of this study are :

– Simulations without snow transport are found unable to capture the spatial variability of snow cover in alpine terrain. The addition of SnowPappus' snow transport in the simulations results in a more physically realistic assessment of snow height and SMOD, regardless of precipitation forcing, and leads to simulated snow height and SMOD closer to the observations. Furthermore, the use of the SnowPappus model increases the variance of simulated snow height regardless of precipitation forcing bias.

– Precipitation errors are identified as the main source of bias and standard deviation in snow height and SMOD at low to medium altitudes. This result underlines that the greatest care must be taken to obtain the most accurate precipitation fields for snow simulation. It also suggests that assessments of spatial snow simulations that do not account for precipitation uncertainty are unlikely to provide informative insights regarding the accuracy of particular simulated snow processes.

– Blowing snow transport impact on snow height, SMOD bias, and standard deviation prevail for high altitudes or pixels classified by the landform classifier as 'Peak (summit)' and 'Ridges'.

– The changes in simulation snow height and SMOD mean SPS when increasing computation resolutions from 250 m to 30 m are minimal. For Pléiades observations, the change is less than 0.01 m, and for SMOD, it is less than 1 day. These changes in spatial variance and bias are also lower than the simulated spatial variability due to precipitation inputs and the impact of blowing snow in the affected areas.

– The addition of the SnowPappus blowing snow model to a 250 m Crocus snowpack simulation consistently increases and improves the simulated snow spatial variability at high elevations in accordance with observations. However, improvement of snow height and SMOD biases are only obtained using precipitation forcing that does not suffer from strong negative biases.

The findings of this study show promising results of using the SnowPappus blowing snow model for large-scale modeling of alpine snowpack. Future work will focus on adding to the simulation workflow assimilation methods to improve precipitation estimates. Our results suggest that snow transport impact on snow height at 250 m resolution is significant at high elevations

and therefore needs to be accounted for. However, unresolved processes at this spatial scale, parameterization uncertainty, and evaluation challenges are responsible for large uncertainties that will need to be appropriately quantified in future works, especially with the goal of designing a snow ensemble assimilation frameworks (e.g. Cluzet et al., 2021; Deschamps-Berger et al., 2022).

*Code and data availability.* The SnowPappus blowing snow model is developed in the framework of the open-source SURFEX project.
The source files of the SURFEX system (Crocus snow model, ISBA ground model, and SnowPappus model) are provided at https://doi.org/10.5281/zenodo.7687821 to guarantee the permanent reproducibility of results. However, we recommend that potential future users and developers access the code from its Git repository to benefit from all tools of code management (history management, bug fixes, documentation, interface for technical support, etc.). This requires a free and quick registration. The procedure is described at https://opensource.umr-cnrm.fr/projects/snowtools_git/wiki/Install_SURFEX. The version used in this work is tagged as SnowPappus-v1.0. A
user manual, describing the SURFEX namelist options related to SnowPappus is available at https://doi.org/10.5281/zenodo.7681340. More general information about SURFEX use can be found at https://github.com/UMR-CNRM/snowtools.

The code and model weights used for DEVINE are available freely at https://github.com/louisletoumelin/neural_network_and_devine/.

The WhiteboxTools Open Core software used to compute the landforms classification is freely available at https://www.whiteboxgeo.com/.

The Earth System Modeling Framework (ESMF) software is freely available at https://earthsystemmodeling.org/.

The GDAL software is also freely available https://doi.org/10.5281/zenodo.8340595.

The DEM used in this study originates from IGN® website. Tiles can be downloaded freely at https://geoservices.ign.fr/rgealti.

Forest mask originates from IGN® and can be downloaded freely at https://geoservices.ign.fr/bdforet.

Waterways, lakes, and cities' spatial extent originate from IGN® and can be downloaded freely at https://geoservices.ign.fr/bdtopo.

The glacier masks can be downloaded freely from the Randolf glacier inventory at http://glims.colorado.edu/glacierdata/.

The SAFRAN reanalysis dataset is freely available on the AERIS data center on the following link https://doi.org/10.25326/37#v2020.2.

The AROME and ANTILOPE precipitation dataset can be requested online https://donneespubliques.meteofrance.fr/.

The Theia level 3B snow products can be accessed at https://www.theia-land.fr/en/product/snow/.

The raw Pléiades images are not publicly available. The 250 m averaged Pléiades images can be found at https://doi.org/10.5281/zenodo.10037253.

(The data accessibility and URL links status is confirmed on March 22, 2024)

*Author contributions.* A.H. ran the simulations, collected and processed the different data, wrote the paper, and made the spatial evaluations and the scientific choices. A.H. and M.B. developed jointly the SnowPappus code with equal contribution. M.B. helped write the Crocus and SnowPappus sections. M.L. supervised the work, helped a lot with the scientific choices, and provided technical support on Crocus and SURFEX. M.L. and M.D. were extensively involved in the proofreading process and the scientific choices. L.LT provided the DEVINE
downscaled wind forcing and helped in writing the wind DEVINE section. M.V. provided the ANTILOPE precipitation forcing and helped the writing of the ANTILOPE section. C.D.B. and S.G. provided the Pléiades images and the stereo-imagery treatment. C.D.B., S.G., and V.V. helped with the evaluation methodology, participated in theoretical discussions, and proofread the article.

*Competing interests.* One of the co-authors is a member of the editorial board of The Cryosphere. Other authors have no competing interests to declare.

*Acknowledgements.* This investigation was done using Meteo France's CNRM's, and CEN's computational and human resources. CNRM/-CEN is part of LabEx OSUG@2020. The main source of funding is the Region Auvergne Rhône-Alpes (France) SENSASS project and CNES TOSCA. We gratefully acknowledge Sabine Radanovics, Diego Monteiro, and Matthieu Fructus for their expertise.

## Appendix A:  Additional domain characteristics

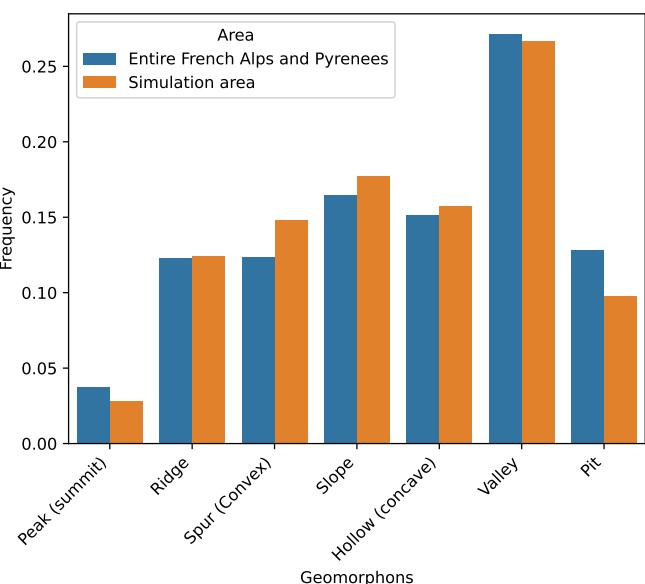

**Figure A1.** Landforms classification frequency (at 250 m) for our simulation domain and the entire French Alps and Pyrenees (as an area defined in Vernay et al. (2022)). We found a good agreement between our simulation pixel classification and the frequency of the French Alps and Pyrenees. We note that using 250 m resolution, no pixel corresponding to shoulder, footslope, and flat are classified as such.

**Appendix B:  Additional map of mean daily precipitation variability**

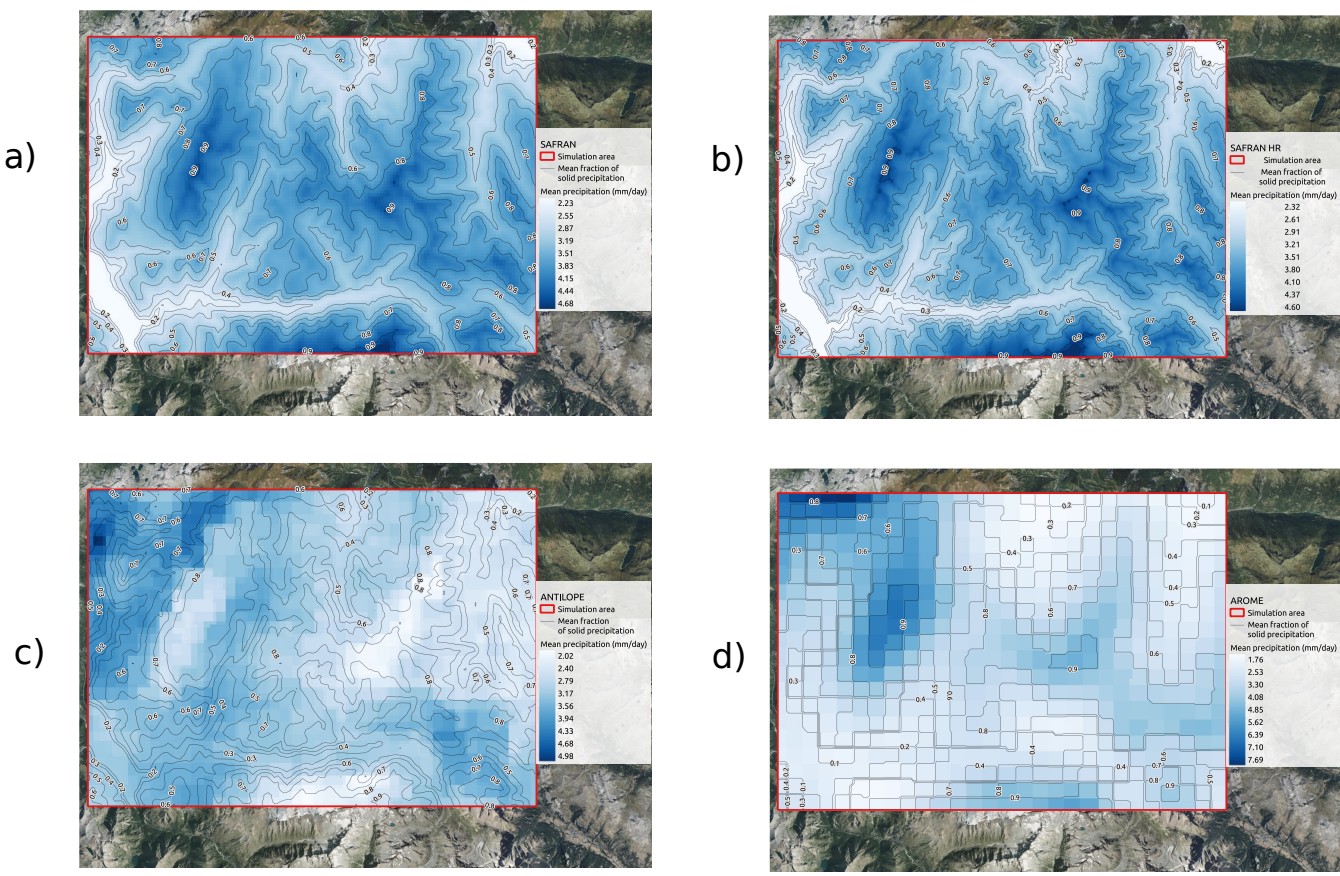

**Figure B1.** Map of the mean daily precipitation over 2017-2020 in (mm/day) for the 4 different forcings. This illustrates the spatial variability of each precipitation dataset (a. SAFRAN, b. SAFRAN HR, c. ANTILOPE, d. AROME). In each map, the mean solid fraction of precipitation is displayed as a contour plot (i.e. 1=only solid precipitation). The summer aerial photography base map is from IGN© (2022).

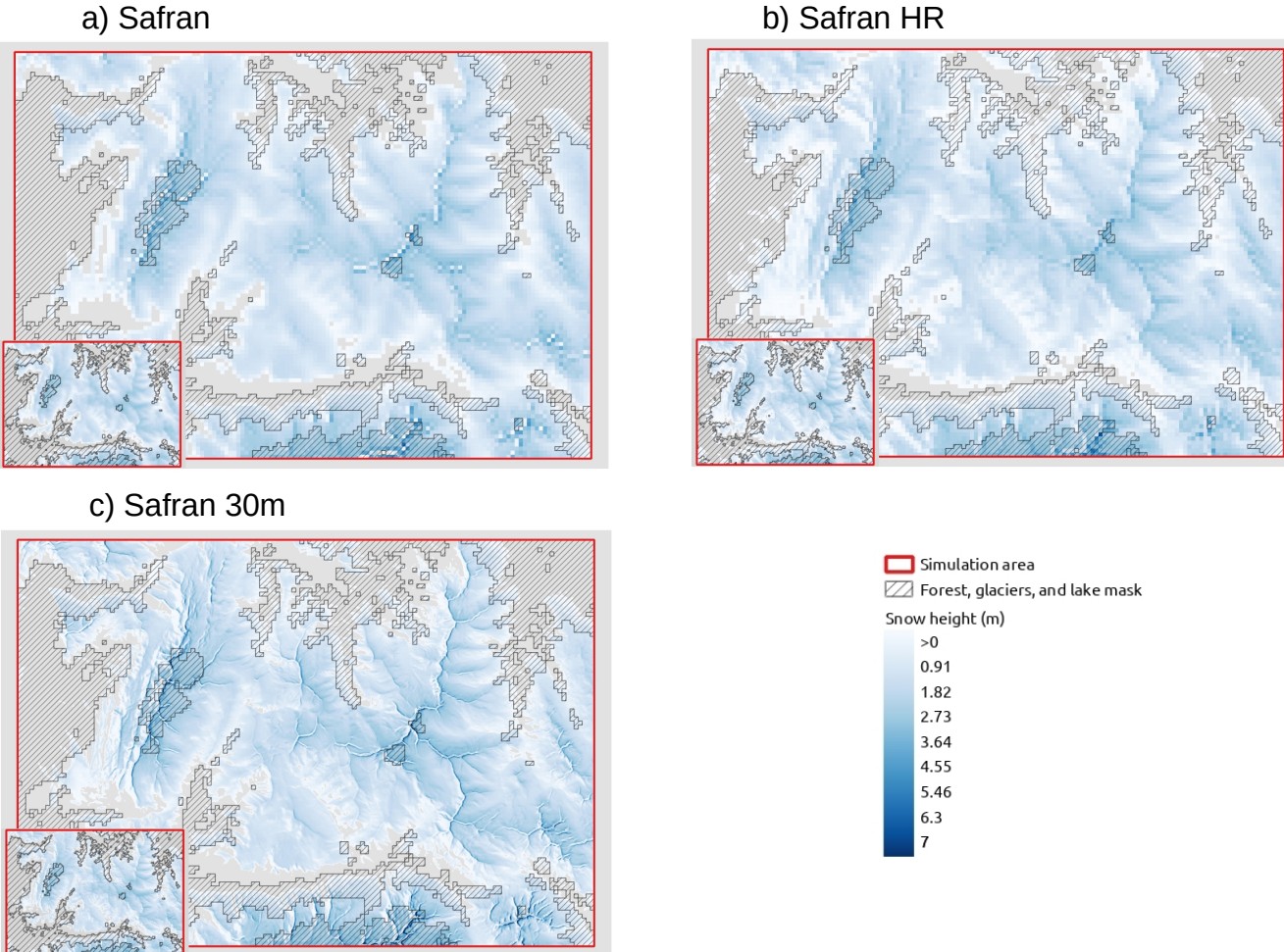

**Figure C1.** Map of simulated snow height on the 13 May 2019 10:00. (a) Safran 250m simulation with snow transport and without snow transport in the left submap. (b) Safran HR simulation (30 m computed resampled to 250m) with snow transport and without snow transport in the left submap. (c) Raw Safran 30m simulation with snow transport and without snow transport in the left submap. The absence of snow is indicated in grey.

## Appendix D: Additional landform analysis

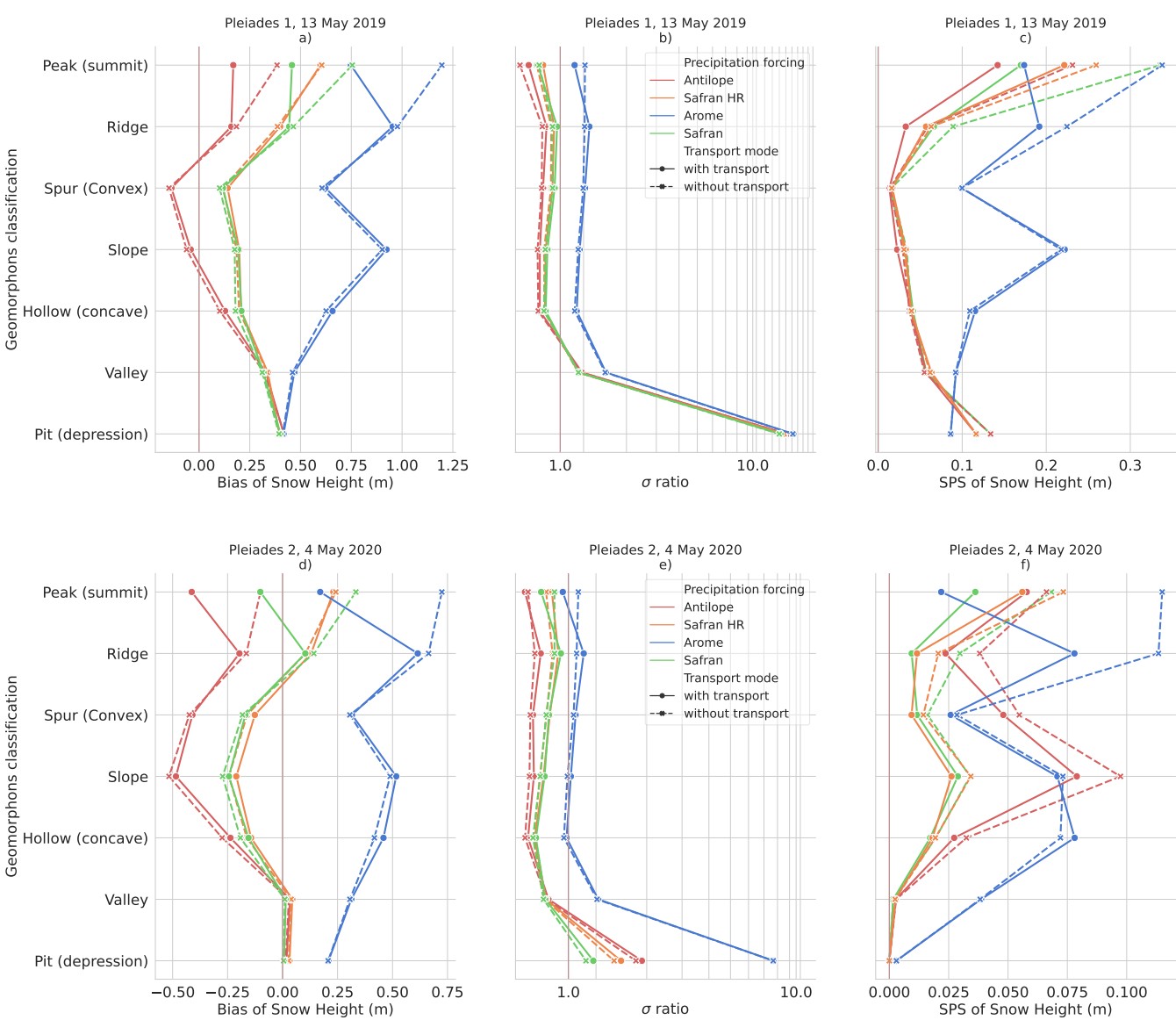

**Figure D1.** Scores comparing simulations with the Grandes Rousses Pléiades observations. (a-b-c) compares simulations with the 13 May 2019 Pléiades 1 image ; (d-e-f) is based on the 4 May 2020 Pléiades 2 image. (a-d) Bias (m) ; (b-e) $\sigma$ ratio (no unit) ; (c-f) SPS. The important standard deviation ratio values found for the 'Pit' landform type in (b) and (e) are explained because almost all pixels of this type have no observed snow whereas the simulations give snow on every pixel. This is particularly visible in Subfigure (d) and (e) where Antilope, Safran, and Safran HR have almost no simulated snow and a fair standard deviation ratio but Antilope. All scores are presented as a function of landform pixel classification.

**Appendix E:  Additional resolution analysis**

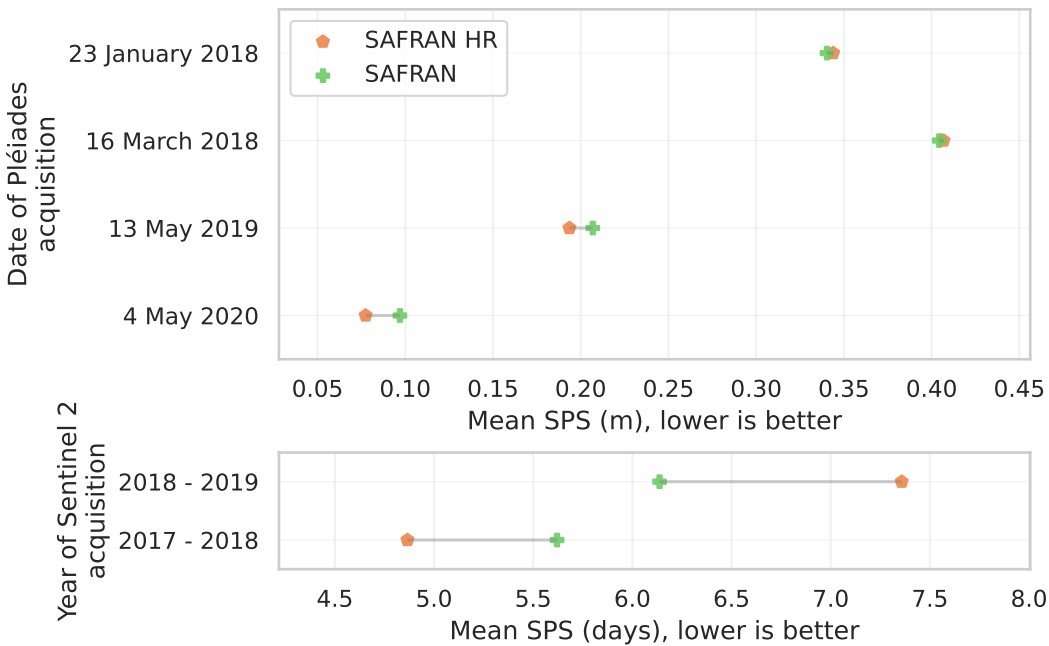

**Figure E1.** Plots of average SPS values for the SAFRAN and SAFRAN HR simulations when grouped with 300 m elevation bands. This graph illustrates the impact of the computing horizontal resolution on snow height and snow melt-out date.

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
