# Peer review of "Analysing the sensitivity of a blowing snow model (SnowPappus) to precipitation forcing, blowing snow, and spatial resolution"

_EGUsphere, 2023_

## Author Comment (AC1)

**Responses to Anonymous Referee #2**

Addressed Comments for Publication to

The Cryosphere

by

Ange HADDJERI,

Matthieu BARON,

Matthieu LAFAYSSE,

Louis LE TOUMELIN,

César DESCHAMPS-BERGER,

Vincent VIONNET,

Simon GASCOIN,

Matthieu VERNAY,

and Marie DUMONT

Dear Masashi Niwano,

Please find enclosed the detailed responses to Anonymous Referee #2 for the manuscript entitled 'Exploring the sensitivity to precipitation, blowing snow, and horizontal resolution of the spatial distribution of simulated snow cover' with manuscript number EGUSPHERE-2023-2604. We would like to thank you and the reviewers for the valuable comments which help improving the quality of our manuscript. In this revision, we have carefully addressed the reviewers' comments. A summary of the main modifications and a detailed point-by-point response to the comments from Reviewer #2 (following the reviewers' order in the decision letter) are given below.

Sincerely,

Ange HADDJERI,
Matthieu BARON,
Matthieu LAFAYSSE,
Louis LE TOUMELIN,
César DESCHAMPS-BERGER,
Vincent VIONNET,
Simon GASCOIN,
Matthieu VERNAY,
and Marie DUMONT

**Note:** To enhance the legibility of this response letter, all the editor's and reviewers' comments are typeset in boxes. Rephrased or added sentences are typeset in color. The respective parts in the manuscript are highlighted to indicate changes.

**Authors' Response to Reviewer 2**

**General Comments.**

The novelty of this work lies in the introduction and assessment of a new model architecture, none of the results are particularly surprising or ground-breaking. Comparisons are performed at 250 m scales; it is subject to debate whether this is 'snow-drift-permitting' but, in any case, it is no surprise that the key driver of spatial variability at these scales is the precipitation forcing dataset. Despite this, it is clear to see the significant added value of the SnowPappus blowing snow model that arises at high elevations and on ridges and peaks/summits.

One problem I have with the manuscript is that, to me, it seems you haven't quite decided on the ultimate goal of your study or the result which you really want to highlight, I find this dilutes the really interesting and valuable results held within. There are many points that you are claiming to address: spatially evaluate the SnowPappus model, analyse how precipitation forcing impacts models, analyse how spatial resolution impacts models, analyse how snow simulations can be assessed by landform or by elevation bands. There is a lot going on and I think the manuscript in its current structure does not aid the digestion of all this information. I think a clearer separation between these (e.g. separate figures for the analysis of spatial resolution and precipitation forcing) and the addition of some combined metrics quantifying the results would help a lot (e.g. a metric combining the results of all four precipitation forcings to directly compare with transport against without transport would help to disentangle precipitation biases from the snow transport results).

Sometimes you suggest you are trying to identify the optimal precipitation forcing dataset, e.g. P17 L 334 and L359. However, I don't think this is a key objective of the study (and not something that is found in the study), and therefore not worth including because it weakens the message you are trying to get across. Along similar lines, I think there is too much focus on the impact of the precipitation forcing and not enough emphasis on the impacts of the snow transport model, which is I believe the intended focus point of this study. In some ways, it feels like the analysis of precipitation forcings is unnecessary for the assessment of SnowPappus, we already know that the precipitation forcing data will control simulated snow, it just adds another (albeit interesting) aspect to the paper that detracts from the key point you are trying to make: SnowPappus IS improving the representation of blowing snow at 250 m resolution.

**Response:** Thank you for your thoughtful review, we appreciate your comments on our work.

We have accepted the vast majority of your comments. You are right, attempting to address multiple goals can dilute the message. We restructured the discussion to provide a clearer separation of objectives. The discussion's structure is now (1) Added value of the SnowPappus model, (2) Sensitivity to precipitation forcings, (3) Sensitivity to horizontal resolution, (4) Impact of the grid cells classification, (5) Limitations of simulations, (6) Uncertainties and limitation of observations, (7) validity of the conclusions.

We kept the same type of graphs for Fig. 5 to 12 because we believe that these two types of graphs summarise the information well and that it has value to compare directly the influence of snow transport and precipitation on the same plots. We have, however, increased the size of the figures for better visibility.

As suggested, we added 3 combined metrics to better quantify the conclusions and isolate the contribution of snow transport: Fig. 13 to quantify the impact of the grouping axis, Fig. 15 to quantify the added value of the SnowPappus model, and Appendix Fig. F1 to quantify the impact of computing resolution in simulations.

We agree that finding the best precipitation forcing is not an objective of this study. Therefore, in the new manuscript version, we removed or reformulated all sentences referring to finding "an optimal precipitation forcing".

Consistently, we have corrected the slightly unbalanced perspective to better focus on the impact of the SnowPappus transport model in the manuscript. The discussion has been rewritten to include a dedicated paragraph on the impact of the SnowPappus model on simulations. This section now clearly demonstrates that the model improves the representation of snow at 250 m horizontal resolution.

**Authors' Response to major comments**

**Comment 1**

Fig 12. Comparison of Sentinel-2 SMODs with simulations for 2017-2018 and 2018-2019.

- Has there been a mistake? Results seem to be identical for 2017-2018 and 2018-2019

- Fig 12. cannot make sense of text describing results (L468-474). Presumably because figure is incorrect.

**Response:** Thank you for the comment.

We agree with this comment and apologize for this mistake. We therefore corrected the figure in the new version of the manuscript and checked references to Figure 12 (a-b-c) in the manuscript.

The upper panel of Figure 12 (a-b-c) now rightly corresponds to the 2017-2018 year.

**Comment 2**

One major problem I have with this manuscript is the presentation of the results and consequently confusing discussion. I think the figures used could be improved and results quantified with combined metrics to (1) better support your arguments, make the results clearer/easier to discuss, potentially provide further insight that is not currently obvious, and (2) ease understanding for reader and give others greater confidence in your results. You repeatedly state it is hard to disentangle impacts of precipitation forcing and snow transport and I think this is partly due to the way you have present your results. Here are a few thoughts on how I think you could change your results section to help with your analysis:

1. You do not quantify your results at all. If you introduce some combined metrics which you could quote in your discussion, it would improve clarity and give the reader much more confidence in your results. Several examples are provided in the minor comments, e.g. P31 L621 a comparison of the mean SPS scores for landforms vs elevation bands would clearly quantify that the landform SPS values are more homogeneous.

2. I like that your figures 6,8,10,11,12 are all the same format and summarise a lot of information. These could be supplemented with figures displaying combined results, e.g. combined SPS of all precipitation forcings so that you are just showing the impact of with transport vs without transport. This could help disentangle precipitation from snow transport.

3. Providing plots (e.g. scatter) of observations vs simulation would show nicely how correlated the simulation results are to the observations. This could then be quantified with e.g. Pearson correlation coefficient.

**Response:** Thank you for the comment.

We have followed your recommendations to better support our arguments and ease the understanding of the results.

1. We have increased the quantification of results throughout the text, for both the original figures and with new combined metric figures. New Figure 13 represents a comparison of the means SPS score for landforms and elevation bands. It is used as a combined

metric in the discussion to quantify the impact of the grouping axis. New Figure 15 illustrates the mean SPS values (grouped with elevation bands) of simulations with and without snow transport. It is used as a combined metric for quantifying the added value of the SnowPappus model. Appendix Fig. F1 shows the mean SPS value for the two resolutions SAFRAN (250 m) and SAFRAN HR (30 m). It is used as a combined metric for quantifying the impact of computing resolution in simulations.

> Added combined metric Figure 13, 15, F1, and increased numerical quantification of results throughout the text.

2. Similarly to the point above, we added 3 combined metrics: Fig. 13 to quantify the impact of the grouping axis, Fig. 15 to quantify the added value of the SnowPappus model, and Appendix Fig. F1 to quantify the impact of computing resolution in simulations. For Figures 5 to 12, we kept the same format but increased the size of the figure for more visibility.

> Added combined metric Figure 13, 15, F1, and increased numerical quantification of results throughout the text.

3. As suggested, we added scatter plots of snow height for Pléiades observations vs simulations (Fig. 14) representing the four different Pléiades observations to the eight different simulations (SAFRAN, SAFRAN HR, AROME, ANTILOPE, with and without activation of SnowPappus). For each simulation, we see more dispersed scatter plots for simulations with snow transport (more points at high and low snow heights). It can be noted that the mean difference in slope caused by the addition of snow transport is small ($<0.02$). Overall, AROME simulations have the steeper slopes. We also quantified and discussed the spatial correlation between simulations and Pléiades observations using the Pearson correlation coefficient (Table 1, Table 2). Looking at the Pearson correlation for the entire Pléiades domain, the addition of snow transport in simulations decreases the correlation coefficient in 15 of 16 experiments. Restricting the correlation to pixels to elevation above 2700 m (pixels with high snow transport probability) the addition of snow transport in simulations increases the Pearson correlation for each experiment. This result is found statistically significant (with significance set at p-value $<0.05$) in 15 out of 16 experiments.

Added Fig. 14 (scatter plot) and discussion of Pearson correlation coefficient, Tables 1 and 2.

**Authors' Response to minor comments**

**Title**

> **Comment 1**
>
> I think the title is a bit vague especially the word 'exploring'. "Analysing the sensitivity of a blowing snow model (SnowPappus) to precipitation forcing, blowing snow, and spatial resolution".

**Response:** Thank you for the comment.

We agree that the title can be improved. We therefore changed the title in the new version of the manuscript as follows:

> Changed title to : Analysing the sensitivity of a blowing snow model (SnowPappus) to precipitation forcing, blowing snow, and spatial resolution.

**Abstract & Introduction**

> **Comment 2**
>
> Statement in both introduction P1 L8-10 "The results show that the SnowPappus model forced with high-resolution wind fields enhances the snow cover spatial variability at high elevations allowing a better agreement with observations above 2500 m and near peaks and ridges." and conclusion that the addition of SnowPappus provides better agreement with observations and yet the results do not show this; it is stated in the manuscript several times that this is subject to the bias in precipitation forcing. I see that you are just trying to make a generalised statement, which your results do somewhat support, however, I think is slightly inaccurate and should be replaced by a less definitive statement e.g. "SnowPappus provides more physically realistic spatial patterns of simulated snow distribution expected due to the influence of blowing snow mechanisms in the mountains".

**Response:** Thank you for the comment.

In the paper, we demonstrate that the spatial distribution of snow depth and SMOD are closer between simulations and observations (especially in terms of spatial variance) when the

SnowPappus model is activated. This is especially true at high elevations (see Fig. 5, 7, 9) and near ridges (see Fig. 11, 12). We reformulated the sentence to be clear that "better agreement" refers to spatial distribution and not to pixel-pixel comparison :

> L8-10 The results show that the SnowPappus model forced with high-resolution wind fields enhances the snow cover spatial variability at high elevations allowing a better agreement between observed and simulated spatial distributions above 2500 m and near peaks and ridges.

**Comment 3**

P2 L40 "However, the quantification of the precipitation is still impacted by important uncertainties." I think you should at least briefly try to explain what 'important uncertainties' you are referring to.

**Response:** Thank you for the comment.

We totally agree. The quantification of precipitation error is difficult in alpine areas. The modeled precipitation uncertainties come from errors in precipitation amounts and phases, localization, and timing. Therefore this information was added in the text.

> L41 However, the quantification of the precipitation is still impacted by important uncertainties with errors in precipitation amounts and phases, localization, and timing increased with coarse grid size

**Comment 4**

P2 L40-45 I think this needs rewording, it isn't clear what you are trying to say, especially the last sentence ("Eventually, the representation of post-depositional processes, notably blowing snow transport, benefits from the development of dedicated high-resolution models coupled to or forced by atmospheric models.") given you refer to it several times in the next paragraph. On that note, P2 L46 "The spatial evaluation of this last type of system is a challenge in itself." I don't think you should refer to the last paragraph ('this last type of system') without more explanation. It makes it unclear and hard to follow.

**Response:** Thank you for the comment.

We apologize that the paragraph was unclear. We have changed the paragraph structure and made the following changes.

L45-49 Due to the complex intertwining of snow variability processes, in particular blowing snow transport, the community found benefits in the development of dedicated high-resolution models coupled to (e.g. Vionnet et al., 2014; Sharma et al., 2021) or forced by atmospheric models (e.g., Lehning et al., 2006; Liston et al., 2007; Marsh et al., 2020; Baron et al., 2024; Quéno et al., 2023). The spatial evaluation of this type of system, dedicated to modeling part of the observed snow spatial variability is a challenge in itself.

**Data and Methods**

**Comment 5**

P5 L125-126 I like the decomposition into landforms. I know you have 250m pixels, but is there really nowhere in the entire French Alps and Pyrenees that is classified as flat, shoulder, or footslope?

**Response:** Thank you for the comment.

The definition of a geographical surface that someone considers to be flat can vary greatly depending on the scale considered and the variation in height of the surrounding area. In the algorithm used for pixel classification (WhiteBox geo), the definition of a flat pixel requires the

8 line-of-sight analysis made in the cardinal directions to be at the exact same altitude plus or minus a flatness threshold defined in degrees for the chosen search radius. We have defined this threshold at 1 degree and the search radius of 2.5km. This explains why for a horizontal resolution of 250m there are no pixels classified as rigorously flat by the algorithm. The same is true for footslopes and shoulders that need 5 of the 8 cardinal directions to be considered "flat". In our classification almost "flat" or 'footslopes' areas are instead considered valley or pit. Ambiguous 'shoulder' areas are instead considered as slopes, hollow or ridges.

[Figure]

Figure 1: The 10 different landforms defined by Jasiewicz et al. 2013.

We added the following note in Fig. A1 legend and updated the figure to remove the unused keys.

> Appendix A Figure A1, We note that using 250 m resolution, no pixel corresponding to shoulder, footslope, and flat are classified as such.

**Comment 6**

Inaccuracies in remote sensing datasets (Pléiades and S2 Theia snow collection). These are mentioned briefly in passing. But I think it would be worth specifically mentioning the uncertainty in the 'observations' you are assessing the model performance against.

**Response:** Thank you for the comment.

The vertical Pléiades error depends on the resampling resolution. At the 250 m resolution, as stated in section 2.8.2, the Pléiades vertical snow height standard error is between 0.3 and 0.4m (Deschamps-Berger et al., 2020). (S2.8.2 L252) For Sentinel 2 in section 2.8.1, the accuracy

depends on the snow height threshold chosen. For our study, with 20 cm as our threshold, the Sentinel 2 accuracy is 91.5%, see Figure 2. We added the following sentence in the text.

L233 For this threshold, the accuracy is found to be 91.5% (See Fig 10 Gascoin et al., 2019).

[Figure]

Figure 2: Figure 10 from (Gascoin et al., 2019), Sensitivity of the agreement between the in situ and Theia snow product to SD0 in meters (threshold to convert the measured snow depth to snow presence or absence). The inset shows a close-up of the region of 0–0.2 m.

**Comment 7**

I'm unsure why you needed to use the very high-resolution (and expensive) Pléiades imagery for this experiment given you are resampling to 250 m anyway, and even the SAFRAN HR is 30 m. With Sentinel-2 offering free, open-access imagery at 10 m (20 m including the SWIR), was there much benefit form using this expensive, super high-res imagery? As stated in your discussion, P30 L603 "Unsimulated sub-grid processes are captured by the observation [Pléiades], which complexifies the analysis." So why use such high resolution and complicate analysis?

**Response:** Thank you for the comment.

Indeed, we do not benefit from the highest spatial resolution of Pléiades observations but their main interest compared to Sentinel 2 or any other available data is the fact that they produce spatialized estimates of snow height which are much more informative than using snow cover fractions and their derivatives, SCD, SMOD, etc. and are not available from other sensors. The use of several different variables for the evaluation increases the robustness of the evaluations.

**Comment 8**

P8 L174 Precipitation phase threshold. If this is not the method of precipitation retrieval used in the operational AROME NWP then this seems like an unfair comparison, especially if you are trying to decipher the optimal precipitation forcing (which you state a few times e.g. P17 L358-359). I understand for comparison purpose keeping the threshold consistency between forcing data is a sensible thing to do. Perhaps don't say that you are looking for the optimal precipitation forcing dataset because, as far as I understand, this is not the objective of the study.

**Response:** Thank you for the comment.

We totally agree. Finding the best precipitation forcing is not an objective of this study. As you have understood, we chose an identical precipitation phase threshold to increase consistency between forcings. Therefore, in the new manuscript version, we removed all references referring to finding "an optimal precipitation forcing". Note also that the raw precipitation phase of AROME forecasts is probably not more accurate than the estimates of the method. Indeed, in

another context (Vionnet et al., 2022) have demonstrated that post-processing methods can easily outperform the raw outputs of the microphysics scheme of the atmospheric model.

**Comment 9**

P11 L236-237 "A reference snow-free surface elevation is acquired during the summer." I was unable to decipher what is used as your reference surface elevation for Pléiades snow heights. I think it would be worth clearly stating this.

**Response:** Thank you for the comment.

The production of Pléiades snow height maps involves capturing multiple snapshots of the same area with different view angles to derive a surface elevation field. To produce snow height maps, this method needs to be done two times. A first acquisition gathers the reference ground surface elevation, ideally in summer or autumn during a snow-free day over the entire domain. A second acquisition is done during winter with snow on the ground. For each grid point, snow height is defined as the difference between the two surface heights. For the Grandes Rousses area, the snow-off acquisition is 2016-09-28. For the Galibier area, the acquisition is 2018-10-24. The aim was to obtain a snow-off image during the minimal snow cover period of the study area. The paragraph describing the Pléiades images has been revised for more clarity and the exact dates of each acquisition for each image have been added.

L237-L244 In order to evaluate more directly the spatial variability of the simulated snow height, we used snow height maps derived from Pléiades stereo-images (Deschamps-Berger et al., 2020; Marti et al., 2016). In both cases, snow height is defined following a vertical axis. The Pléiades raw images have a 0.5 m horizontal resolution and are acquired on request. Snow height maps are obtained by combining two surface elevations of the same area, one with snow and one snow-free. Each surface elevation map is derived from pairs or triplets of Pléiades images. The reference snow-off surface elevation is calculated from Pléiades stereo-images acquired on 28 September 2016 for the Galibier area and 24 October 2018 for the Grandes Rousses area. A winter image acquisition gives a second elevation map with snow on the ground. The difference between the two elevation fields provides a snow height map. The generated snow height map has a horizontal resolution

of 2m.

**Comment 10**

P11 L252 no justification for 70% threshold

**Response:** Thank you for the comment.

This is a necessary threshold that balances the number of missing pixels and the representativeness of the obtained 250m average. For our experimental zone, we found that 70% was a good compromise between the number of 250 m pixels (a lower threshold would significantly increase the number of missing pixels and reduce the evaluation area). This threshold also permits the resampled value to be at least representative of 30% of the area. We changed the manuscript as follows :

L257-L258 The entire 250 m pixel is set to No Data if more than 70% of the 2 m pixels is No Data, this threshold ensures the minimum representativeness of the pixel's snow height.

**Comment 11**

P11 It is unclear if your Pléiades snow height maps are 0.5 m or 2 m resolution? Might be worth clarifying.

**Response:** Thank you for the comment.

We apologize this was not very clear. The raw Pléiades snapshots are 50cm resolution but with a restraint access. The raw snapshots are used to reconstruct the height fields using stereo-imagery. During this process we have a loss of resolution and the obtained snow height fields are 2 m horizontal resolution. We cleared up the Pléiades product resolution in the text.

> L239 Pléiades images have a 0.5 m horizontal resolution and are acquired on request. [...]
> L244 The generated snow height map has a horizontal resolution of 2m. [...] L256 For
> comparison with snowpack simulations, the 2 m resolution Pléiades snow height map is
> resampled to the 250 m horizontal resolution grid using a conservative method [...]

**Results**

**Comment 12**

Jumping between figures sometimes makes text hard to follow. Consider describing one at a time.

**Response:** Thank you for the comment.

To improve the paper's clarity and flow, we have chosen to describe the results from the same area and date in the same section. We have used similar types of figures to display the different data. This avoids redundancy between sections because various features can be seen both in violin plots and in scores and therefore are easier to describe only once. This choice also makes it easier to compare similar graphs and results and helps shorten the paper. We think this is the appropriate choice due to the significant amount of figures that require description, especially with the inclusion of new figures as requested by the reviewers.

**Comment 13**

P13 First paragraph in section 3.1.1 is not specific to Grandes Rousses area, perhaps this should be moved out of this subsection (e.g. just above the heading for this section). Personally, I think information of the Galibier region should be added to fig 4.

**Response:** Thank you for the comment.

We agree with this comment, the first paragraph in section 3.1.1 is not specific to the Grandes Rousses area and aims to spatially illustrate the simulated and observed spatial variability. We moved this paragraph to section 3.1. This paragraph and figure 4, contribute to illustrate the

variability of the simulated snow height on our domain. In addition to the simulations, we choose to add the Pléiades image of 13 May 2019, enabling to visually see the data used in the first violin plot Fig. 5 a). As all the maps in Fig. 4 are from 13 May 2019, while the Pléiades images over the Galibier area are only available for other dates, we feel that it is not relevant to add a Pléiades image from another area on another date to this illustration.

We have reworked paragraph 3.1 and moved figure 4 into it. We have modified the beginning of the following paragraph 3.1.1 accordingly.

> L295-302 An example of simulated and observed snow height variability can be found in Fig. 4. In this figure, snow height is simulated on 13 May 2019 for the Safran, Arome, and Antilope precipitation forcings, and observed for the Grandes Rousses area on the same date. For each precipitation forcing, the subfigure shows from one side the impact of the SnowPappus blowing snow model on the simulated snow height, where we can see areas of increased and decreased snow height, most visible along the ridges and peaks, and in each left sub-maps the same simulation without snow transport. Additional maps for the 13 May 2019 comparing resolutions can be found in Appendix Fig. C1.

**Comment 14**

P17 L334 "Thus, no optimal precipitation forcing can be identified systematically." Did you really expect to with this study? You could get a combined metric of the best of these precipitation forcings for the test region. Unsure if analysis of precipitation forcing datasets is necessary for the point you are trying to make with this paper.

**Response:** Thank you for the comment.

This question is similar to Comment 8, finding the best precipitation forcing is not an objective of this study. We have removed from the manuscript sentences suggesting we intended to find an optimal precipitation forcing.

**Comment 15**

P17 L335 "ANTILOPE forcing leads to the simulated snow height the closest to the observations for both dates above 2200 m" this isn't completely true. On 16 March in 2800 to 3100 m elevation band ANTILOPE is not closest.

**Response:** Thank you for the comment.

Indeed, in Fig. 8 (c-f) the ANTILOPE forcing is not the lowest score for the upper elevation on March 2018. We corrected this sentence as follows :

L363 For the SPS score (Fig. 8 (c-f)), the ANTILOPE forcing produces low SPS scores ($\leq$ 0.1 m) between 2200 and 2800 m, and for both dates, in contrast to the results obtained over the Grandes Rousses area.

**Comment 16**

P18 L366 "However, no precipitation forcing leads to the simulation of the most unbiased snow heights for all areas and evaluation dates." Unclear how you got to this conclusion without quantifying with a combined metric for each area. If you did this, it would be worth stating the results.

**Response:** Thank you for the comment.

First, as said before, finding the best precipitation forcing is not an objective of this study. However, without using a dedicated metric, a look at Figs. 6, 8, and 10 (a-d) show us that no forcing produces a systematically less biased snow height for all domains and dates. We changed the sentence to convey this idea more clearly :

L374 However, upon examining the various bias scores (as shown in Fig.6, 8, 10 (a-d)), it is clear that no precipitation forcing produces a snow height simulation that is systematically less biased than the others over the different evaluation dates and areas.

**Comment 17**

P18 L369 "The addition of snow transport increases and improves the simulated standard deviation" quantify this, i.e. how much closer adding snow transport bring results to the Pléiades standard deviation.

**Response:** Thank you for the comment.

On average for the Pléiades observation, the addition of snow transport in the simulation increases the standard deviation ratio of 0.25. In addition to this result, we added section 3.4 and Fig. 13 describing and quantifying more precisely the impact impact of the SnowPappus model on the simulations.

> We quantified the change in simulated standard deviation. Added section 3.4 and Fig. 13.

**Comment 18**

P20 section 3.2.1 states that it is about "differences between years" when it is only a concerning the differences in variability between years. Much more could be included here.

**Response:** Thank you for the comment.

We have expanded this paragraph by adding more comparisons between years, the most important information is that 2018-2019 has an earlier melt date than the previous year and that this result does not seem to be sufficiently well reproduced in the simulations.

> Expanded section 3.2.1

**Comment 19**

P20 L403 "On average, the spatial variance of SMOD is underestimated except for simulations with AROME forcing at the two lowest elevation bands where the $\sigma$ ratio is found close to 1." It is unclear to me how this conclusion is drawn.

**Response:** Thank you for the comment.

Our apologies, there was an error in this statement and we have corrected it.

> L420 On average, the spatial variance of SMOD is underestimated except for simulations with the ANTILOPE forcing at the upper elevation where the $\sigma$ ratio is found much above 1.
* * *
**Comment 20**

P21 L 404-406 "It is interesting to note that in the lowest elevation band Fig. 9 (b), an AROME pixel never has a snow height greater than the SMOD threshold during the year (see Sect. 2.8.1) and is therefore given 405 a 0-day SMOD value, contrary to what happens for all the other simulations and the observation." This is not shown in figure 9, the AROME distribution does not stretch to 0, why?

**Response:** Thank you for the comment.

We apologize for the error in this statement. It comes from a preliminary result, before a change in masks of glaciers and forests. After this, we forgot to remove this sentence. This statement is no longer true, so we removed it.

> We removed this statement.
* * *
**Comment 21**

Why did you choose to have a figure for the distribution of snow heights and SMOD for elevation intervals but not for landform subgroups?

**Response:** Thank you for the comment.

We decided to not add a violin plot graph to represent landforms with Sentinel 2 in the manuscript to reduce the number and redundancy of graphs. We found that the information was adequately summarised in the scores of Figure 12.

**Comment 22**

P25 I found the content concerning figure 12 confusing. Likely because of the accidental use of the same plots for 2017-2018 and 2018-2019.

**Response:** Thank you for the comment.

Similar to Major comment 1. We apologize for this mistake.

> We corrected the upper panel of Figure 12 in the new version of the manuscript.

**Comment 23**

Following on from my 'major comment' on figures, I have outlined why I struggled with the current figures and a few areas I think could be improved if you choose to stick with the current format. A specific and repeated comment is the figures are too small to see or overcrowded.

- Fig 1 personal preference, but the inverse colourmap is more intuitive in my opinion (light as high elevation, dark as low elevation).

- Fig 4 and C1

  – please explain weird scalebar

  – hard to see smaller submaps, consider using difference maps to aid comparison.

  – Fig 4, if this is specific to analyse the results for Grandes Rousses you do not need the entire domain showing. If it is to visualise the entire area, why have you not shown the Galibier area?

  – Areas that are masked are still included in the plot. You are not using these masked areas in your analysis, so is there a justification for including it? If not it is just unnecessary noise and should be removed. I understand it aids your description of the impacts of SnowPappus e.g. P13 L293, however, you have decided to mask these regions for several valid reasons, and therefore it is not a reliable result to draw conclusions from.

**Response:** Thank you for the comment.

- We have changed the colormap of Fig. 1 to a more classical colormap used for terrain elevations (white is high elevation and dark green is low elevation).

- Fig. 4 and C1

  – The scalebar values were set automatically. It was changed to display clearer values.

  – Comment linked to Comment 13. The point of Fig. 4 was to illustrate visually simulation outputs and we chose the same date as for the next Figure (5a). Because snow simulations without snow transport are strongly homogenous, we decided to put them in small. The added value to put them in full scale was found to be low. Adding different maps does not go with the aim of illustrating simulation outputs.

  – Comment linked to Comment 13 and the above. The point of Fig. 4 was to illustrate raw simulation outputs. The interest of showing the whole simulation domain is that ridges of larger extents can be seen illustrating well the impact of snow transport in our simulations. This graphic was not intended to be specific to the Grandes Rousses, but we saw the opportunity to show a Pléiades image as well for a better understanding of the results of Figure 5. We have chosen the Pléiades image of 13 May 2019, which is the first to be described in the paragraph directly following figure 4. As explained before, the Galibier area is not covered by Pléiades images at this date. As discussed in the response to comment 13, we have reworked paragraphs 3.1 and 3.1.1.

  – Comment linked to Comment 13 and the two above. The hatched areas in Figure 4 are areas that have been masked by the analyses due to the presence of forests, towns, or glaciers on these pixels affecting the validity of observations. However, as noticed by the reviewer, to better illustrate the impact of SnowPappus on the simulations, it is interesting to show the simulated values inside the masked areas even if they have not been used in the evaluations against satellite observations.

> **Comment 23**
>
> - Fig 5,7,9 - VIOLIN PLOTS. . . .
>
>   - Size needs to be increased dramatically. I strongly recommend taking the height of the entire page if sticking with this format, so that the distributions can be interpreted. In print I can barely see the hatching, the white dots for median values disappear entirely. Even on my monitor I have to zoom in massively to decern what you are discussing.
>
>   - I feel the figure needs more explanation, even if to you it is self-explanatory why make it hard for someone who doesn't use violin plots. No mention that white dot is median and black box with first and third quartile (presumably). No mention of what the plot is showing (I know it might be obvious but still worth stating), presumably a frequency density of data. N is presumably the number of sample pixels in that subgroup.
>
> - Fig 6,8,10,11,12 – comparison metrics
>
>   - No need to remove acronyms in legend.
>
>   - Orange and red not easily distinguishable, especially in print.
>
>   - Consider more contrasting colours.

**Response:** Thank you for the comment.

- Violin Plot

  - The size aspect ratio of every violin plot figure has been changed so that each figure takes up a full page.

  - Following your comment, we improved the violin plot legend, adding a description on the inner black box (1 and 3rd quartile), white dot (median), and the N value (number of samples). The following sentences have been added.

    > Added in the legend of Fig 5, 7, 9 Each violin contains a small inner black box representing the first and third quartile, and a white dot representing the median distribution value. The number of samples for each subgroup is denoted by $N$.

- Comparison metrics
  - We like the legends on our graphs to be as self-explanatory as possible, and as we couldn't add the identical symbol for $\sigma$ ration x-axis legend, we preferred to include the full description of the variable.
  - For these figures we use the color palette recommended by Okabe and Ito, 2002 (green, blue, orange, vermillion), defined by the authors as being the best set of colors that is unambiguous both to colorblinds and non-colorblinds.
  - Please refer to the answer above for the colors. To improve furthermore the visibility of the different lines, we have tripled the line thickness.

**Discussion**

**Comment 24**

Discussion could be condensed to convey your key points more clearly and concisely.

**Response:** Thank you for the comment.

We have reworked the content of the discussion and changed the structure to convey our key points more clearly and concisely. The discussion's structure is now (1) The added value of the SnowPappus model, (2) Sensitivity to precipitation forcings, (3) Sensitivity to horizontal resolution, (4) Impact of the grid cells classification, (5) Limitations of simulations, (6) Uncertainties and limitation of observations.

**Comment 25**

Substantial regions of your domain are being masked. Some of these are the regions that are most affected by blowing snow, e.g. topographic shadows. I understand this is often done given the limitations of both models and remote sensing products in these regions, but I think it would be worth mentioning how this could have influenced the results. E.g. there is a substantial amount of snow on glaciers, and topographically shaded regions are where the snow will be redeposited by wind blow processes and therefore be deepest and last longest.

**Response:** Thank you for the comment.

This comment is linked with comment 27.

That is right, the masking of glaciers and forested pixels influences the results. Glacier masking reduces the number of usable pixels at high altitudes, pixels usually strongly prone to snow transport. The same is true for pixels masked because of observation limitations. We have clarified in the text how this could have influenced the results.

> L667-676 Moreover, in our simulations, not all cover types and processes are represented. Forests and glaciers are masked out, in addition to unusable observed pixels (due to topographic shadows or other observation limitations). The necessity of using these masks inevitably has an impact on the results, reducing the amount of pixels in use. [...] In our simulation domain, the main forested areas are located at low elevations where snow transport is minimal or non-existent. On the other hand, glaciers are generally located in areas prone to blowing snow. Removing glaciers from the analysis significantly decreases the total number of pixels subject to snow transport at higher altitudes, but it does not introduce bias to pixels at the edges of the mask. Although the simulated and observed snow heights are not directly usable, it is important to note that snow transport is still simulated on masked glacier pixels and thus can contribute to neighbor pixels' snow balance.
* * *
**Comment 26**

P25 section '4.1 Sources of snow spatial variability'. When comparing with Sentinel-2 SMODs for 2017-2018 and 2018-2019 I think it is worth mentioning if there were any dramatic differences in precipitation and wind in those years which could impact the spatial variability.
* * *
**Response:** Thank you for the comment.

The S2M (SAFRAN–SURFEX/ISBA–Crocus–MEPRA) meteorological and snow cover reanalysis from 1958 to 2021 (Vernay et al., 2022) can be used to estimate the 3 simulation years chosen (2017-2018, 2018-2019, 2019-2020) in term of inter-annual variability. The reanalysis shows that the 3 years chosen capture most of the inter-annual variability found in air temperature, total

precipitation, and fraction of solid precipitation. Looking at the yearly solid precipitation of the Safran forcing shows the 2018-2019 season was low on solid precipitation with 21% less snowfall compared to 2017-2018. Wind speed is not evaluated in this study but we can compare the wind statistics for the 3 years. In Table 1 and Figure 3 we can see different statistics of the wind used in our simulations. For the 3 years, the mean, median, and 10% quantile are similar. On the 90 and 99% quantile, we see the 2017-2018 and 2018-2019 years are similar. The 2019-2020 values are slightly higher so we can say the 2019-2020 year was a winder year, not 2018-2019. The exact causes of the increase in variance in Fig. 9 in 2018-2019 are not fully determined by our analysis but it is not increased wind. We note that we observe an earlier SMOD for this season at all altitude bands, consistent with a smaller snow accumulation and a smaller amount of solid precipitations.

> We added this result in section 3.2.2

[Figure]

Figure 3: Illustration of the inter-annual wind speed variability for the wind used in our simulations.

Table 1: Wind speed statistics for the simulation period (2017-2020)

|  | 2017-2018 | 2018-2019 | 2019-2020 |
|---|---|---|---|
| Mean wind speed (m/s) | 1.86 | 1.88 | 1.93 |
| Median wind speed (m/s) | 1.36 | 1.37 | 1.37 |
| Quantile 10% (m/s) | 0.43 | 0.43 | 0.42 |
| Quantile 90% (m/s) | 3.94 | 4.01 | 4.14 |
| Quantile 99% (m/s) | 7.71 | 7.72 | 8.38 |

**Comment 27**

P29 L 586-587 "unrealistic snow interactions simulated at the borders of such masked areas" do you need to buffer the masked area then?

**Response:** Thank you for the comment.

In theory, we would need to add a buffer on the edge of the forest mask. Snow transport is strongly inhibited in forested areas leading to the creation of snow accumulation at the forest edges (Bernhardt et al., 2012). In our case, at the 250 m resolution and because forested areas of our domain are located at low elevations where simulated snow transport is minimal or non-existent, we neglected this contribution.

We added clarity about this process in the text.

> L669-673 For masked forest pixels, we know that snow transport is strongly inhibited in forested areas leading to the creation of snow accumulation at the forest edges (Bernhardt et al., 2012). It is not possible to account for this phenomenon solely through post-processing masks. In our simulation domain, the main forested areas are located at low elevations where snow transport is minimal or non-existent.

**Comment 28**

P30 first paragraph. As a non-expert in this model, I am unsure where the soil properties come into play and if it is worth mentioning. Seems a bit out of place.

**Response:** Thank you for the comment.

Soil properties and vegetation covers have mainly an impact on the bottom heat flux at the beginning of the snow season. This flux has a spatial variability due, among others to spatial variability of soil and vegetation cover. However, we agree with the reviewer that this is not a key factor in our analyses and prefer to remove this statement from the text.

> Removed statements on soil variability.

**Comment 29**

P31 L621 "SPS scores are found more homogeneous and smaller using landforms grouping than elevation bands" to me this conclusion isn't sufficiently explained and is hard to see flicking between the different graphs. This is another example of where I think some comparative metrics could really help support your statements, e.g. a comparison of the mean SPS scores for landforms vs elevation bands would clearly quantify that the landform SPS values are more homogeneous.

**Response:** Thank you for the comment.

We added Figure 13, a comparative metric showing the mean SPS value for landform and elevation bands. SPS values are lower when using landform grouping (thus closer to the observations). The SPS variability for different observations is found lower with elevation bands, but the SPS variability for different precipitation datasets is greater. We have added a description of this additional material in the result sections and have rewritten the discussion to improve clarity.

> L630-643 In Sect. 4, results are presented using landform classification instead of elevation bands. The results are partly correlated because summits and ridges mainly cover the highest elevations of the domain (Fig. 3). Thus, we consistently obtain that snow transport mainly affects snow height and SMOD on the upper elevation bands and for geomorphons 'Peaks (summit)' and 'Ridge'. However, some conclusions significantly differ between both space classifications, especially for the $\sigma$ ratio score and the SPS. A notable difference is found in the SPS snow height comparison. SPS scores are found to show lower variability

for the different precipitation forcings and overall smaller SPS values using landforms grouping than elevation bands (Fig. 11, Appendix Fig. D1 (c-f)). This result can be supplemented with Figure 13 showing the mean SPS value being consistently better using landform groups. This suggests simulated and observed distributions are closer when grouped using landforms than elevation bands. Indeed, for the landform grouping, each different landform distribution includes a relatively large spatial variability (intra-class variance) mostly due to elevation, slope, aspect, and precipitation variability (Clark et al., 2011; Freudiger et al., 2017). On the other hand, using elevation grouping the intra-class variance is mostly caused by other processes than elevation. This might suggest that our simulations better capture the altitudinal variability of snow than the other processes and the topographic-dependent variability. This result clearly illustrates the sensitivity to the choice of grid cell classification groups of the scores used.

**Conclusions**

> **Comment 30**
>
> P31 L646 "The addition of snow transport to simulations provides a better estimate of snow height across elevations" What does 'better' mean? If you mean closer to the observations, I don't think you can say this without quantifying it. If you mean 'more physically realistic regardless of precipitation forcing', then say that.

**Response:** Thank you for the comment.

Indeed, the word "better" was too vague in the conclusion. In this sentence, it was meaning both closer to the observations as quantified by Figure 15, and more physically realistic (Fig. 6, 8, 10). We have rewritten this section in the conclusion to remove the ambiguity.

> L702-706 Simulations without snow transport are found unable to capture the spatial variability of snow cover in alpine terrain. The addition of SnowPappus' snow transport in the simulations results in a more physically realistic assessment of snow height and SMOD, regardless of precipitation forcing, and leads to simulated snow height and SMOD closer to the observations. Furthermore, the use of the SnowPappus model increases the

variance of simulated snow height regardless of precipitation forcing bias.

**Comment 31**

P32 L654 "relatively close", vague and unclear. Quantifying this would give me much more confidence in your results.

**Response:** Thank you for the comment.

We have added quantification in the discussion Sec. 6.3 using Appendix Figure F1. This figure represents the mean SPS across all elevations for the SAFRAN and SAFRAN HR simulations. It shows a very small mean SPS change with resolution. The mean SPS difference is found to be lower than 0.01 m for the Pléiades observations and lower than a day (0.98) for SMOD. We also clarified the text in the conclusion.

L714-717 The changes in simulation snow height and SMOD mean SPS when increasing computation resolutions from 250 m to 30 m are minimal. For Pléiades observations, the change is less than 0.01 m, and for SMOD, it is less than 1 day. These changes in spatial variance and bias are also lower than the simulated spatial variability due to precipitation inputs and the impact of blowing snow in the affected areas.

**Authors' Response to technical comments**

**Comment 1**

P25 L464-465 replace "mainly limited in areas..." to "most significant in areas..".

**Response:** Thank you for the comment.

We agree with this remark, the text has been changed accordingly.

**Comment 2**

P3 L60 delete "way"

**Response:** Thank you for the comment.

We agree with this remark, the text has been changed accordingly.

**Comment 3**

P3 L73-74 "In order to assess the value of a distributed snow model in the presence of uncertainty in the meteorological forcing, this uncertainty has to be accounted for to raise robust conclusions." Bad phrasing. Consider replacing with e.g. "To robustly assess the value of a distributed snow model, the uncertainty in meteorological forcing must be considered".

**Response:** Thank you for the comment.

We agree with this remark, the text has been changed accordingly.

**Comment 4**

P8 L147 Grammar/typo "in the $10^5$ km2".

**Response:** Thank you for the comment.

We agree with this remark, the text has been rephrased.

> L150-152 SnowPappus can run over multiple snow seasons and very large domains (about $10^5$ km$^2$ at 250 m resolution or $10^6$ simulation points), in a reasonable computing time (less than half a day on a single computing node, depending on simulation outputs).

**Comment 5**

P8 L172 grammar/typo "... we only consider in our simulations only the precipitation amount from AROME." Change to " we only consider the precipitation amount from AROME in our simulations." Or similar.

**Response:** Thank you for the comment.

We agree with this remark, the text has been rephrased accordingly.

> L176[...] we only consider the total precipitation amount (liquid and solid) from AROME in our simulations.

**Comment 6**

P10 L208 "SBSM- like equations" don't use acronyms without explaining.

**Response:** Thank you for the comment.

We agree with this remark, the text has been changed accordingly.

> L212 Simplified Blowing Snow Model-like equations

**Comment 7**

P11 L245 Don't start paragraph with "Because...".

**Response:** Thank you for the comment.

We agree with this remark, the text has been changed accordingly.

> L251 Due to the method used to reconstruct the snow height map,...

**Comment 8**

P11 L 248 – 249 "...reduces the vertical snow height standard error between 0.3 and 0.4 m." missing 'to' or 'by' so it is unclear to the non-specialist if this is the error on the observations or the reduction in error.

**Response:** Thank you for the comment.

We agree with this remark, the text has been changed accordingly.

> L254 [...] averaging 2 m Pléiades observation resolution to 250 m reduces the vertical snow height standard error to approximately 0.3 and 0.4 m.

**Comment 9**

P12 L275

**Response:**

We didn't understand the remark on this line. The text remains unchanged.

**Comment 10**

P12-13 I think Eq3 should be better integrated in text.

**Response:** Thank you for the comment.

We agree with this remark, equation 3 moved up in the paragraph and integrated into the text.

**Comment 11**

P14 L314-315 "The mean simulated snow height for the 2500 to 2800m elevation band (1.90 m) is consistent with observations on 13 May 2019." Seems like an out of place statement.

**Response:** Thank you for the comment.

We agree with this remark, we removed this phrase.

**Comment 12**

P17 L336 "...(Fig.7 and Fig. 8)..." figures referred to unnecessarily.

**Response:** Thank you for the comment.

We agree with this remark, we removed this text from the sentence.

> L345 The second snow height evaluation is done in the Galibier Mountains area.

**Comment 13**

P17 L342 "(from 0.16 m for SAFRAN to 0.32 m for AROME)" clarify if this is the standard deviation or the magnitude of underestimation of standard deviation.

**Response:** Thank you for the comment.

We agree with this remark, the text has been changed accordingly.

> L350 [...] the standard deviation is underestimated by all simulations (0.71 m for the reference Pléiades observation, 0.16 m for SAFRAN and, 0.32 m for AROME)

> **Comment 14**
>
> P20 L380 "(orange dashed and continuous lines are less distant than green dashed and continuous 380 lines)" discussing figures without referring to which ones.

**Response:** Thank you for the comment.

We agree with this remark, the text has been changed accordingly.

> L390 ( Fig. 6, 8, orange dashed and continuous lines are almost always either less distant or equivalent to the green dashed and continuous lines).

> **Comment 15**
>
> P25 L461 "not shown here" Don't talk about results that you are not showing.

**Response:** Thank you for the comment.

We agree with this remark, the "not shown here" statement is clumsy. However, we believe that this sentence is useful in explaining the result Fig.11 so the sentence has been reformulated.

> L488 Directly examining the SAFRAN HR cumulative distribution function reveals that the shape of the cumulative distribution is more similar to the observation, despite being biased.

> **Comment 16**
>
> P25 L465 "the impact on SMOD is mainly limited in areas classified as 'Peak (summit)' or 'Ridge'." I think 'in' needs to be replaced by 'to'.

**Response:** Thank you for the comment.

We agree with this remark, the text has been changed accordingly.

> L482 As for snow height in Fig. 11, the impact on SMOD is most significant in areas classified as 'Peak (summit)' or 'Ridge'.

**Comment 17**

P25 L474 "standard deviation radio around 0.5-0.75." typo and consider replacing 'around' with 'of'. "standard deviation ratio of 0.5-0.75."

**Response:** Thank you for the comment.

We agree with this remark, the text has been changed accordingly.

**Comment 18**

P25 L480 "On the contrary to Fig. 11, the SAFRAN HR does not have the lowest SPS values", the opposite is true, SAFRAN HR has the highest SPS. Worth stating.

**Response:** Thank you for the comment.

We agree with this remark, the text has been changed accordingly.

> L498 the SAFRAN HR simulations have the overall highest SPS values

**Comment 19**

P29 L567 "On the other hand..." remove or reword.

**Response:** Thank you for the comment.

We agree with this remark, "On the other hand..." has been removed.

P31 L623 "Thus for Grandes Rousses and Galibier areas, we can again conclude that the observed snow height variability due to elevation is better captured than the variability due to landform." I think this is misworded. Unless I've misunderstood, the opposite is true: landform variability is better captured that elevation.

**Response:** Thank you for the comment.

This comment is linked to minor Comment 29. We disagree with this comment. Looking at SPS values shows lower variability for the different precipitation forcings and overall smaller SPS values using landforms grouping than elevation bands. Averaging the SPS values for the different precipitation sources like in Figure 13 shows SPS value being consistently better (lower) using landform groups.

When data is grouped using landforms, for each landform type, the inner distribution variability is mostly caused by the different points of different elevation, slope, aspect, and the precipitation variability. When grouped in elevation bands, the inner variance of each elevation band distribution is the result of the various slopes, aspects, precipitation variability, and topographic-dependent variability. The fact that SPS values are consistently lower for landform grouping suggests that our simulations better capture the altitudinal variability of snow (appearing in distributions for the landform grouping) than the other processes and the topographic-dependent variability (that remain in distributions for the elevation bands grouping).

We have improved the explanation and clarity of this by rewriting the discussion on this subject.

L637-643 [...] This result can be supplemented with Figure 13 showing the mean SPS value being consistently better using landform groups. This suggests simulated and observed distributions are closer when grouped using landforms than elevation bands. Indeed, for the landform grouping, each different landform distribution includes a relatively large spatial variability (intra-class variance) mostly due to elevation, slope, aspect, and precipitation variability (Clark et al., 2011; Freudiger et al., 2017). On the other hand, using elevation grouping the intra-class variance is mostly caused by other processes than elevation. This might suggest that our simulations better capture the altitudinal variability of snow than the other processes and the topographic-dependent variability. This result clearly illustrates the sensitivity to the choice of grid cell classification groups

of the scores used.

> **Comment 21**
>
> P31 L 641 "and in between two snow seasons" confusing statement, should be "between two dates in different snow seasons". 'in between' suggests it was dates outside of the snow season.

**Response:** Thank you for the comment.

We agree with this remark, the text has been changed accordingly.

**References**

[1] M. Bernhardt et al. 'The influence of lateral snow redistribution processes on snow melt and sublimation in alpine regions'. In: *Journal of Hydrology* 424–425 (Mar. 2012), pp. 196–206. ISSN: 0022-1694. DOI: `10.1016/j.jhydrol.2012.01.001`.

[2] M. P. Clark et al. 'Representing spatial variability of snow water equivalent in hydrologic and land-surface models: A review'. In: *Water Resour. Res.* 47 (2011), W07539. DOI: `10.1029/2011WR010745`.

[3] César Deschamps-Berger et al. 'Snow depth mapping from stereo satellite imagery in mountainous terrain: evaluation using airborne laser-scanning data'. In: *The Cryosphere* 14.9 (2020), pp. 2925–2940.

[4] Daphné Freudiger et al. 'Snow redistribution for the hydrological modeling of alpine catchments'. en. In: *WIREs Water* 4.5 (2017), e1232. ISSN: 2049-1948. DOI: `10.1002/wat2.1232`.

[5] S. Gascoin et al. 'Theia Snow collection: high-resolution operational snow cover maps from Sentinel-2 and Landsat-8 data'. In: 11.2 (2019), pp. 493–514. DOI: `10.5194/essd-11-493-2019`. URL: `https://essd.copernicus.org/articles/11/493/2019/`.

[6]  R. Marti et al. 'Mapping snow depth in open alpine terrain from stereo satellite imagery'. English. In: *The Cryosphere* 10.4 (July 2016), pp. 1361–1380. ISSN: 1994-0416. DOI: `10.5194/tc-10-1361-2016`.

[7]  Masataka Okabe and Kei Ito. *Color Universal Design (CUD) - How to make figures and presentations that are friendly to Colorblind people -*. English. 2002. URL: `https://jfly.uni-koeln.de/color/`.

[8]  M. Vernay et al. 'The S2M meteorological and snow cover reanalysis over the French mountainous areas: description and evaluation (1958–2021)'. In: *Earth System Science Data* 14.4 (2022), pp. 1707–1733. DOI: `10.5194/essd-14-1707-2022`. URL: `https://essd.copernicus.org/articles/14/1707/2022/`.

[9]  V. Vionnet et al. 'Snow Level From Post-Processing of Atmospheric Model Improves Snowfall Estimate and Snowpack Prediction in Mountains'. In: 58.12 (2022), e2021WR031778. DOI: `10.1029/2021WR031778`. eprint: `https://agupubs.onlinelibrary.wiley.com/doi/pdf/10.1029/2021WR031778`. URL: `https://agupubs.onlinelibrary.wiley.com/doi/abs/10.1029/2021WR031778`.

---

## Author Comment (AC2)

**Responses to Anonymous Referee #1**

Addressed Comments for Publication to

The Cryosphere

by

Ange HADDJERI,

Matthieu BARON,

Matthieu LAFAYSSE,

Louis LE TOUMELIN,

César DESCHAMPS-BERGER,

Vincent VIONNET,

Simon GASCOIN,

Matthieu VERNAY,

and Marie DUMONT

Dear Masashi Niwano,

Please find enclosed the detailed responses to Anonymous Referee #1 for the manuscript entitled 'Exploring the sensitivity to precipitation, blowing snow, and horizontal resolution of the spatial distribution of simulated snow cover' with manuscript number EGUSPHERE-2023-2604. We would like to thank you and the reviewers for the valuable comments which help improving the quality of our manuscript. In this revision, we have carefully addressed the reviewers' comments. A summary of the main modifications and a detailed point-by-point response to the comments from Reviewer #1 (following the reviewers' order in the decision letter) are given below.

Sincerely,

Ange HADDJERI,
Matthieu BARON,
Matthieu LAFAYSSE,
Louis LE TOUMELIN,
César DESCHAMPS-BERGER,
Vincent VIONNET,
Simon GASCOIN,
Matthieu VERNAY,
and Marie DUMONT

**Note:** To enhance the legibility of this response letter, all the editor's and reviewers' comments are typeset in boxes. Rephrased or added sentences are typeset in color. The respective parts in the manuscript are highlighted to indicate changes.

**Authors' Response to Reviewer 1**

> **General Comments.** This paper has carried out the validation of simulated snow cover distribution in alpine areas using a combination of state-of-the-art techniques. Also, some concerns I had while reading are already explained in the discussion chapter. In my opinion, there are no problems with publishing the paper in TC. I have summarized some of the concerns in the minor comments below. This is not a requirement for acceptance. Please use them as a reference to improve the content in the revised draft.

**Response:**

Thank you for your positive assessment of our manuscript and your recommendation for publication in TC. We appreciate your time and valuable feedback.

We carefully considered the minor comments you have outlined and addressed them bellow.

**Authors' Response to minor comments**

> **Comment 1**
>
> Fig. 5: In case of "with transport", there are some areas with locally large snow depths. This is probably a depression or downwind slope, but are these localized areas of high snow cover consistent with those observed by the satellite? If it has also been confirmed at visual level or already verified by Baron et al (2023) etc., it would be good to have a mention of this. In section 4.3.3 (L626-635), it is written that the simulation shows a larger snow depth in the depression, it would be good to write whether the large snow depth area is consistent with the satellite as well.

**Response:** Thank you for the comment.

The fair localization of snow accumulation is demonstrated in Sect.5.6 of Baron et al., 2024 using Pearson correlation. First, the analysis of our simulations' spatial patterns shows that the accumulation zones correspond to the areas sheltered from the wind and ablation zones to the most exposed zones, in line with the theory. The main modeled accumulation zones are well located in high-observed snow height areas (e.g. Figure 4). However, the reciprocal is not exact: many areas of high observed snow accumulation are not present in the simulations. Indeed, the 2 m high-resolution image shows a complexity of spatial variability (mainly caused by the high-resolution topography) that is only approached in the 250 m simulation. Even in the Pléiades observation aggregated to 250 m, the impact of the sub-mesh topography can be felt and these sub-mesh accumulation zones are not simulated. (see discussion 6.6)

We added a sentence describing the consistency between simulated snow accumulation and the observation. We also added observation to simulations scatter plots (Fig. 14) and Pearson correlation coefficient (Tables 1 & 2) which helps demonstrate this consistency.

> Added Fig. 14, Table 1 & 2
>
> L299-L302 It can be observed that the primary areas of model snow accumulation are located in regions of high observed snow height. However, the opposite is not true, many areas with high observed snow accumulation are not present in the simulations. Additional maps for the 13 May 2019 comparing resolutions can be found in Appendix Fig. C1

> ### Comment 2
>
> 3.1.1 Figure 6: Figure 6 shows the results of the bias and other validation results. Also, I thought it would be easier to visualize the degree of agreement if there was a scatter diagram from the snow depth data at each location, with the snow depth from the satellite on the horizontal axis and the simulation snow depth on the vertical axis.

**Response:** Thank you for the comment.

Initially, we chose to use the graphs summarised in Figs. 6, 8, and 10 to get straight to the point and reduce the number of graphs. At the request of both reviewers, we have added the different scatter plots for the Pléiades observation in Fig. 14, E1, E2, E3. Those figures allow for a visualization of the degree of pixel-to-pixel agreement between observations and simulations. Additionally, quantification of this agreement is added using Pearson correlation coefficients computed in Tables 1 and 2. It is important to remember that direct pixel-to-pixel evaluations are extremely challenging from this kind of spatialized model and are generally not promoted in the literature (Vionnet et al., 2021; Sharma, Gerber and Lehning, 2021; Quéno et al., 2023). However, we agree that these diagnostics help to provide a fair overview of the limitations of currently available snow modeling systems.

> Added scatter plot Fig. 14, E1, E2, E3, Tables 1 and 2 and discussion of Pearson correlation coefficient.

> ### Comment 3
>
> L392-395 As the result of Figure 9, the difference between years states that 2018-19 showed wide spatial variability, but what are the differences between these two years in terms of characteristics of the weather conditions? For example, was it a windier winter in 2018-19?

**Response:** Thank you for the comment.

The S2M (SAFRAN–SURFEX/ISBA–Crocus–MEPRA) meteorological and snow cover reanalysis from 1958 to 2021 (Vernay et al., 2022) can be used to estimate the 3 simulation years chosen (2017-2018, 2018-2019, 2019-2020) in term of inter-annual variability. The reanalysis shows that

the 3 years chosen capture most of the inter-annual variability found in air temperature, total precipitation, and fraction of solid precipitation. Looking at the yearly solid precipitation of the Safran forcing shows the 2018-2019 season was low on solid precipitation with 21% less snowfall compared to 2017-2018. Wind speed is not evaluated in this study but we can compare the wind statistics for the 3 years. In Table 1 and Figure 1 we can see different statistics of the wind used in our simulations. For the 3 years, the mean, median, and 10% quantile are similar. On the 90 and 99% quantile, we see the 2017-2018 and 2018-2019 years are similar. The 2019-2020 values are slightly higher so we can say the 2019-2020 year was a winder year, not 2018-2019. The exact causes of the increase in variance in Fig. 9 in 2018-2019 not precisely determined by our analysis is not increased wind. We note that we observe an earlier SMOD for this season at all altitude bands, consistent with a smaller snow accumulation and a smaller amount of solid precipitations.

L426-L429 Analysis of wind speed and direction over the three simulation years indicates that the values remained consistent. Examination of the yearly solid precipitation of the SAFRAN forcing reveals that the 2018-2019 season had 21% less snowfall compared to 2017-2018. This result is consistent with a smaller amount of solid precipitation leading to a reduced amount of snow accumulation and the observed earlier SMOD for the 2018-2019 snow season.

|  | 2017-2018 | 2018-2019 | 2019-2020 |
|---|---|---|---|
| Mean wind speed (m/s) | 1.86 | 1.88 | 1.93 |
| Median wind speed (m/s) | 1.36 | 1.37 | 1.37 |
| Quantile 10% (m/s) | 0.43 | 0.43 | 0.42 |
| Quantile 90% (m/s) | 3.94 | 4.01 | 4.14 |
| Quantile 99% (m/s) | 7.71 | 7.72 | 8.38 |

[Figure]

Figure 1: Illustration of the inter-annual wind speed variability for the wind used in our simulations.

**Comment 4**

Fig. 12: It compares simulation data and satellite data in 2017-2018 and 2018-2019. Comparing these figures, 2017-2018 (a-c) and 2018-2019 (d-f) seems to be the same figure, is it possible that you have misplaced the figures?

**Response:** Thank you for the comment.

We agree with this comment and apologize for this mistake. We therefore corrected the figure in the new version of the manuscript and checked references to Figure 12 in the manuscript.

> The upper panel of Figure 12 (a-b-c) now rightly corresponds to the 2017-2018 year.

**References**

[1] M. Baron et al. 'SnowPappus v1.0, a blowing-snow model for large-scale applications of the Crocus snow scheme'. In: *Geoscientific Model Development* 17.3 (2024), pp. 1297–1326.

DOI: 10.5194/gmd-17-1297-2024. URL: https://gmd.copernicus.org/articles/17/1297/2024/.

[2] L. Quéno et al. 'Snow redistribution in an intermediate-complexity snow hydrology modelling framework'. In: *EGUsphere* 2023 (2023), pp. 1–32. DOI: 10.5194/egusphere-2023-2071. URL: https://egusphere.copernicus.org/preprints/2023/egusphere-2023-2071/.

[3] V. Sharma, F. Gerber and M. Lehning. 'Introducing CRYOWRF v1.0: Multiscale atmospheric flow simulations with advanced snow cover modelling'. In: *Geoscientific Model Development Discussions* 2021 (2021), pp. 1–46. DOI: 10.5194/gmd-2021-231. URL: https://gmd.copernicus.org/preprints/gmd-2021-231/.

[4] M. Vernay et al. 'The S2M meteorological and snow cover reanalysis over the French mountainous areas: description and evaluation (1958–2021)'. In: *Earth System Science Data* 14.4 (2022), pp. 1707–1733. DOI: 10.5194/essd-14-1707-2022. URL: https://essd.copernicus.org/articles/14/1707/2022/.

[5] V. Vionnet et al. 'Multi-scale snowdrift-permitting modelling of mountain snowpack'. In: *The Cryosphere* 15.2 (2021), pp. 743–769. DOI: 10.5194/tc-15-743-2021. URL: https://tc.copernicus.org/articles/15/743/2021/.

---

## Author Response (AR2)

**Responses to Anonymous Referee #2**

Addressed Comments for Publication to

The Cryosphere

by

Ange HADDJERI,

Matthieu BARON,

Matthieu LAFAYSSE,

Louis LE TOUMELIN,

César DESCHAMPS-BERGER,

Vincent VIONNET,

Simon GASCOIN,

Matthieu VERNAY,

and Marie DUMONT

Dear Masashi Niwano,

Please find enclosed the detailed responses to Anonymous Referee #2 for the manuscript entitled 'Analysing the sensitivity of a blowing snow model (SnowPappus) to precipitation forcing, blowing snow, and spatial resolution' with manuscript number EGUSPHERE-2023-2604. We would like to thank you and both reviewers for the valuable comments that helped improving the quality of our manuscript.

Sincerely,

Ange HADDJERI,

Matthieu BARON,

Matthieu LAFAYSSE,

Louis LE TOUMELIN,

César DESCHAMPS-BERGER,

Vincent VIONNET,

Simon GASCOIN,

Matthieu VERNAY,

and Marie DUMONT

**Note:** To enhance the legibility of this response letter, all the editor's and reviewers' comments are typeset in boxes. Rephrased or added sentences are typeset in color. The respective parts in the manuscript are highlighted to indicate changes.

**Authors' Response to Reviewer 2**

**General Comments.**

The authors have taken into account my comments and made substantial changes. I believe this results in a much improved manuscript that more clearly quantifies the improvement offered by the SnowPappus model. I recommend to publish if the authors address the corrections listed below.

I list 'Technical corrections' which must be addressed before publication followed by 'Minor comments for consideration'.

**Response:**

Thank you for the positive assessment of our manuscript changes and your previous recommendation for publication. We appreciate your time and valuable feedback.

We carefully considered the Technical corrections and Minor comments you have outlined. We have followed all your Technical corrections and addressed your minor comments.

**Authors' Response to technical comments**

> **Comment 1**
>
> Section 3.4 P31 L539 "(more points at high and low snow heights)" this isn't true. It is more disperse, yes, meaning a greater variance, less consistent/reliable.

**Response:** Thank you for the comment.

We agree with this remark, we removed this part of the sentence for better clarity and changed the text accordingly.

> For each simulation, we see more dispersed scatter plots for simulations with snow transport (greater variance).

> **Comment 2**
>
> Section 4.1 P35 L570 "Looking at the Pléiades bias values in Fig.6 and Fig. 8 (a) and (d), blowing snow reduces the absolute mean bias value of snow height in all simulations." No it doesn't. Fig 6a both SAFRAN simulations, 6d both SAFRANs and ANTILOPE, 8a ANTILOPE, 8d ANTILOPE. Perhaps you have jus misphrased it because the bias do all lower in values, but a more negative bias is not a reduced absolute mean bias. Clarify what you mean.

**Response:** Thank you for the comment.

We've changed the sentence to make it clearer to understand.

> Looking at the Pléiades bias values in Fig. 6 and Fig. 8 (a) and (d), blowing snow lowers in value the mean bias of all simulations with snow transport compared to simulations without snow transport.

**Comment 3**

L432 grammar "...and every the elevation bands."

**Response:** Thank you for the comment.

We agree with this remark and changed the sentence for better understanding.

> Continuing the analysis, the observed median SMOD shows an earlier melt at all elevation for the 2018-2019 season.

**Comment 4**

Inconsistent use of words and numerals to represent numbers, e.g. P42 L790: "4 different dates over 3 different snow seasons. Eight different snow simulations.."

**Response:** Thank you for the comment.

We agree with this remark and changed the text accordingly.

> The Pléiades comparisons were conducted in two different areas, on four different dates and during three different snow seasons.

**Authors' Response to minor comments**

**Comment 1**

Section 3.1.4 p 17 L411-414 – don't understand what you are trying to say here, worth making clearer.

**Response:** Thank you for the comment.

We agree with this remark and improve the sentence to make the paragraph clearer.

> In Fig. 6, 8, the difference in the different scores between a simulation with and without snow transport is smaller for simulations computed at a 30 m resolution, compared to the 250 m computed simulations (orange dashed and continuous lines are less distant or equivalent to the green dashed and continuous lines).

**Comment 2**

Figure 14 and tables 3 and 4:

- Not keen on the way you've presented this, you could have one legend for all 16 subplots and fit the correlation values on the graphs.

- Figure 14 doesn't really support that SnowPappus is improving the simulated snow height. As you say in L 545, correlation decreases. Would be more supportive if you did these for your Z>2700m.

- Table 3 and 4. Why not one overall summary statistic? This more clearly separates the impact SnowPappus from the precipitation forcing, assuming your choice of precipitation forcings is representative.

**Response:** Thank you for the comment.

- We agree with this comment, we changed the Figure 14 legend so it appears once and added Pearson correlation values in each plot.
- We agree with this comment. Therefore, we have modified Fig. 14 and Fig. 15 to better support our result and clearly show that SnowPappus improves the simulated snow height

distribution for both pixels with elevation > 2700m and pixels >0.

Firstly, fig. 14 was added to the manuscript at the request of both reviewers in the previous review round. The aim was to show the degree of agreement between observations and simulations directly from the dataset. From this graph and Tables 3 and 4, we can see that the correlation slightly deteriorates with the addition of transport (looking at all heights). However, as shown in Tables 3 and 4, by restricting the points to transport zones to elevations >2700m, the correlation values are better in the snow transport simulations, but much lower. Given the low correlation values and the following analysis, we felt that there was no great added value in showing a restricted part of the dataset in the scatter plots. However, you can find the scatter plots you requested for points with elevations > 2700m in Fig. 1 below.

Secondly, in the following part of the manuscript we summarise the effect of snow transport on simulations mainly for areas >2700m using person correlation coefficients (Tables 3 and 4) and SPS values (Fig. 15). We agree that Fig. 14 looking at all elevation shows a reduction in correlations and is not very supportive of the model's effect on snow depth. To demonstrate more clearly the impact of SnowPappus at all elevations, we extended the analysis of Figure 15 to also show SPS values for the entire snow depth distribution (all elevations). The new version of Figure 15 now shows that the snow transport model improves the representation of the snow depth distribution for all observations for points with an elevation > 2700m, while not degrading the snow depth distribution, looking at the entire domain.

- In this manuscript, we show that spatial correlation plays an important role in the spatial variability of snow. We found it interesting to highlight this dependence on the source of precipitation in the tables. Figure 15 shows an overall summary statistic of all the precipitation sources.

[Figure]

Figure 1: Observed vs simulated snow height scatter plots for the four different Pléiades observations (for elevation > 2700m) and the different precipitation forcings. For each precipitation forcing, the simulation with snow transport is in blue and without transport in red. Linear regression line synthesizes distribution changes when adding snow transport.